# Does Stochastic Gradient really succeed for Bandits?

**Dorian Baudry**[1,2]          **Emmeran Johnson**[3]          **Simon Vary**[2]

**Ciara Pike-Burke**[3]          **Patrick Rebeschini**[2]

[1] Univ. Grenoble Alpes, Inria, CNRS, Grenoble INP, LIG, 38000 Grenoble, France[1]
[2] Department of Statistics, University of Oxford
[3] Department of Mathematics, Imperial College London

## Abstract

Recent works of Mei et al. [1, 2] have deepened the theoretical understanding of the *Stochastic Gradient Bandit* (SGB) policy, showing that using a constant learning rate guarantees asymptotic convergence to the optimal policy, and that sufficiently *small* learning rates can yield logarithmic regret. However, whether logarithmic regret holds beyond small learning rates remains unclear. In this work, we take a step towards characterizing the regret *regimes* of SGB as a function of its learning rate. For two–armed bandits, we identify a sharp threshold, scaling with the sub-optimality gap $\Delta$, below which SGB achieves *logarithmic* regret on all instances, and above which it can incur *polynomial* regret on some instances. This result highlights the necessity of knowing (or estimating) $\Delta$ to ensure logarithmic regret with a constant learning rate. For general $K$-armed bandits, we further show the learning rate must additionally scale inversely with $K$ to avoid polynomial regret. We introduce novel techniques to derive regret upper bounds for SGB, laying the groundwork for future advances in the theory of gradient-based bandit algorithms.

## 1 Introduction

A $K$-armed bandit is a sequential decision-making problem where, at each time $t \in [T]$, a learner chooses an action $A_t \in [K]$ and receives a reward $r_t$ drawn independently at random from the fixed distribution $\nu_{A_t}$. The objective is to select a policy $\pi$, which at each step $t$ maps past observations $\mathcal{H}_{t-1} = (A_s, r_s)_{s \in [t-1]}$ to a *sampling probability* $p_{k,t}^\pi = \mathbb{P}_\pi(A_t = k \mid \mathcal{H}_{t-1})$ for each arm $k \in [K]$, in order to maximize the expected cumulative reward $\mathbb{E}_\pi[\sum_{t=1}^T r_t]$ when actions are chosen according to $\pi$. This is equivalent to minimizing the *regret*, defined by

$$\mathcal{R}_T^\pi := \mathbb{E}_\pi \left[ \sum_{t=1}^T \sum_{k=1}^K \Delta_k \mathbb{1}(A_t = k) \right], \text{ with } \forall k, \ \Delta_k = \max_{j \in [K]} \mu_j - \mu_k \text{ and } \mu_k = \mathbb{E}_{r \sim \nu_k}[r], \quad (1)$$

$$\text{or equivalently} \quad \mathcal{R}_T^\pi = \mathbb{E}_\pi \left[ \sum_{t=1}^T \sum_{k=1}^K p_{k,t}^\pi \Delta_k \right] \leq \mathbb{E}_\pi \left[ \sum_{t=1}^T (1 - p_{1,t}^\pi) \right] \cdot \max_{k \in [K]} \Delta_k, \quad (2)$$

where for convenience we assume that arm 1 is the unique optimal arm, i.e., $\mu_1 > \max_{j \in \{2,...,K\}} \mu_j$. We further define the *minimum gap* $\Delta := \min_{k:\Delta_k > 0} \Delta_k$. Finally, we denote by $\mathcal{F}$ the family of reward distributions supported on $[-1, 1]$, and assume that $\nu_k \in \mathcal{F}$ for all $k \in [K]$.

**Notation** We now detail the symbols used throughout the paper. Let $(A, B) \in \mathbb{R}^2$, then: $A \wedge B = \min\{A, B\}$, $A \vee B = \max\{A, B\}$, and $(A)_+ = A \vee 0$. If $A, B$ depend on the horizon $T$, we use

---

[1]Corresponding author: dorian.baudry@inria.fr

39th Conference on Neural Information Processing Systems (NeurIPS 2025).

interchangeably $A \lesssim B$ or $A = \mathcal{O}(B)$ (resp. $A \gtrsim B$ and $A = \Omega(B)$) when there exists $c > 0$ s.t. $A \leq cB$ (resp. $\geq$) and $c$ is independent of $T$ but can be problem-dependent. To hide poly-logarithmic factors in $T$, we respectively use $\widetilde{\mathcal{O}}$ and $\widetilde{\Omega}$, for example $A \geq T/\log(T) \implies A = \widetilde{\Omega}(T)$. Finally, we use $A \asymp B$ when both $A \lesssim B$ and $B \lesssim A$ hold.

## 1.1 Motivation and related work

Stochastic gradient ascent methods have been extensively studied in the context of bandits and reinforcement learning, tracing back to foundational works such as Robbins and Monro's stochastic approximation [3], the REINFORCE algorithm [4], and subsequent developments in policy gradient (PG) methods [5]. Despite the empirical success of these algorithms in modern deep reinforcement learning [6–10], the theoretical understanding of their convergence and regret guarantees remains limited. This motivates a closer examination of their foundational components in simpler settings. We study one such component, the Stochastic Gradient Bandit algorithm (SGB) [11, Chapter 2.8], a softmax policy gradient method for Multi-Armed Bandits (MAB).

A central challenge in analyzing SGB stems from the weakness of its underlying optimization structure, making the analysis difficult even in the case of PG with access to exact gradient information [12]. The optimization challenge arises from the softmax parameterization satisfying only a non-uniform version of the Polyak–Łojasiewicz (PL) inequality [13, Lemma 3]. Consequently, if the sampling probability of the best arm becomes too small, the gradient signal vanishes and SGB may require an exponentially long time to recover, even with access to exact gradients [14, 15]. Nevertheless, PG with softmax parameterization and exact gradients has been shown to converge to a globally optimal policy asymptotically [16, Theorem 5.1], and with a $\mathcal{O}(1/T)$ rate [13], although the rate of convergence depends on suitable initialization and problem dependent constants [14, 15]. Convergence guarantees have also been obtained with regularization [12] or by modifying the softmax function [17].

The analysis of SGB with stochastic gradients becomes even more convoluted. Contrary to typical policies like UCB [18] or Thompson Sampling [19], the decisions of SGB depend intricately on the order in which all past rewards were collected, and thus cannot be analyzed through simple summary statistics; see Appendix A.3 for a detailed comparison with standard bandit policies. Despite these difficulties, a number of recent works are able to show global convergence of variants of SGB (and its generalization for Markov Decision Process) [20–27]. However, these result require decaying learning rate or regularization resulting in at best $\mathcal{O}(1/\sqrt{T})$ convergence rate [20–23]. Mei et al. [28] showed that natural policy gradient with oracle baselines achieves a $\mathcal{O}(1/T)$ convergence rate. A closely related algorithm to SGB is SAMBA [29], a policy that achieves logarithmic regret for Bernoulli rewards by performing stochastic gradient ascent directly on the sampling probabilities without relying on the softmax transformation; for a detailed comparison to SGB see Appendix B.2.

Recently, Mei et al. [1] established a regret upper bound of $\mathcal{O}(\log(T))$ for SGB with a *small* constant learning rate $\eta$ satisfying $\eta \lesssim \Delta^2/K^{3/2}$ where $K$ is the number of arms. In a follow-up work, Mei et al. [2] further proved that SGB asymptotically converges to a globally optimal policy for *any* constant learning rate. While asymptotic convergence is a desirable property, it does not guarantee favorable regret guarantees, as we discuss in Appendix A.4. This motivates further investigation of the regret properties of SGB without the small rate constraint.

The goal of our work is to characterize the strengths and limitations of SGB, as a principled yet simple and scalable learning rule; which is representative of algorithms that have shown empirical success in more complex, large-scale settings. Importantly, our aim is not to promote SGB over state-of-the-art bandit algorithms such as KL-UCB [30] or Thompson Sampling [31, 32]. To achieve our goal, in this work, we depart from optimization-based analyses of SGB and develop a regret-based analysis. Our analysis exploits a novel decomposition into a term that remains logarithmic for **all** learning rates, and a second term capturing the probability of *failure*–that is, the chance that the optimal arm fails to stand out–which crucially depends on $\eta$. This enables us to determine regimes where SGB succeeds.

## 1.2 The Stochastic Gradient Bandit policy (SGB)

Following Sutton and Barto [11, Section 2.8], we define SGB as a randomized policy, with sampling probabilities at time $t \geq 1$ given by the softmax transform of some parameters $(\theta_{k,t})_{k \in [K]} \in \mathbb{R}^K$,

$$\forall t \in [T], \ \forall k \in [K]: \quad p_{k,t}^{\text{SGB}} \propto \exp(\theta_{k,t}) \ . \tag{3}$$

In the following, we drop the superscript for simplicity of notation. Given a fixed learning rate $\eta$, the parameter of each arm $k \in [K]$ is initialized at $\theta_{k,1} = 0$, and is updated as follows,

$$\forall t \geq 1 : \; \theta_{k,t+1} = \theta_{k,t} + \eta \cdot \delta\theta_{k,t}, \; \text{with} \; \delta\theta_{k,t} := (1 - p_{k,t}) \cdot r_t \cdot \mathbb{1}(A_t = k) - p_{k,t} \cdot r_t \cdot \mathbb{1}(A_t \neq k) \; . \quad (4)$$

For completeness, we provide the pseudo-code of SGB in Algorithm 1 in Appendix A.1. In the remainder of the paper, we use the shorthand notation $\mathbb{E}_t[\cdot]$ for the conditional expectation $\mathbb{E}[\cdot | \mathcal{H}_{t-1}]$. From Equation (4), we derive the following first fundamental property of SGB,

$$\forall k \in [K], t \geq 1 : \; \mathbb{E}_t[\delta\theta_{k,t}] = p_{k,t}(1 - p_{k,t})\mu_k - p_{k,t} \sum_{j \neq k} p_{j,t}\mu_j = p_{k,t} \cdot (\mathbb{E}_t[\Delta_{A_t}] - \Delta_k) \; . \quad (5)$$

This equation allows to verify (see Appendix A.1) that SGB indeed performs a stochastic gradient ascent in order to maximize the *value* of the policy at time $t$, $V_t^{\text{SGB}} := \sum_{k=1}^{K} p_{k,t} \cdot \mu_k$, since $(\delta_{k,t})_{k \in [K]}$ is an unbiased estimate of $\nabla_\theta V_t^{\text{SGB}}$. It also shows that a parameter $\theta_{k,t}$ is expected to increase only if the gap of arm $k$ is smaller than $\mathbb{E}_t[\Delta_{A_t}] = \sum_{j=1}^{K} p_{j,t}\Delta_j$, the instantaneous regret of SGB. From (5), we further obtain that if arm 1 is the only optimal arm, the parameter of the best arm satisfies

$$\mathbb{E}[\theta_{1,T+1}] = \eta \cdot \mathbb{E}\left[\sum_{t=1}^{T} \delta\theta_{1,t}\right] = \eta \cdot \mathbb{E}\left[\sum_{t=1}^{T} p_{1,t}\mathbb{E}_t[\Delta_{A_t}]\right] \geq \eta \cdot \mathbb{E}\left[\sum_{t=1}^{T} p_{1,t}(1 - p_{1,t})\Delta\right] \quad (6)$$

which is an equality if all sub-optimal arms have the same gap $\Delta$. This allows us to express an upper bound on the regret as a function of the parameter $\theta_{1,T+1}$. Defining $\Delta_{\max} = \max_{k \in [K]} \Delta_k$, and using that $(1 - p_{1,t}) = p_{1,t}(1 - p_{1,t}) + (1 - p_{1,t})^2$, by plugging (6) into (2) we obtain that

$$\mathcal{R}_T \leq \Delta_{\max} \cdot \mathbb{E}\left[\sum_{t=1}^{T}(1 - p_{1,t})\right] \leq \underbrace{\frac{\Delta_{\max}}{\eta\Delta} \cdot \mathbb{E}[\theta_{1,T+1}]}_{\text{Post-convergence term}} + \underbrace{\Delta_{\max} \cdot \mathbb{E}\left[\sum_{t=1}^{T}(1 - p_{1,t})^2\right]}_{\text{Failure regret}} \; . \quad (7)$$

This regret decomposition, though obtained via a straightforward reformulation, appears to be novel and, to the best of our knowledge, has not been exploited in prior analyses of SGB. In fact, Equation (7) provides a powerful lens through which we develop all the upper bounds in this work, as it splits the regret into two interpretable components. In Section 3, we prove that the *post-convergence* term is bounded by $\mathcal{O}(\log T)$ for **any** learning rate. Therefore, whether SGB achieves logarithmic regret or not depends entirely on the scaling of the *failure regret*. We justify this terminology in Appendix F.1, where we discuss an alternative formulation of Eq. (7) establishing the relation between failure regret and the sum of probabilities that $p_{1,t}$ stays bounded away from 1. The empirical study in Section 4 further supports this decomposition: most trajectories of SGB fall into one of two distinct regimes–*successful runs*, with convergence to the optimal policy at rate $\mathcal{O}(t^{-1})$, and *failed runs*, where the optimal arm remains persistently under-sampled up to horizon $T$.

In addition to the above regret bound, we present in Appendix A.2 some other elementary properties of SGB that can be directly derived from its update rule (4), and will be useful in our analyses.

## 1.3 Outline and contributions

We introduced in Section 1.2 an original decomposition of the regret of SGB into a *post-convergence term* and a *failure regret*. In Section 2, we build on Eq. (7) to provide a tight regret analysis for two-armed bandits when $\eta \lesssim \Delta$ (Thm. 1). We then prove in Thm. 2 that the regret of SGB can be **polynomial** on some instances if $\eta \gtrsim \Delta$. Hence, we characterize the regret of SGB in two-armed bandits, except for a critical regime $\eta \approx \Delta$ that we discuss in Section 2.2. In particular, *knowing a lower bound on the gap $\Delta$ is necessary and sufficient to tune $\eta$ to guarantee logarithmic regret*. We summarize our results for the two-armed bandit problem in Table 1 below, denoting by $\mathcal{F}_\Delta^2$ the class of two-armed bandits with distributions supported on $[-1, 1]$ and gap larger than $\Delta \in (0, 1)$.

In Section 3, we consider a general number of arms $K$. Using lower bound arguments, we first prove a second necessary condition: logarithmic regret can be guaranteed *only* if $\eta$ **decreases linearly** in $K$ (Thm. 3). By combining this result with Thm. 2 we conjecture that the critical tuning of $\eta$ might be proportional to $\Delta/K$. For the regret analysis, we establish in Lemma 2 that the post-convergence term is logarithmic **for any constant learning rate**. Then, in Lemma 3, we derive an intermediate

| $\eta$ | $\leq \Delta - \mathcal{O}(\Delta^2)$ | $\approx \Delta$ | $\geq \Delta + \mathcal{O}(\Delta^2)$ |
|---|---|---|---|
| $\mathcal{R}_T$ | $\forall \nu \in \mathcal{F}_\Delta^2,$ $\mathcal{R}_T \leq \frac{\log(T)}{2\eta}$ (Thm. 1) | $\exists \nu_0 \in \mathcal{F}_\Delta^2$ s.t. $\mathbb{E}[(1 - p_{1,t})^2] \asymp t^{-1}$ (*Conjecture* from the proofs of Thm. 1 and 2) | $\exists \nu_1 \in \mathcal{F}_\Delta^2$ s.t. $\mathcal{R}_T = \widetilde{\Omega}\left(T^{1 - \frac{\Delta}{\eta(1-\Delta)}}\right)$ (Thm. 2) |
| Regime | **Logarithmic** | **Critical** | **Polynomial** |

Table 1: Regret guarantees for SGB as a function of $\eta$ for two-armed bandits

upper bound on the failure regret. The upper bound contains additional terms compared to the case $K = 2$, causing significant technical challenges that we detail, but leave their dedicated analysis for future work. Nonetheless, as a promising first complete result for $K$-armed problems, we prove in Theorem 4 that the regret of SGB is logarithmic if $\eta e^{2\eta} \leq \frac{2\Delta}{K+2}$, when *all arms have identical gap* $\Delta$.

Finally, in Section 4 we present some synthetic experiments that illustrate our theoretical findings.

## 2 Characterizing the regret regimes for two-armed bandits

In this section we propose a tight characterization of the regret regimes of SGB when $K = 2$. We exhibit a separation between logarithmic and polynomial regret close to the gap $\Delta$.

### 2.1 Regret analysis for $\eta \leq \Delta - \mathcal{O}(\Delta^2)$

We directly detail the main result of this section and its complete proof.

**Theorem 1** (Regret upper bound for $K = 2$). *For a two-armed bandit instance $\nu \in \mathcal{F}^2$ with a gap $\Delta \in (0, 1)$, the regret of SGB tuned with a learning rate $\eta$ satisfying $\eta C_\eta < \Delta$ is upper bounded by*

$$\mathcal{R}_T^{SGB} \leq \frac{\log(1 + 4\eta\Delta T)}{2\eta} + \frac{\Delta}{2\eta(\Delta - \eta C_\eta)} , \quad with \ C_\eta := 2\sum_{n=0}^{+\infty} \frac{(2\eta)^n}{(n+2)!} \leq e^{2\eta}.$$

*Proof.* Since $\theta_{2,t} = -\theta_{1,t}$ for all steps $t$, by (9), we drop the subscript 1 to simplify the notation. Starting from the regret decomposition of Equation (7), we readily obtain that for any instance $\nu$,

$$\mathcal{R}_T^{\text{SGB}} \leq \frac{\mathbb{E}[\theta_{T+1}]}{\eta} + \Delta \cdot \mathbb{E}\left[\sum_{t=1}^{T}(1 - p_t)^2\right] .$$

We start by upper bounding $\mathbb{E}[\theta_{T+1}]$. Jensen inequality provides that

$$\mathbb{E}[\theta_{T+1}] = \frac{1}{2} \cdot \mathbb{E}\left[\log\left(e^{2\theta_{T+1}}\right)\right] \leq \frac{1}{2} \cdot \log\left(\mathbb{E}\left[e^{2\theta_{T+1}}\right]\right) .$$

We then use the update rule of $(\theta_t)_{t \geq 1}$ to upper bound $\mathbb{E}[e^{2\theta_t}]$. From the Taylor expansion of the exponential, for any constant $q \in \mathbb{R}$ and random variable $r$ supported on $[-1, 1]$, it holds that

$$\mathbb{E}[e^{qr}] = 1 + q\mathbb{E}[r] + \sum_{n=2}^{+\infty} \frac{q^n}{n!}\mathbb{E}[r^n] \leq 1 + q\mathbb{E}[r] + q^2 \cdot \sum_{n=2}^{+\infty} \frac{q^{n-2}}{n!} \leq 1 + q\mathbb{E}[r] + \frac{q^2}{2} \cdot C_{\frac{|q|}{2}}. \quad (8)$$

Using the notation $a_t = 2\eta(1 - p_t)$ and $b_t = 2\eta p_t$, for any $t \geq 1$ Eq. (4) and (8) give that

$$\mathbb{E}_t[e^{2\theta_{t+1}}] = e^{2\theta_t} \cdot \mathbb{E}_t[e^{2\eta\delta\theta_t}] = e^{2\theta_t} \cdot \left(p_t\mathbb{E}_t[e^{a_t r_t}|A_t = 1] + (1 - p_t)\mathbb{E}[e^{-b_t r_t}|A_t = 2]\right)$$

$$\leq e^{2\theta_t} \cdot \left(1 + p_t \cdot \left(a_t\mu_1 + a_t^2 \cdot 0.5 \cdot C_{a_t/2}\right) + (1 - p_t) \cdot \left(-b_t\mu_2 + b_t^2 \cdot 0.5 \cdot C_{b_t/2}\right)\right)$$

$$\leq e^{2\theta_t} \cdot \left(1 + 2p_t(1 - p_t) \cdot \left(\eta\Delta + \eta^2 C_\eta\right)\right) , \quad \text{since } a_t \vee b_t \leq 2\eta .$$

Then, we use the relation $e^{2\theta_t}(1 - p_t) = p_t$ (Eq. (11) in Appendix A) to obtain that

$$\forall t \geq 1, \ \mathbb{E}_t[e^{2\theta_{t+1}}] \leq e^{2\theta_t} + p_t^2 \cdot 2\left(\eta\Delta + \eta^2 C_\eta\right) \implies \mathbb{E}\left[e^{2\theta_{T+1}}\right] \leq 1 + 2\left(\eta\Delta + \eta^2 C_\eta\right) \cdot T,$$

which gives the logarithmic term in the theorem. It remains to upper bound $\mathbb{E}[\sum_{t=1}^{T}(1 - p_t)^2]$. For $t \geq 1$, we define $x_t := \frac{1-p_t}{p_t} = e^{-2\theta_t}$, and obtain with the same arguments as above that

$$\mathbb{E}_t[x_{t+1}] \leq x_t \cdot \left(1 - 2p_t(1 - p_t)\eta\Delta + 2p_t(1 - p_t)\eta^2 C_\eta\right) = x_t - 2\eta(1 - p_t)^2 \cdot \left(\Delta - \eta C_\eta\right) .$$

By taking expectation on both sides and summing over time steps, we thus obtain that

$$\mathbb{E}[x_{T+1} - x_1] \leq -2\eta \cdot (\Delta - \eta C_\eta) \cdot \mathbb{E}\left[\sum_{t=1}^{T}(1 - p_t)^2\right],$$

from which the result follows by using that $\mathbb{E}[x_1 - x_{T+1}] \leq x_1 = 1$ and that $\Delta - \eta C_\eta > 0$. $\quad\square$

**Tightness of the bound** In addition to being strikingly simple, the proof of Thm. 1 is also tight under the assumption that rewards are bounded in $[-1, 1]$. First, for $\nu_1 = \text{Rad}(\Delta)$ and $\nu_2 = \text{Rad}(0)$, and $\eta C_\eta = (1-\varepsilon)\Delta$ for some $\varepsilon > 0$, the logarithmic (post-convergence) term matches the asymptotic lower bound up to a factor $(1-\varepsilon)^{-1}(1+\mathcal{O}(\Delta))$ (see D.1 for details). It thus cannot be much improved if $\varepsilon, \Delta$ are small. For the constant term, the only possible improvement would be to use higher-order moments for a tighter approximation in Eq.(8). However, for Rademacher distributions we could at most replace $C_\eta$ by 1, so the gain would be minor for small $\eta$ since we already have that $C_\eta \leq e^{2\eta}$.

**Comparison with existing results** Theorem 1 is a fully explicit regret upper bound, which surprisingly does not rely on any techniques from standard analyses of gradient ascent policies. For two arms, we obtain guarantees for a much broader range of learning rates[2] than Mei et al. [1], and even obtain near-optimal logarithmic scaling of the regret if $\eta$ happens to be close to $\Delta$. We then compare SGB with SAMBA, another policy performing (non-parametric) gradient ascent. For two arms, SAMBA running with parameter $\alpha < \Delta$ achieves a regret bound of $\alpha^{-1}\log(T)$ [29], which is close to our result. We elaborate on the comparison between the two policies in Appendix B.2. While their analysis extends to $K > 2$, it is more involved than the proof of Thm. 1, is restricted to Bernoulli rewards, and involves non-explicit and potentially large constants. Finally, we highlight that the logarithmic component of Thm. 1 is valid for **all** constant $\eta$. Hence, for SGB restricting $\eta$ is required **only** to guarantee that the failure regret converges, which is a novel insight compared to [29].

**Alternative moment-based condition** A closer examination of Eq. (8) reveals that the analysis of Thm. 1 can be extended or sharpened under general moment assumptions on the reward distribution. For example, if one assumes that $\sup_{m \geq 2} \mathbb{E}[r_t^m] \leq s^2$ for some $s > 0$, then the result continues to hold by scaling the term $\eta C_\eta$ by $s^2$ when bounding the failure regret, so $\eta C_\eta < \frac{\Delta}{s^2}$ is sufficient for logarithmic regret. Hence, SGB can be fine-tuned under higher-moment conditions, akin to how Bernstein-type inequalities [33, Chapter 2.8] improve confidence intervals in UCB-style algorithms. Such assumptions are often realistic in applications like marketing or online advertising, where reward distributions are typically bounded and the average outcomes (e.g. click rates) are small [34, 35]. In Appendix F.5 we also discuss the case of unbounded rewards, e.g. sub-Gaussian. Notably, for SGB these assumptions merely expand the admissible range of learning rates. By contrast, standard approaches would require substantial structural changes, such as tighter confidence intervals for UCB or modified priors/posteriors for TS, to exploit higher-order moment information.

**Knowledge of $\Delta$** While assuming that a lower bound on $\Delta$ is known remains a strong requirement, we conclude this section by showing that, even without this knowledge, a learning rate that depends only on the time horizon $T$ leads to meaningful regret guarantees in the two-armed bandit setting.

**Corollary 1** (of Theorem 1). *If $\Delta$ is unknown, one can set $\eta = \sqrt{\frac{\log(T)}{T}}$, which yields*

$$\forall \nu \in \mathcal{F}^2, \ \mathcal{R}_T \lesssim \frac{\log(T)}{\eta} \cdot \mathbb{1}(2\eta C_\eta \leq \Delta) + \Delta T \cdot \mathbb{1}(\Delta < 2\eta C_\eta) \lesssim \sqrt{T\log(T)},$$

*by using the upper bound from the theorem if $\eta$ is small enough, and $\mathcal{R}_T \leq \Delta T$ otherwise.*

In Proposition 4 (App. F.3) we further prove that a time-varying learning rate $\eta_t = \sqrt{\log(e \vee t)/t}$ also yields $\mathcal{R}_T = \widetilde{\mathcal{O}}(\sqrt{T})$. While Mei et al. [1, Section 3] hint that such a rate may be an appropriate

---

[2]We document in Appendix B.1 a potential issue with their result, and propose a non-trivial fix for $K = 2$.

tuning for SGB, it appears that comparable results to ours have only been been obtained with regularization added to the policy update, as in [20, 21]. On the other hand, the regret upper bound of the gap-free variant of SAMBA [29, Thm.2] still involves large problem-dependent constants, while the constants in Corollary 1 are absolute. Hence, the time-dependent tuning of $\eta$ that we propose stands out as a practical choice when no lower bound on $\Delta$ is available (see Appendix G.1 for experiments). For the two-armed case, we believe that these theoretical results offer a comprehensive guideline on how to choose $\eta$ when using SGB in practice: if the learner has access to a lower bound on $\bar{\Delta} < \Delta$ such that the gap-dependent tuning $\eta \approx \bar{\Delta}$ guarantees that $\frac{\log(1+4\eta\Delta T)}{2\eta} \leq \sqrt{T \log(T)}$ for horizon $T$, then the gap-dependent tuning should yield better performance, since we believe that the bound of the theorem is tight. Otherwise, the horizon-dependent tuning that we propose can be safely used.

## 2.2 Necessary condition on fixed $\eta$ for logarithmic regret

In this section, we prove that the knowledge of the minimum gap $\Delta$ is necessary to tune the fixed learning rate of SGB in order to obtain logarithmic regret. The following Thm. 2 shows that if $\eta$ is larger than a fixed constant that depends on the gap $\Delta$ and the number of arms $K$, there exists a $K$-armed bandit problem with gap $\Delta$ for which SGB has polynomial regret. To define it, for $\mu \in [-1, 1]$ we denote resp. by $\text{Rad}(\mu)$ and $\delta_\mu$ the Rademacher and Dirac distributions of mean $\mu$.

**Theorem 2** (Polynomial regret). *Fix $\Delta \in (0, 1)$, and consider the instance $\nu = (\nu_k)_{k \in [K]}$ with $\nu_1 = Rad(\Delta)$ and $\nu_2 = \cdots = \nu_K = \delta_0$. If the learning rate of SGB satisfies*

$$\eta > \lambda_\Delta := \frac{K-1}{K} \log\left(1 + \frac{2\Delta}{1-\Delta}\right),$$

*then its regret on the instance $\nu$ is lower bounded as follows,*

$$\forall \varepsilon > 0, \qquad \mathcal{R}_T = \widetilde{\Omega}\left(T^{1-(1+\varepsilon)\lambda_\Delta/\eta}\right),$$

*where $\widetilde{\Omega}$ hides polylogarithmic terms in $T$ and constants depending on $\eta$ and $\Delta$.*

*Proof sketch.* Let us assume $K = 2$ in the following, and introduce the main steps of the proof of the theorem. Details and supporting results can be found in Appendix C, where we directly consider a general number of arms. Since only arm 1 yields non-zero rewards, we directly have

$$\forall t \geq 1, \; \theta_{1,t+1} = \theta_{1,t} + \eta(1 - p_{1,t}) \cdot r_t \cdot \mathbb{1}(A_t = 1),$$

so we denote by $\widetilde{p}_{n+1}$ the sampling probability of arm 1 after its $n$ first pulls. The proof consists of identifying a scenario with linear regret, and lower bounding its probability as a function of $\eta$.

Step 1: Let $\mathcal{S}$ be the event that $\widetilde{p}_{n+1} \leq \frac{1}{2T}$, for some fixed value of $n$. Then, under $\mathcal{S}$ the probability that arm 1 is never pulled again after its $n$ first selections is larger than $1/2$, in which case the regret is larger than $\Delta(T - n)$. It thus holds that $\mathcal{R}_T \geq 0.5 \cdot \Delta \cdot \mathbb{P}(\mathcal{S}) \cdot (T - n)_+$.

Step 2: We lower bound $\mathbb{P}(\mathcal{S})$ for a well-chosen value of $n = n_0 + n_1$, for two integers $n_0$ and $n_1$. Consider the following scenario: in a preliminary phase, arm 1 collects only $-1$ rewards from its first $n_0$ pulls, and in a next phase it collects $n_1$ rewards with an empirical mean satisfying $\widehat{\mu}_{n_1} \approx -\Delta$, while the number of $+1$s received is never more than the number of $-1$s throughout this phase. First, we derive in Lemma 6 a lower bound on the probability of this scenario. Then, we prove (Lemma 7) that setting $n_1 = \mathcal{O}(\log(T))$ and $n_0 = \mathcal{O}(\log(n_1))$ guarantees that this scenario is included in the event $\mathcal{S}$. From a high-level perspective, the preliminary phase only serves to make $\widetilde{p}_{n_0+1}$ small enough to reduce the impact of the ordering of the rewards in the next phase, and the scaling of the lower bound (ignoring log terms) comes from the choice of $n_1$.

Lastly, we emphasize that the result should hold with random sub-optimal arms (further assuming that they are "lucky enough" throughout the trajectory), but constant rewards simplify the proof.

$\square$

**Critical regime** We remark that Theorems 1 and 2 do not provide explicit results about the regret of SGB when $\eta \approx \Delta$, that we thus call the *critical regime*. However, by comparing both theorems and their proofs we conjecture that, on some difficult instances, SGB satisfies $\mathbb{E}\left[(1 - p_t)^2\right] \asymp t^{-\frac{\Delta}{\eta}}$.

Indeed, after summation this scaling makes the failure regret admit a $\Omega(T^{1-\frac{\Delta}{\eta}})$ lower bound when $\eta > \Delta$ (see Thm. 2). If there is a smooth interpolation between the logarithmic and polynomial regime, then there might exists a small range of learning rates $\eta \approx \Delta$ for which the **failure regret** could be logarithmic *on difficult instances*. This is supported by experiments from Section 4 and Appendix G: we observe a relatively smooth evolution of performance metrics (regret, percentage of failed runs) with the learning rate. In the experiments, the critical learning rates, such as $\eta = \Delta$ when $K = 2$, provide good empirical performance.

## 3 Results for $K > 2$ arms

In this section, we explore the theoretical guarantees of SGB for $K > 2$. We start by proving that the learning rate must decrease with the number of arms in order to guarantee logarithmic regret. Then, we provide preliminary results for upper bounding the regret of SGB by generalizing the proof of Theorem 1. We also provide some intuition about why this case is significantly harder than $K = 2$, and what we believe to be the right critical scaling for $\eta$.

### 3.1 Necessary scaling of $\eta$ in $K$ for logarithmic regret

In this part, we exhibit a *necessary* condition on the learning rate for logarithmic regret, depending on the number of arms $K$. While we proved Theorem 2 by directly analyzing a difficult instance, on the contrary, the proof technique of Theorem 3 exploits the efficiency of SGB on an *easy* problem instance, from which we deduce the result using lower bound arguments. We start by presenting the regret upper bound leading to this conclusion.

**Lemma 1** (Regret upper bound on an easy instance)**.** *Let $\nu \in \mathcal{F}^K$ be a MAB defied by $\nu_1 = \delta_0$ and $\nu_2 = \cdots = \nu_K = \delta_{-\Delta}$, for some $\Delta > 0$. Then, for **any** learning rate $\eta$, SGB satisfies*

$$\forall \varepsilon \in (0,1), \ \mathcal{R}_T^{SGB} \leq \frac{1 + \log\left(1 + (K-1)T\eta\Delta\right)}{(1-\varepsilon)\eta} + \frac{K^2}{\varepsilon} \cdot \left(\Delta + \frac{1}{\eta}\log\left(\frac{K}{\varepsilon}\right)\right).$$

*Proof sketch.* We present the full proof in Appendix D.2. We first show that, since rewards are deterministic, arm 1 remains the mode of the sampling distribution for all steps $t$, and $p_{k,t}/p_{1,t}$ is non-increasing for any sub-optimal arm $k$. Hence, we establish that $p_{1,t} \geq 1 - \varepsilon$ happens *in finite expected time* for any threshold $\varepsilon \in (0,1)$, which gives the $\mathcal{O}(\varepsilon^{-1})$ term of the bound. We then prove that, from that stage, $1 - p_{1,t}$ decreases exponentially fast with the **total number of sub-optimal plays**. This careful decomposition allows us to obtain a logarithmic term that does not involve a multiplicative factor of $K - 1$. In Appendix D.2 and G.3 we further discuss how this result might be tightened by a factor $\frac{K-1}{K}$, which we prove formally for $K = 2$ (Lemma 14). $\qquad\square$

**Implication on the consistency of** SGB  Lemma 1 suggests that, for a fixed horizon, the regret of SGB can be arbitrarily small for large $\eta$ in some easy bandit instances. As a consequence of the well-known asymptotic regret lower bound [36, 37], this must come at a cost on *other instances*. We dedicate Appendix D.1 for a thorough presentation of the lower bound we use in the proof (restated in Theorem. 6), and other technical results needed to derive the following theorem.

**Theorem 3** (Polynomial regret for $\eta \gtrsim K^{-1}$)**.** *Let $\Delta \in (0,1)$ and $\alpha \in (0,1)$. Consider the class $\mathcal{F}_\Delta^K$ of $K$-armed bandit instances with minimum gap at least $\Delta$. If the learning rate satisfies*

$$\eta > \frac{1}{\Delta(1-\alpha)}\log\left(1 + \frac{2\Delta}{1-\Delta}\right) \cdot \frac{1}{K-1},$$

*then there exists an instance $\nu \in \mathcal{F}_\Delta^K$ such that the regret of SGB satisfies*
$$\mathcal{R}_T = \Omega(T^\alpha).$$

*Proof.* Setting $\varepsilon^{-1} = \sqrt{\log(3+T)}$, Lemma 1 yields $\mathcal{R}_T = \eta^{-1} \cdot \log(T) + \mathcal{O}(\sqrt{\log(T)})$ for the instance $\nu_1 = \{\delta_0, \delta_{-\Delta}, \ldots, \delta_{-\Delta}\} \in \mathcal{F}_\Delta^K$, for any $\eta > 0$. In contrast, Theorem 6 (adaptation of the Lai & Robbins lower bound) and Lemma 11 (specialization to Dirac distributions) imply that any policy with regret $\mathcal{R}_T = \mathcal{O}(T^\alpha)$ on all instances in $\mathcal{F}_\Delta^K$ should satisfy $\liminf_{T\to\infty} \frac{\mathcal{R}_T}{\log(T)} > \frac{(K-1)(1-\alpha)\Delta \log(T)}{\log\left(1+\frac{2\Delta}{1-\Delta}\right)}$ on this deterministic instance $\nu_1$. Combining these two results proves the theorem.
$\qquad\square$

**Discussion** The fact that the learning rate has to be inversely proportional to the number of arms might be surprising at first sight. An intuitive explanation is that the update rule of $\theta_{1,t}$ is agnostic to the number of arms: for any step $t \geq 1$, the distribution of $\delta\theta_{1,t}$ conditioned on $\mathcal{H}_{t-1}$ is the same if arm 1 faces a single arm ($K = 2$) or a mixture of $K - 1$ identical arms. Hence, in order to reduce the speed of convergence of $p_{1,t}$, and thus guarantee sufficient exploration of the $K - 1$ alternatives, it is necessary to make the learning rate decrease with $K$. In contrast, standard bandit policies typically have a separate exploration mechanism for each arm, determined by their respective observations.

**Conjectures** Combining Thm. 2 and 3, we establish that logarithmic regret is only achievable for $\eta \lesssim \Delta \wedge K^{-1}$. This suggests that the critical threshold separating logarithmic from polynomial regret may depend on the ratio $\Delta/K$. We explore this idea further in Appendix D.3, where we present and motivate two conjectures. Conjecture 1, inspired by Theorem 3, posits that regret cannot be too small on an "easy" instance. Conjecture 2 considers a construction with one slightly sub-optimal arm and many very sub-optimal arms: $\nu_1 = \mathrm{Rad}(\Delta)$, $\nu_2 = \delta_0$, and $\nu_3 = \cdots = \nu_K = \delta_{-1}$. Both suggest that the critical rate, above which SGB may suffer polynomial regret, is near $\frac{2\Delta}{K}$ for $K$-armed bandits.

## 3.2 Tools for upper bounding the regret

In this section we propose some tools to derive upper bounds on the regret of SGB for $K > 2$. While our current results do not lead to a complete regret upper bound in general, we introduce promising preliminary results, with full regret bounds in some cases, showing the potential of the techniques introduced in this paper for future investigation on this problem.

Let us introduce $M_T = \sup_{k \in \{2,\ldots,K\}} \sup_{t \in [T]} \mathbb{E}[e^{\theta_{k,t}}]$. Then, our first result of this section states that $\mathbb{E}[\theta_{1,T+1}]$ is **always** logarithmic in $T$ when $K > 2$, as detailed below.

**Lemma 2.** *For any horizon $T \geq 1$ and for **any learning rate** $\eta$, parameter $\theta_{1,T+1}$ of SGB satisfies*

$$\mathbb{E}[\theta_{1,T+1}] \leq \log\left(1 + \left(\eta\Delta + \frac{\eta^2}{2}e^\eta\right)(K-1) \cdot M_T \cdot T\right) \wedge (K-1) \cdot (\log(T) + 4\eta) ,$$

We prove the lemma in Appendix F.2, with distinct proofs for the two bounds. The first one is based on Jensen inequality, similarly to the proof of Thm. 1. For the second bound, we first prove that the expectation of the *minimum* parameter $\mathbb{E}[\min_{k \in [K]} \theta_{k,t}]$ cannot be smaller than $-\log(T)$. We then deduce the bound scaling with $(K-1)\log(T)$ by using the linear relationship (9) between all parameters. The first bound is tighter asymptotically if $M_T = o(T^{K-2})$, which we think should hold at least for small enough $\eta$, but proving it formally requires new techniques. Nonetheless, the potentially sub-optimal $K - 1$ factor only appears in the upper bound, and does not translate into a supplementary constraint on how to tune SGB to achieve logarithmic regret.

We now introduce an upper bound on the failure regret obtained by generalizing the proof of Thm. 1.

**Lemma 3.** *For any instance $\nu \in \mathcal{F}_\Delta^K$ and learning rate satisfying $\eta C_\eta < \Delta$, SGB satisfies*

$$\mathbb{E}\left[\sum_{t=1}^T (1 - p_{1,t})^2\right] \leq \frac{K-1}{K \cdot \eta(\Delta - \eta C_\eta)}\left(K + \mathbb{E}\left[\sum_{t=1}^T \frac{\eta H_t + \frac{\eta^2}{2}W_t}{p_{1,t}}\right]\right) , \text{ with}$$

$H_t = \sum_{k=2}^K p_{k,t}^2 \left(\mathbb{E}_t[\Delta_{A_t}] - (1 - p_{1,t})\Delta_k\right)$ *and* $W_t = \sum_{k=2}^K p_{k,t}^2(1 - p_{k,t}) - p_{1,t}(1 - p_{1,t})^2$.

This result, proved in Appendix F.4, matches the upper bound on the failure regret in Thm. 1 for $K = 2$, since $H_t = W_t = 0$ in this scenario. Unfortunately, upper bounding the additional terms depending on $H_t$ and $W_t$ is intricate in general. However, these terms can be non-positive under favorable scenarios, and are thus not necessarily causing regret. In Appendix F.4.1 we detail some properties and intuition about $H_t$ and $W_t$, that can be useful for future works. Intuitively, $W_t$ reflects how uniformly the probability $1 - p_{1,t}$ is divided among sub-optimal arms, leading to increased variance for parameter updates. Moreover, $H_t > 0$ can occur in a scenario where increasing the probability of an arm whose gap is lower than the expected gap of the policy might be preferable to increasing $p_{1,t}$. Finally, we remark that Lemma 3 does not hint that $\eta$ should depend on $K$, suggesting that the scaling found in Thm. 3 is required to further upper bound the newly introduced terms.

We finally present a complete regret upper bound when the gaps are identical among all arms, that we prove in Appendix E. As expected from previous results, the main difficulty is to upper bound the

failure regret, although we can exploit the fact that $H_t = 0$ in the case of identical gaps. The factor $K$ in the tuning of the learning rate emerges from the initialization $p_{1,1} = K^{-1}$

**Theorem 4** (Logarithmic regret for identical gaps). *Let $\nu \in \mathcal{F}^K$ be a MAB instance satisfying $\Delta_2 = \cdots = \Delta_K = \Delta$, for some $\Delta \in (0,1)$. Then, the regret of SGB tuned with a learning rate $\eta$ satisfying $\eta C_\eta < \frac{2\Delta}{K+2}$ is upper bounded as follows,*

$$\mathcal{R}_T^{SGB}(\nu) \leq \frac{2}{\eta} \cdot \mathbb{E}[\theta_{1,T+1}] + \Gamma_\nu \leq \frac{2(K-1)}{\eta} \cdot \log(T) + \Gamma_\nu \text{ , for some constant } \Gamma_\nu > 0.$$

Finally, we present in Lemma 15 in Appendix E a result analogous to Corollary 1. With a refined analysis of the constant $\Gamma_\nu$, we prove that a tuning $\eta_T = 1/\sqrt{T}$ yields a regret bound $\mathcal{R}_T \lesssim K\sqrt{T}$. Thus, in this setting a horizon-dependent learning rate can also be used when $\Delta$ is unknown.

## 4  Experiments

We now provide empirical support for the theoretical results presented in previous sections, focusing on the performance of SGB as a function of its learning rate $\eta$. We present additional experiments in Appendix G, where we also empirically compare SGB to some standard bandit algorithms.

**Experimental Setups**  To illustrate Thm. 1 and 2 we first consider a two-armed bandit experiment with distributions $\nu_1 = \text{Rad}(0.1)$ and $\nu_2 = \text{Rad}(0)$, on which we test SGB with four different learning rates: $(\eta_i)_{i \in [4]} = \left\{ \frac{\Delta}{2}, \Delta, 2\Delta, 5\Delta \right\}$. Among these, $\eta_1 = \frac{\Delta}{2}$ yields logarithmic regret (Thm. 1), and $\eta_2 = \Delta$ is the critical threshold identified in Section 2. Assuming that Thm. 2 is tight, and using $\lambda_\Delta \approx \Delta$, we expect $\eta_3$ and $\eta_4$ to suffer polynomial regret of resp. order $\sqrt{T}$ and $T^{4/5}$. For the second experiment, we consider $K = 10$ arms and the instance defined by $\nu_1 = \text{Rad}(0.1)$, $\nu_2 = \delta_0$ and $\nu_3 = \cdots = \nu_{10} = \delta_{-1}$, in order to support Thm. 3, and more precisely the conjecture that the critical learning rate is $\eta = 2\Delta/K$ for $K$-armed problems. Thus, we compare the performance of SGB with learning rates $(\eta_i)_{i \in [5]} = \left\{ \frac{\Delta}{K}, \frac{2\Delta}{K} = \frac{\Delta}{5}, \frac{\Delta}{2}, \Delta, 5\Delta \right\}$. For each setup, we run $10^4$ independent trajectories of SGB over horizon $T = 2 \cdot 10^4$. Our results are displayed in Figures 1 and 2 respectively.

**Results**  The results of both experiments support the theoretical findings developed in this work. In each setting, only learning rates below the critical threshold $2\Delta/K$ (specifically, $\eta_1$ and $\eta_2$) lead to a logarithmic average regret. For the rate just above the critical threshold ($\eta_3$), the additional statistics presented alongside the average regret curves highlight the fragility of this tuning. In particular, the average regret shifts closer to the 90th percentile across runs, indicating a noticeable increase in failed trajectories compared to smaller learning rates. For $K = 2$, the table in Fig. 1 confirms this finding.

The figures also remarkably illustrate the regret decomposition of Eq.(7), particularly the right panel of Figure 2, which shows the empirical distributions of regret across trajectories at time $T$. These distributions are distinctly *bimodal*. A subset of runs exhibits a *post-convergence* behavior, with regret concentrated around a value proportional to $\eta^{-1} \log(T)$ (see Eq. (7) and Lemma 2). We highlight this by plotting dotted lines that scale the average regret of SGB tuned with $\eta_1$—used as a reference due to its negligible failure rate—by the factor $\eta_1/\eta_i$ for each $i \in [5]$. These reference lines align well with the "success" mode of their respective learning rates. At the same time, the distributions also reveal a *failure mode*, whose mass substantially increases with the learning rate (30% for $\eta_5$). Across all configurations, only a small fraction of trajectories deviate from these two dominant modes.

**Connections with related works**  Prior works [38, 39] documented multimodal regret distributions for certain UCB-based algorithms, and derived impossibility results for the tail behavior of regret, later strengthened in [40]. Prop. 7 from [40] particularly resonates with our findings on SGB : let $b > 0$, and $\pi$ be a policy with regret $R_T \leq (1 + b)A_\nu \log(T)$ on every instance $\nu \in \mathcal{F}^K$, where $A_\nu \log(T)$ denotes the Lai–Robbins lower bound (Eq. (17), letting $\alpha = 0$). Then, for some $c > 0$, the empirical regret $\bar{R}_T$ satisfies $\mathbb{P}_\pi(\bar{R}_T > cT) \asymp T^{-(1+b)}$ (we refer to the paper for a more formal statement). This closely mirrors our theoretical and empirical observations on SGB, in particular the connection between the learning rate $\eta$ and the failure regret. We conjecture that the critical value of $\eta$ aligns the logarithmic regret term with the best logarithmic bound $A_\nu \log(T)$ in some problem instances, and that the failure regret can be derived from the above polynomial-decay bound, probably

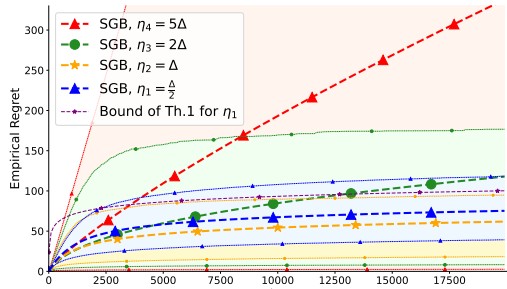

| $T$ \ $\eta$ | $\frac{\Delta}{2}$ | $\Delta$ | $2\Delta$ | $5\Delta$ |
|---|---|---|---|---|
| $5 \cdot 10^3$ | 0.7 | 2.3 | 7.6 | 19.9 |
| $10^4$ | 0.2 | 1 | 5.4 | 17.6 |
| $2 \cdot 10^4$ | 0.06 | 0.5 | 3.6 | 15.5 |

% of the $10^4$ runs with
empirical regret larger than $\frac{T\Delta}{2}$
(regret of the uniform policy)

Figure 1: $K = 2$, average regret and $10 - 90\%$ percentiles (dashed lines) for $10^4$ independent runs up to $T = 2 \cdot 10^4$ (Left), and percentage of trajectories with emp. regret larger than $\frac{T\Delta}{2}$ (Right).

extending to negative $b \in (-1, 0]$. These connections suggest that the aforementioned results could be generalized further, at least for some policy classes such as SGB. Finally, we note that [41] studied conditions under which exponential decay of the regret distribution is possible, but unfortunately one of them is that the problem-dependent regret is polynomial. Strong decay of the distribution of empirical regret is incompatible with logarithmic regret.

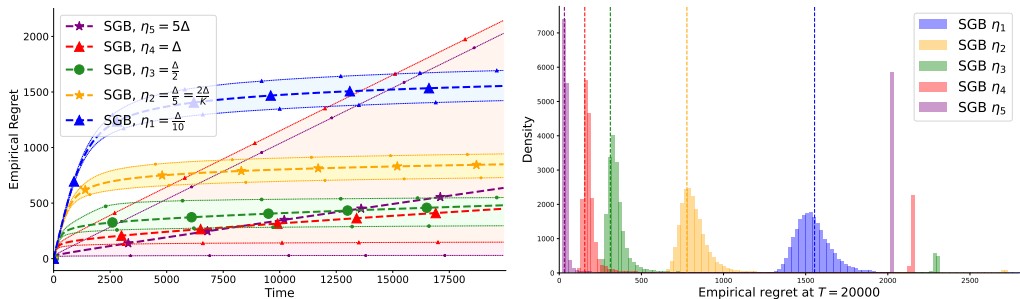

Figure 2: $K = 10$, average regret and $10 - 90\%$ percentiles on $10^4$ independent runs up to horizon $T = 2 \cdot 10^4$ (Left), and distribution of the empirical regret at time $T$ (Right).

## 5 Conclusion

This work sheds light on the theoretical properties of SGB, a simple policy gradient algorithm for bandits, whose behavior is governed by a learning rate $\eta$. In contrast to much of the existing literature, which largely adopts an optimization perspective, we approach the problem through the lens of regret analysis. For two-armed bandits, we show that knowledge of a lower bound on the sub-optimality gap $\Delta$ is both necessary and sufficient to achieve logarithmic regret. For general $K$-armed settings, our results indicate that the critical learning rate threshold scales as $2\Delta/K$, and we provide technical tools and insights that could serve as a foundation for a full regret characterization of SGB in broader contexts. Our theoretical and empirical findings reveal a fundamental trade-off in tuning SGB: while larger learning rates may enhance performance in favorable scenarios, exceeding the critical threshold significantly raises the risk of *failure*—that is, trajectories in which the optimal arm is consistently under-sampled. Further understanding and managing this trade-off is essential for deploying SGB and its variants reliably and effectively, and offers a compelling direction for future research.

## Acknowledgments and Disclosure of Funding

D. Baudry, S. Vary, and P. Rebeschini were funded by UK Research and Innovation (UKRI) under the UK government's Horizon Europe funding guarantee [grant number EP/Y028333/1]. E. Johnson is funded by EPSRC through the Modern Statistics and Statistical Machine Learning (StatML) CDT (grant no. EP/S023151/1). The authors would also like to thank the anonymous reviewers for their insights and suggestions, Tor Lattimore for directing them to relevant papers at an early stage of the project, Julien Aubert for generously sharing his insights and ideas on the topic, and Rémy Degenne and Emilie Kaufmann for a discussion that helped us improve the statement of Theorem 3.

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

# A  Supplementary material for Section 1

## A.1  Pseudo-code of SGB

We detail below the pseudo-code of SGB for a constant learning rate $\eta$. This implementation only needs a slight modification to consider a time-varying learning rate, as discussed in Section 2: an input sequence $(\eta_t)_{t \geq 1}$ should be provided, and $\eta$ should be replaced by $\eta_t$ in the parameter update.

---

**Algorithm 1** Stochastic Gradient Bandit (SGB)

---

**Input:** Learning rate $\eta > 0$

Set $(\theta_{k,1})_{k \in [K]} = (0)_{k \in [K]}$ ;                                  ▷ Uniform initialization

**for** $t \geq 1$ **do**

    For $k \in [K]$: set $p_{k,t} = \frac{e^{\theta_{k,t}}}{\sum_{j=1}^{K} e^{\theta_{j,t}}}$ ;                 ▷ Compute the sampling probabilities

    Pull an arm $A_t \sim p_t := (p_{k,t})_{k \in [K]}$, collect reward $r_t$ ;        ▷ Sample arm, collect reward

    **for** $k \in [K]$ **do**

        Set $\theta_{k,t+1} = \theta_{k,t} + \eta r_t \cdot \{(1 - p_{k,t})\mathbb{1}(A_t = k) - p_{k,t}\mathbb{1}(A_t \neq k)\}$ ;   ▷ Parameter update

---

**Verification that SGB performs stochastic gradient ascent**  For simplicity of notation, consider parameters $\theta = (\theta_k)_{k \in [K]}$, and define $p = (p_k)_{k \in [K]}$ by $\forall k \in [K]$, $p_k = \frac{e^{\theta_k}}{\sum_{j=1}^{K} e^{\theta_j}}$. We further define $S = \sum_{j=1}^{K} e^{\theta_j}$ for simplicity, and the value of the policy $V = \sum_{k=1}^{K} p_k \mu_k$. Then, for any $k \in [K]$, it holds that

$$\frac{\partial V}{\partial \theta_k} = \sum_{j=1}^{K} \frac{\partial p_j}{\partial \theta_k} \mu_j \ .$$

We then obtain that $\frac{\partial p_k}{\partial \theta_k} = \frac{e^{\theta_k} S - e^{2\theta_k}}{S^2} = p_k(1 - p_k)$ and, for $j \neq k$, $\frac{\partial p_j}{\partial \theta_k} = \frac{-e^{\theta_j} e^{\theta_k}}{S^2} = -p_j p_k$. Hence, by summing we obtain that

$$\forall k \in [K], \quad \frac{\partial V}{\partial \theta_k} = p_k(1 - p_k)\mu_k - p_k \sum_{j \neq k} p_j \mu_j.$$

which matches the second term of Equation (5), by adapting the notation. This verifies that the expected parameter update of SGB matches the gradient of the policy, up to multiplication by the learning rate $\eta$.

## A.2  Some elementary properties of SGB

To enhance clarity, we recall that at time step $t \geq 1$, the sampling probabilities of all arms for SGB are given by the softmax transform of some parameters

$$(\theta_{k,t})_{k \in [K]} \in \mathbb{R}^K, \ \forall k \in [K]: \qquad p_{k,t} \propto \exp(\theta_{k,t}) \ ,$$

and that for a learning rate $\eta$, the parameter are updated as follows,

$$\forall t \geq 1: \ \theta_{k,t+1} = \theta_{k,t} + \eta \cdot \delta\theta_{k,t}, \ \text{with} \ \delta\theta_{k,t} := (1 - p_{k,t}) \cdot r_t \cdot \mathbb{1}(A_t = k) - p_{k,t} \cdot r_t \cdot \mathbb{1}(A_t \neq k) \ .$$

We present some properties that are particularly useful for our analysis.

**The parameters are linearly dependent**  From (4), we can verify that

$$\sum_{k=1}^{K} \delta\theta_{k,t} = r_t \cdot \left( (1 - p_{A_t}) - \sum_{j \neq A_t} p_{j,t} \right) = 0,$$

and as a consequence we obtain the following relation

$$\forall t \geq 1, \ \sum_{k=1}^{K} \delta\theta_{k,t} = 0 \quad \Longrightarrow \quad \forall t \geq 1, \ \sum_{k=1}^{K} \theta_{k,t} = 0 \ . \tag{9}$$

This equation shows that the growth of the parameter of the best arm $\theta_{1,t}$ is bound to how much the parameters of the sub-optimal arm can be negative, which is a crucial ingredient in the proof of Lemma 2. Furthermore, for $K = 2$ it holds that $\theta_{1,t} = -\theta_{2,t}$, which simplifies the expression of the sampling probability. For instance, for the sub-optimal arm 2 it holds that

$$\text{If } K = 2: \ \forall t \geq 1, \quad p_{2,t} = \frac{e^{-\theta_{1,t}}}{e^{-\theta_{1,t}} + e^{\theta_{1,t}}} = \frac{e^{-2\theta_{1,t}}}{1 + e^{-2\theta_{1,t}}} \leq e^{-2\theta_{1,t}} \ . \tag{10}$$

In the same setting this relation furthermore leads to

$$\frac{p_{1,t}}{1 - p_{1,t}} = \frac{p_{1,t}}{p_{2,t}} = e^{2\theta_{1,t}} \ . \tag{11}$$

**The variance of updates scales with the expectation**   Again starting from (4), we analyze the squared variation of the parameters. For all arms $k \in [K]$, it holds that

$$(\delta\theta_{k,t})^2 = r_t^2 \cdot \left( (1 - p_{k,t})^2 \mathbb{1}(A_t = k) + p_{k,t}^2 \mathbb{1}(A_t \neq k) \right),$$

using that $\mathbb{1}(A_t = \cdot)^2 = \mathbb{1}(A_t = \cdot)$ and that $\mathbb{1}(A_t = k)\mathbb{1}(A_t \neq k) = 0$. For all $k \in [K]$, we now introduce $s_k^2 = \mathbb{E}[r_t^2 | A_t = k]$, and define $s^2 = \max_{j \in [K]} s_k^2$. We then obtain that

$$\mathbb{E}_t \left[ (\delta\theta_{k,t})^2 \right] \leq s_k^2 \cdot p_{k,t}(1 - p_{k,t})^2 + p_{k,t}^2 \cdot \sum_{j \neq k} s_j^2 \cdot p_{j,t},$$

that we can upper bound as follows,

$$\mathbb{E}_t \left[ (\delta\theta_{k,t})^2 \right] \leq s^2 \cdot \left( p_{k,t}(1 - p_{k,t})^2 + p_{k,t}^2 (1 - p_{k,t}) \right) = s^2 \cdot p_{k,t}(1 - p_{k,t}) \ . \tag{12}$$

Again, this property is essential for our analysis. For instance, it implies that

$$\mathbb{E}_t[\delta\theta_{1,t}] \geq \Delta \cdot p_{1,t}(1 - p_{1,t}) \geq \frac{\Delta}{s^2} \cdot \mathbb{E}_t[(\delta\theta_{1,t})^2] \ .$$

**Iterative formula for the sampling probabilities**   Lastly, inspired by standard proofs in bandits (see Chapter 11 in [42]), we can express the following recursion on the sampling probabilities, which we use in the proof of all the regret upper bounds presented in this paper,

$$\forall k \in [K], \ t \geq 1: \quad p_{k,t+1} = \frac{e^{\theta_{k,t+1}}}{\sum_{j=1}^K e^{\theta_{j,t+1}}} = p_{k,t} \cdot \frac{e^{\eta\delta\theta_{k,t}}}{\sum_{j=1}^K p_{j,t} \cdot e^{\eta\delta\theta_{j,t}}} \ , \tag{13}$$

which is direct using that $\forall j \in [K], \theta_{j,t+1} = \theta_{j,t} + \eta \cdot \delta\theta_{j,t}$, and by dividing both the numerator and denominator by $\sum_{j=1}^K e^{\theta_{j,t}}$.

### A.3   Detailed literature review on Multi-Arm Bandits

The multi-armed bandit (MAB) problem has inspired a vast body of research, with many algorithms proposed to balance exploration and exploitation. In this section, we introduce some of the most standard frameworks developed in the literature. In particular, we focus on algorithms proposed to optimize *problem-dependent* regret bounds in *stochastic bandits*, where rewards are drawn independently at random from fixed distributions, since this is the setting considered in this paper. We refer to [42, 43] for comprehensive surveys and broader discussions, including (but not limited to) adversarial bandits, problem-independent bounds, and Bayesian regret bounds in stochastic bandits.

The classical *optimism in the face of uncertainty* principle underlies the popular UCB family of algorithms. The original UCB1 policy [44, 18] achieves logarithmic regret under bounded or sub-Gaussian rewards. Later, the KL-UCB framework [30] was proposed, and proved to yield asymptotically optimal policies for various families of distribution (e.g. bounded distributions and single-parameter exponential families), in the sense that its regret upper bounds match the asymptotic lower bound of Lai and Robbins [36] (see Appendix D.1 for details). These algorithms rely on carefully constructed confidence bounds, involving the information-theoretic $\mathcal{K}_{\inf}$ divergence (formally defined in Theorem 6), and thus better exploit the statistical properties of distribution classes than simpler UCB variants.

A second major class of policies is *Thompson Sampling* (TS), which dates back to Thompson [31] and has enjoyed renewed popularity since the empirical study of Chapelle and Li [45]. Regarding frequentist analyses, its theoretical foundations were first rigorously established for Bernoulli rewards [19, 46], and further extended to Gaussian [47], single-parametric exponential families [32], and non-parametric bounded distributions [48]. TS is inherently Bayesian, using posterior samples of arm parameters to guide exploration. In all of these settings, TS has been proved to achieve asymptotic optimality under an appropriate choice of prior distribution. For instance, for Bernoulli rewards a uniform prior and Beta posterior yields such guarantees.

A third family, *Minimum Empirical Divergence* (MED) and its deterministic counterpart IMED aim to directly minimize regret by exploiting empirical divergence estimates [49, 50], often inspired by the $\mathcal{K}_{\mathrm{inf}}$ quantity like KL-UCB approaches, but do not rely on confidence bounds in the algorithm designs. Recently, this approach has been rediscovered under the name *SoftElim* [51] or *Maillard Sampling* (MS) [52, 53], and shown to perform optimally under general sub-Gaussian and bounded settings. Interestingly, [54] have analyzed MED through a policy-gradient lens in contextual extensions. A recent work [55] proposes a unified problem-dependent analysis of MED for generic distribution classes, and establishes a strong connection between MED and (a variant of) Thompson Sampling using non-informative prior, showing that the sampling probabilities of sub-optimal arms under the two policies decrease at the same rate asymptotically.

Beyond these three canonical families, a number of works have explored nonparametric randomized strategies. These include bootstrapping-based methods [56, 57], sub-sampling algorithms [58–61], and variants of the nonparametric TS algorithm [48, 62, 63], which connects to the Bayesian bootstrap [63]. While SGB could be perceived as fitting into this line of randomized, data-driven policies, these approaches rely on fundamentally different principles. Indeed, the core ingredients of SGB are gradient ascent and stochastic approximation [3], while these approaches exploit the properties of sample statistics which, in particular, requires storing the entire history of observed rewards.

Hence, it is clear that the SGB policy stands apart from the classical design principles that shape the landscape of bandit algorithms with optimal problem-dependent regret guarantees. In the following we elaborate on why this implies that the analysis of SGB must depart from standard techniques derived in previous bandit analyses. Nonetheless, we then detail how familiarity with these standard approaches still guided our investigation of the theoretical properties of SGB .

**Key differences between SGB and standard policies**   Fix a generic time step $t$. It is clear from (4) that the sampling probabilities of SGB depend in a complex, nonlinear way on the entire sequence of past rewards, across all arms, and the exact order in which they were collected. For instance, consider the ratio $p_{k,t}/p_{j,t}$ for two arms $(k, j) \in [K]^2$. Using (4), we obtain

$$\frac{p_{k,t}^{\mathrm{SGB}}}{p_{j,t}^{\mathrm{SGB}}} = e^{\eta \sum_{s=1}^{t} r_s \cdot \{(1-p_{k,s})\mathbb{1}(A_s=k) - p_{k,s}\mathbb{1}(A_s \neq k) - (1-p_{j,s})\mathbb{1}(A_s=j) + p_{j,s}\mathbb{1}(A_s \neq j)\}} \ .$$

This expression shows that the current sampling ratio cannot be deduced from simple summary statistics such as empirical means or counts. In contrast, standard policies typically admit arm-specific exploration mechanisms, whose analyses rely on the concentration of statistics derived from each arm's individual history.

To illustrate this, consider first the MED algorithm, which, like SGB, is a randomized policy with explicit sampling probabilities. As shown by Baudry et al. [55], MED can be abstractly described using a divergence function $D$ mapping the empirical distributions $(F_{k,t})_{k \in [K]}$ (distibutions of rewards collected by each arm, respectively) and the best empirical mean $\mu^\star(t) = \max_{k \in [K]} \mu_{k,t}$, to positive values. Then, using the sample sizes $(N_k(t))_{k \in [K]}$, the sampling probability ratio is given by

$$\frac{p_{k,t}^{\mathrm{MED}}}{p_{j,t}^{\mathrm{MED}}} = \frac{e^{-N_k(t)D(F_{k,t},\mu^\star(t))}}{e^{-N_j(t)D(F_{j,t},\mu^\star(t))}} \ ,$$

Under mild conditions for divergences $D$, with similar properties to the $\mathcal{K}_{\mathrm{inf}}$ function, this form allows the regret to be decomposed according to events involving $\mu^\star(t)$ and concentration of the empirical distributions, leading to a complete regret analysis [55, Theorem 1].

A similar arm-specific structure underpins Thompson Sampling (TS), perhaps the most well-known class of randomized policies. While TS does not admit a closed-form expression for its sampling

probabilities, its analyses rely on the arm-wise posterior distributions. In particular, modern problem-dependent analyses [19, 46, 32] exploit quantities of the form

$$\mathbb{P}^{\texttt{TS}}(\widetilde{\mu}_{k,t} \geq \mu | \mathcal{H}_{t-1}) = \mathbb{P}^{\texttt{TS}}(\widetilde{\mu}_{k,t} \geq \mu | F_{k,t-1}), \text{ for } \mu \in \mathbb{R}$$

or their lower-tail counterparts, where $\widetilde{\mu}_{k,t}$ is a sample from the posterior distribution of arm $k$ at time $t$. Crucially, these events depend only on the history of arm $k$. This structural decoupling is entirely absent in SGB, which lacks any notion of per-arm sampling mechanism or posterior.

Finally, consider the deterministic UCB family. Given arm-specific confidence indices $\text{UCB}_{k,t}$, the action at time $t$ is chosen as

$$A_t^{\texttt{UCB}} = \text{argmax}_{k \in [K]} \text{UCB}_{k,t} \ ,$$

and the analysis focuses on events such as $\{\text{UCB}_{k,t} \geq \mu_k + c\Delta_k\}$, for some values of $c \in (0, 1)$. As with TS, the confidence bounds are based solely on arm-specific empirical distributions and are thus fully decomposable across arms. In stark contrast, even in highly simplified environments (e.g., when only one arm yields random non-zero rewards), deriving concentration inequalities for the parameters $(\theta_{k,t})_{k \in [K]}$ in SGB is challenging due to their entangled dependence on the global history.

**Comparison of Eq. (7) with similar regret decompositions in the literature**   While we have highlighted the challenges of applying classical bandit analysis techniques to SGB, we now draw parallels between our approach and prior works on randomized policies for MAB.

Several studies have proposed general frameworks for analyzing the regret of randomized algorithms, including Thompson Sampling [42, Thm. 36.2], MED [55, Lemma 3], and bootstrap-based strategies [57, Thm. 1], [62, Thm. 3.1]. Although these analyses differ in presentation, they typically yield a regret decomposition of the following idealized form (using the notation of the present paper):

$$\mathcal{R}_T^\pi(\nu) \leq C_\nu^\pi \log(T) + \sum_{t=1}^{T} \mathbb{P}(p_{1,t} \geq 1 - \varepsilon) \ , \text{ for some constant } C_\nu^\pi \geq 0 \ .$$

These decompositions directly inspired our derivation of Eq.(7) and our interpretation of the two terms in the regret bound. In particular, Riou and Honda [48] introduced and motivated the terminology of post-convergence and pre-convergence terms to describe the first (logarithmic term) and second components, respectively. We adopt a similar language in this work: we prove that the *post-convergence* term in Eq. (7) is logarithmic in $T$ (Lemma 2), and we show in Appendix F.1 that the second term of Eq. (7), that we call *failure regret*, satisfies

$$\mathbb{E}\left[\sum_{t=1}^{T}(1 - p_{1,t})^2\right] \asymp \sum_{t=1}^{T} \mathbb{P}(p_{1,t} \geq 1 - \varepsilon).$$

In regimes where the pre-convergence term dominates the regret, this equivalence shows that the second term of Eq. (7) highlights a failure of the policy to consistently converge to the optimal action at a logarithmic rate. This motivates our use of the term *failure regret* to characterize this term.

### A.4   Detailed discussion on global convergence and finite-time regret

In this section we complete the discussion from Section 1.1 that the global convergence properties of SGB does not preclude poor performance over finite horizons.

**A globally convergent policy with poor regret**   We start by formally proving the statement that asymptotic convergence guarantees do not necessarily imply non-trivial regret guarantees.

**Proposition 1.** *Define an $\varepsilon$-Greedy policy $\pi$ as follows: for $t \leq K$ sample each arm once, and compute their empirical averages. Then, for the rest of time steps $t \geq K + 1$, select an arm uniformly at random with probability $\varepsilon_t = (t \log t)^{-1}$, and otherwise play the arm with best sample average.*

*Then, the policy $\pi$ converges asymptotically to an optimal policy, but its regret satisfies $\mathcal{R}_T \gtrsim \frac{T\Delta}{\log(T)}$ for any $K$-armed bandit problem with Bernoulli distributions of non-zero means.*

*Proof.* The proof below assumes that arm 1 is the only sub-optimal arm, but the arguments easily extend to cases where multiple arms are optimal. We start by proving the convergence claim. Since

$\sum_{t \geq 1}(t \log(t))^{-1} = +\infty$, each arm is guaranteed to be explored infinitely often. Hence, after waiting long enough, the optimal arm is guaranteed to obtain sufficiently enough samples so that its empirical average will remain, for instance, over $\mu_1 - \frac{\Delta}{2}$ for the rest of the trajectory. Similarly, over-estimation of any sub-optimal arm will be corrected, either thanks to the forced exploration scheme or because the arm was played a lot by appearing optimal for some time. This establishes that the policy $\pi$ converges to an optimal arm almost surely.

We now prove the (almost) linear regret lower bound. We prove that $\varepsilon_t$ is decaying fast enough so that there is a non-negligible probability that there is *no exploration at all* during a trajectory of length $T > K$. In particular, this probability is lower bounded by

$$p_{\text{no exploration}} := \prod_{t=K}^{T}\left(1 - \frac{1}{t\log(t)}\right) = e^{\sum_{t=K}^{T}\log\left(1 - \frac{1}{t\log(t)}\right)}$$

$$\geq e^{-\sum_{t=K}^{T}\frac{1}{t\log(t)} - \sum_{t=K}^{T}\frac{1}{2t^2\log(t)^2}\cdot\frac{t\log(t)}{t\log(t)-1}}$$

$$\gtrsim e^{-\sum_{t=K}^{T}\frac{1}{t\log(t)}} \gtrsim \frac{1}{\log(T)} \ .$$

Hence, if this scenario occurs and the best arm collected a reward $0$ for its first pull, while at least one other arm received a $+1$ reward, then a sub-optimal arm is selected for the $T - K$ remaining steps. The probability of this specific event is e.g. lower bounded by $(1 - \mu_1)\max_{k \geq 2}\mu_k > 0$, by assumption that $\max_{k \geq 2}\mu_k > 0$. Hence, in this setting the regret of $\pi$ satisfies $\mathcal{R}_T \gtrsim \frac{T\Delta}{\log(T)}$. $\qquad\square$

**Linear regret on finite horizon**   We now formalize the intuition that, for a fixed horizon $T$, using a learning rate that scales with $\log(T)$ in the softmax transformation may be problematic: a small number of unfavorable rewards can drastically reduce the probability of selecting arm 1 in subsequent rounds. Since $\log(T)$ remains moderate for practical values of $T$, this highlights a potential fragility of SGB when large learning rates are used without additional assumptions on the reward distributions.

**Proposition 2** (Linear regret). *Let $\nu \in \mathcal{F}^K$ defined by $\nu_1 = Rad(1/2)$ and $\nu_2 = \cdots = \nu_K = \delta_0$. Then, for any $\varepsilon \in (0, 1)$, $\eta \geq \log\left(\frac{T-1}{(K-1)(1-\varepsilon)}\right) \implies \mathcal{R}_T^{SGB}(\nu) \geq \frac{\varepsilon(T-1)}{8}$.*

The proof of Proposition 2 follows directly from the more general result presented below, instantiated with $1 - \mu_1 = \frac{1}{2}$, $\Delta_k = \frac{1}{2}$ for all $k \in \{2, \ldots, K\}$. We highlight that the $K - 1$ term in the proposition is permitted, by a slight change in the last steps of the proof of the lemma, that we detail below.

**Lemma 4** (Example 1: optimal Rademacher vs non-negative rewards). *Let $\nu_1$ be a Rademacher distribution of mean $\mu_1 > 0$, and $(\nu_2, \ldots, \nu_k)$ be any distributions supported on $[0, 1]$. Then,*

$$\forall \varepsilon \in (0, 1): \ \eta \geq \log\left(\frac{T-1}{1-\varepsilon}\right) \implies \mathcal{R}_T(\nu) \geq \varepsilon \cdot \frac{1-\mu_1}{2} \cdot \Delta \cdot (T-1) \ .$$

*Proof.* As in the main paper, we use the notation $\Delta := \min_{k \geq 2}\Delta_k$. For this example, since the sub-optimal arms have non-negative rewards, then $\delta\theta_{1,t} \leq 0$ if $A_t \neq 1$. Let $t_0$ denote the first (random) time for which $A_t = 1$ holds. Then, it holds that

$$\mathcal{R}_T \geq \Delta(T-1) \cdot \mathbb{P}(N_1(T) \leq 1)$$
$$\geq \Delta(T-1) \cdot \mathbb{P}(N_1(T) \leq 1 | r_{t_0} = -1) \cdot \mathbb{P}_\nu(r_{t_0} = -1) \ .$$
$$= \Delta(T-1) \cdot \mathbb{P}(N_1(T) \leq 1 | r_{t_0} = -1) \cdot \frac{1-\mu_1}{2}$$
$$\geq \Delta(T-1) \cdot \frac{1-\mu_1}{2} \cdot \left(1 - \mathbb{P}\left(\exists t \in \{2, \ldots, T\} : A_t = 1 | p_{1,t} \leq e^{-\eta}\right)\right) \ ,$$

where the last line comes from a worst case bound for the value of $t_0$ and the fact that after receiving $r_{t_0} = -1$ then

$$p_{1,t_0+1} \leq \frac{e^{-\eta\frac{K-1}{K}}}{e^{-\eta\frac{K-1}{K}} + e^{\frac{\eta}{K}}} \leq e^{-\eta}.$$

Indeed, since all rewards from sub-optimal arms are non-negative, then (1) $\theta_{1,t} \leq 0$ for $t \leq t_0$, and $p_{1,t}$ remains the minimum of the sampling distribution, so $1 - p_{1,t} \geq \frac{K-1}{K}$; and (2) from (9) at least one sub-optimal arm has a non-negative parameter and probability larger than $1/K$, thus the bound on the denominator. We now use the union bound

$$\mathbb{P}\left(\exists t \in \{2, \ldots, T\} : A_t = 1 | p_{1,t} \leq e^{-\eta}\right) \leq (T-1)e^{-\eta},$$

which, for any $\varepsilon \in (0, 1)$, is smaller than $1 - \varepsilon$ if $\eta \geq \log\left(\frac{T-1}{1-\varepsilon}\right)$. Note that for the proof of Proposition 2, since sub-optimal arms does not yield non-zero rewards the sampling probabilities remain uniform until $t_0$, and then it holds that

$$p_{1,t_0+1} = \frac{e^{-\eta\frac{K-1}{K}}}{e^{-\eta\frac{K-1}{K}} + (K-1)e^{\frac{\eta}{K}}} \leq \frac{e^{-\eta}}{K-1},$$

which by substituting in above steps allows to include a $K - 1$ term in the logarithm. $\square$

We now complement Lemma 4 by showing that choosing $\eta = \Omega(\log(KT))$ leads to linear regret for a broad class of reward distributions. This further demonstrates that large learning rates can be harmful in the absence of stronger assumptions on the distributional structure.

**Lemma 5** (Example 2: Non-negative support). *Assume that the rewards are supported on $[0, 1]$. Then if $\eta > \frac{1}{\max_{k \geq 2} \mu_k} \cdot \log((K-1)(T-1))$ then the regret is linear, and more precisely,*

$$\forall \varepsilon \in (0, 1) : \mathcal{R}_T \geq \frac{\varepsilon}{K} \cdot \left( \sum_{k=2}^{K} \mathbb{P}_{r \sim \nu_k}(r \geq \mu_k)\Delta_k \cdot \mathbb{1}\left(\eta \geq \frac{1}{\mu_k} \log\left(\frac{(K-1)(T-1)}{1-\varepsilon}\right)\right)\right) \cdot T,$$

*which holds for $\eta > \log((K-1)(T-1))$ directly if the rewards are Bernoulli.*

*Proof.* For each distribution $\nu_k$ with $k \in [K]$, there exists a constant $q_k > 0$ such that $q_k = \mathbb{P}(r_t \geq \mu_k \mid A_t = k)$. Therefore, at time $t = 1$, each arm $k$ has a probability $\frac{q_k}{K}$ of being selected and yielding a reward $r_1(k) \geq \mu_k$. Conditioned on this event for arm $k$, the probability assigned to arm $k$ at the beginning of the next round satisfies

$$p_{k,2} \geq \frac{e^{\frac{\eta(K-1)\mu_k}{K}}}{(K-1)e^{-\frac{\eta\mu_k}{K}} + e^{\frac{\eta(K-1)\mu_k}{K}}} \implies 1 - p_{k,2} \leq (K-1)e^{-\eta\mu_k},$$

and arm $k$ is played for all remaining rounds with probability larger than $1 - (T-1)(1 - p_{k,2})$, using a union bound argument. In particular, this probability is larger than $\varepsilon > 0$ if $\eta \geq \frac{1}{\mu_k} \cdot \log\left(\frac{(K-1)(T-1)}{1-\varepsilon}\right)$. The result then comes by summing over all sub-optimal arms. Finally, the refinement for Bernoulli distributions is straightforward by replacing $\mu_k$ by 1 in the bound on $p_{k,2}$. $\square$

# B    Detailed comparison with related works

In this appendix, we elaborate on the comparison between the analysis of SGB proposed in this paper and the two most closely related works [1, 29]. Mei et al. [1] introduced the only existing prior regret analysis of SGB but, as we will show, suffers from certain limitations, and Walton and Denisov [29] introduced the SAMBA algorithm, another gradient ascent-based policy, for which they prove logarithmic regret for any number of arms and Bernoulli rewards, under proper gap-dependent tuning of the learning rate. As we will discuss, while SAMBA shares some similarities with SGB , it also exhibits key differences that impact both its theoretical and practical behavior.

## B.1    Comments on the previous analysis of SGB for small learning rates

In this appendix, we compare the results we obtained in this work with the previous guarantees derived by [1], for a general number of arms $K$ but small learning rates. We also document a potential issue in their analysis, which might invalidate the proof of the main regret upper bound [1, Theorem 5.5]. We propose a correction (Theorem 5), that yields a fully explicit logarithmic regret bound for

$K = 2$ (Eq. (15)), under the same restriction on the learning rate as in the original paper. To do that, we used some of the results introduced in this paper, more precisely from the proof of Theorem 1.

Unfortunately, this technique is specific to the case $K = 2$, and we are not aware of a straightforward extension to settings with $K > 2$. In fact, the missing arguments appear closely related to those required to complete the regret analysis in Corollary 3, discussed in Section 3. For this reason, we present Section 3 with the view that a full regret analysis of SGB in the general $K$-armed setting remains an open and challenging problem.

**Comparison of the regret upper bounds**    Adopting the notation of the present paper, we can restate Theorem 5.5 of Mei et al. [1] as follows,

$$\eta \leq \frac{\Delta^2}{40K^{3/2}} \quad \Longrightarrow \quad \forall \nu \in \mathcal{F}^K : \quad \mathcal{R}_T^{\text{SGB}} \leq \frac{2}{\mathbb{E}[c^2]} \cdot \frac{\log(T)}{\eta} + 1 \ , \quad \text{where } c = \inf_{t \geq 1} p_{1,t} \ .$$

Beyond the requirement that the learning rate must scale with $\Delta^2$ (as opposed to $\Delta$), we observe that $c^2 \leq p_{1,1}^2 = \frac{1}{K^2}$, so the regret bound scales at least as $\frac{2K^2 \log(T)}{\eta}$. In contrast, in the symmetric case where $\Delta_2 = \cdots = \Delta_K = \Delta$, Theorem 4 shows that the regret is at most $\frac{2K}{\eta} \log(T)$, for learning rates that may be as large as $\eta \approx \frac{2\Delta}{K+2}$. This improves the bound by a factor at least $\frac{80K^{5/2}}{(K+2)\Delta}$, which is significant in many settings.

Based on the theoretical developments in this work, we further conjecture that the regret of SGB in the general $K$-armed setting should be $\frac{K-1}{K} \cdot \frac{\log(T)}{\eta} + o(\log(T))$, for learning rates that could be only required to satisfy $\eta \leq \frac{2\Delta}{K}$. This suggests that the analysis in [1] may be conservative, potentially leading to a suboptimal characterization of SGB's regret. It remains an open question whether the techniques from this work—closely related to those used in the analysis of stochastic gradient descent in optimization—and the tools introduced in the present paper can be combined to obtain tighter regret bounds in the general $K$-armed case.

**Issue with the proof**    The development in Equation (282) in the proof of [1, Theorem 5.5] assumes that the random variable $c$ is independent of $\delta(\theta_t)$ (using the notation of their paper where $\delta(\theta_t) := (\pi^* - \pi_{\theta_t})^\top r$ denotes the sub-optimality gap at the current state of the policy) which we believe does not hold. Indeed, the variable $c$ satisfies $c = p_{1,t} \wedge \inf_{s \geq 1, s \neq t} p_{1,s}$ for any $t$ including the one that defines $\delta(\theta_t)$, and thus it cannot be independent of $c$. In addition, it is clear that the correlation between $c$ and $\delta(\theta_t)$ is in the wrong direction for Equation (282) to hold.

In order to overcome this issue, we suggest to get back to their Equation (277), and to keep the $t$-dependent quantity $\pi_{\theta_t}(a^\star)$ (or $p_{1,t}$ for us) instead of taking the infimum over all time steps. Then, we can adapt their Equation (281) to obtain, with their notation,

$$\forall t \geq 1, \quad \delta(\theta_t) - \mathbb{E}_t[\delta(\theta_{t+1})] \geq \frac{\eta \cdot \pi_{\theta_t}(a^\star)^2}{2} \cdot \delta(\theta_t)^2 \ .$$

Here, to correctly account for the dependence between $\delta(\theta_t)$ and $\pi_{\theta_t}(a^\star)$, we can use Cauchy-Schwartz inequality to obtain that

$$\mathbb{E}[\delta(\theta_t)^2 \pi_{\theta_t}(a^\star)^2] \geq \frac{\mathbb{E}[\delta(\theta_t)]^2}{\mathbb{E}\left[\frac{1}{\pi_{\theta_t}(a^\star)^2}\right]} \ .$$

As a remark, we think that taking the supremum outside the expectation makes further analysis of this term easier. From that step, we can directly follow the proof steps of Mei et al. [1] to adapt their result and obtain the following correction.

**Theorem 5** (Corrected version of Theorem 5.5 from [1]). *For any instance $\nu \in \mathcal{F}^K$, SGB tuned with learning rate $\eta$ admits the following regret bound,*

$$\eta \leq \frac{\Delta^2}{40K^{3/2}} \quad \Longrightarrow \quad \mathcal{R}_T^{SGB} \leq \sup_{t \in [T]} \mathbb{E}\left[\frac{1}{\pi_{\theta_t}(a^\star)^2}\right] \cdot \frac{2\log(T)}{\eta} + 1 \ . \tag{14}$$

We recall that the original proof steps could have led to a factor $\frac{1}{\inf_{t \in [T]} \mathbb{E}[\pi_{\theta_t}(a^\star)^2]}$, which the authors rightfully treated as a *positive problem-dependent constant*. Indeed, since there exist some favorable

scenarios for which $\pi_{\theta_t}(a^\star)$ rapidly starts converging to 1, it is clear that on can assume that there exists a finite constant $\gamma > 0$ (depending on $\nu$ and $\eta$) such that $\inf_{t \in [T]} \mathbb{E}[\pi_{\theta_t}(a^\star)^2] > \gamma$.

Unfortunately, after correction it is **non-trivial to deduce that the regret is logarithmic**: proving that $\mathbb{E}\left[\frac{1}{\pi_{\theta_t}(a^\star)^2}\right] < C$ for some constant $C > 0$ requires a much more careful analysis, since it becomes necessary to consider *extreme events* that could make the ratio large. For instance, if $\pi_{\theta_T}(a^\star) = \mathcal{O}(T^{-1})$ with probability $\Omega(T^{-1})$ then the regret bound becomes linear in $T$. This scenario needs to be carefully considered, since it is typical for optimized bandit policies, like Thompson Sampling, as documented by [40].

For the case $K = 2$, however, we can apply the techniques from the proof of Theorem 1 to show that this upper bound is non-vacuous. Indeed, using our notation (and dropping the 1 subscript), we obtain by expanding the fraction that

$$\mathbb{E}\left[\frac{1}{p_t^2}\right] = \mathbb{E}\left[\left(\frac{1 - p_t}{p_t}\right)^2 + \frac{2}{p_t} - 1\right] ,$$

and from the second to last equation in the proof of Thm. 1, we know that for $\eta \leq \Delta e^{-2\Delta}$ the expectation $\mathbb{E}[2/p_t]$ initialized at $p_1 = 1/2$ is decreasing and thus

$$\mathbb{E}\left[\frac{1}{p_t^2}\right] = \mathbb{E}\left[\left(\frac{1 - p_t}{p_t}\right)^2 + \frac{2}{p_t} - 1\right] \leq \mathbb{E}\left[\left(\frac{1 - p_t}{p_t}\right)^2\right] + 3$$

Using the notation $x_t^2 := \left(\frac{1 - p_t}{p_t}\right)^2$ used in the proof of Thm 1, we have $x_t^2 = e^{-4\theta_t}$ by Eq. (11) in Appendix A. The exponential form allows to derive an equivalent bound on the expectation of $x_t^2$ as done for the expectation of $x_t$ in the proof of Thm. 1 but with $2\eta$ instead of $\eta$. Consequently, we have that $x_t^2$ is a supermartingagele for $2\eta C_{2\eta} \leq \Delta$ and with $x_1 = 1/2$ its expectation is bounded as

$$\text{for } K = 2, \quad 2\eta C_{2\eta} \leq \Delta \implies \sup_{t \geq 1} \mathbb{E}\left[\left(\frac{1 - p_t}{p_t}\right)^2\right] \leq \frac{(1 - p_1)^2}{p_1^2} = 1 ,$$

which yields that $\mathbb{E}[1/p_t^2] \leq 4$.

This condition on $\eta$ required for $\mathbb{E}[1/p_t^2] \leq 4$ is weaker than the one $\eta \leq \Delta^2/(40K^{3/2})$ needed for the rest of the convergence proof of Theorem 5. Hence, for $K = 2$, Theorem 5 yields the upper bound on the regret as

$$\eta \leq \frac{\Delta^2}{40K^{3/2}} \implies \forall \nu \in \mathcal{F}^2 : \quad \mathcal{R}_T^{\text{SGB}} \leq \frac{8 \log(T)}{\eta} + 1 . \tag{15}$$

We can comment this result is strictly weaker than the one we establish in Theorem 1, which yields the following bound for the same range of learning rates,

$$\mathcal{R}_T^{\text{SGB}} \leq \frac{\log\left(1 + \frac{\Delta^3}{25}T\right) + 1}{2\eta} ,$$

where in the logarithm we used that $\eta \leq \frac{\Delta^2}{40 \cdot 2^{3/2}} \leq \frac{\Delta^2}{100}$. Notably, our result allows for learning rates that are approximately a factor $100/\Delta$ larger than the ones permitted by Thm. 5, while still maintaining a tighter regret guarantee (for a fixed learning rate).

Unfortunately, for the general $K$-arms case fixing the proof of Thm. 5.5 from [1] by bounding $\mathbb{E}[1/p_{1,t}^2]$ appears difficult, potentially requiring to derive the same tools needed to complete an independent regret analysis from the upper bounds presented in Section 3.

An interestingly comparison between the two regret bounds arises by noticing that Thm. 5 yields an upper bound that *multiplies* a (logarithmic) term, resembling the post-convergence component of Eq. (7), by a term related to the failure regret (proportional to $\mathbb{E}[1/p_t^2]$). On the other hand our regret decomposition in (7) is the result of *adding* the two components.

**Comment on the small learning rate** Lastly, we investigated whether the small learning rate requirement in [1] is a byproduct of some looseness in some technical results or a more intrinsic limitation of the policy. We found that the restriction arises from the foundational technical results used to derive the analysis. Consider the two-armed case. In this paper, we showed that for small $\eta, \Delta$, ignoring for convenience the term $C_\eta$ in the following lines, it holds that

$$\mathbb{E}_t[1 - p_{t+1}] \approx \mathbb{E}_t[e^{-2\theta_{t+1}}] = e^{-2\theta_t}\mathbb{E}_t[e^{-2\eta\delta\theta_t}] \approx (1 - p_t) - 2\eta(\Delta - \eta) \cdot p_t(1 - p_t)^2,$$

using that $e^x \approx 1 + x + x^2/2$ and the expressions for $\delta\theta_t$ and $(\delta\theta_t)^2$ from Appendix A.2. This indicates that the policy improves in expectation between step $t$ and step $t + 1$ when $\eta < \Delta$ holds, for any value of $p_t$, but with an amplitude that depends on the current value of $p_t$.

In contrast, the analysis in [1] is based on the equation stated above their Proposition 3.1, which follows from the $\frac{5}{2}$-smoothness of the current value of the policy under the softmax parametrization [13, Lemma 2]. Specializing this equation to the case of $K = 2$ arms yields the following bound in expectation:

$$\mathbb{E}_t[p_t - p_{t+1}] \cdot \Delta \leq -2\eta \cdot p_t^2(1 - p_t)^2\Delta^2 + \frac{5}{2} \cdot \eta^2 \cdot p_t(1 - p_t),$$

where we used Eq. (12) with $s \leq 1$ to bound the term $\mathbb{E}_t[\|\theta_{t+1} - \theta_t\|_2^2]$. The correctness of the scaling of all terms in the inequality above can be confirmed by comparison with Lemma 4.3 in [1].

However, this bound is difficult to interpret, especially when $p_t(1 - p_t)$ is small, due to the mismatch in the dependence on $p_t$ between the progress (negative) and noise (positive) terms. Even assuming that both terms had the same dependence on some common factor $A_t$, we would still obtain that the policy's value increases in expectation only if $\eta \lesssim \Delta^2$. This indicates that the overly conservative scaling of $\eta$ in terms of $\Delta$ arises from looseness in this preliminary bound, rather than from any fundamental limitation of SGB itself: the $\frac{5}{2}$-smoothness property of the softmax-parametrized policy does not appear to accurately capture the dynamics of the SGB update.

## B.2 Detailed comparison between SGB and SAMBA

In SAMBA the gradient ascent is directly performed on the sampling probabilities, with no use of parametrization. Starting from the uniform probability, at each time step $t > 1$ SAMBA performs the following steps:

1. Define $a_t^\star = \mathrm{argmax}_{a \in [K]} \ p_t(a)$, draw $a_t \sim p_t$, collect $R_t$.
2. If $a_t = a_t^\star$, then for $a' \neq a_t^\star$: $p_{t+1}(a') = p_t(a') - \alpha p_t(a')^2 \cdot \frac{R_t}{p_t(a_t^\star)}$
3. Else, $p_{t+1}(a_t) = p_t(a_t) + \alpha p_t(a_t)^2 \cdot \frac{R_t}{p_t(a_t)}$.
4. Define $p_{t+1}(a_t^\star) = 1 - \sum_{a \neq a_t^\star} p_{t+1}(a)$

It is easy to verify that if $\alpha \leq 1$, the update remains within the probability simplex. Interestingly, under this scheme a probability $p_t(a)$ is updated only if it was pulled or if the arm with the highest current probability was pulled. From a gradient ascent perspective, this is a natural approach: the arm with the highest probability serves as the current best guess for the optimal arm, making it reasonable to form importance-weighted estimates of the gap $\Delta_a$ relative to this reference. However, this feature introduces a structural asymmetry, between the *leading* arm and the others, that complicates direct comparisons between SGB and SAMBA when $K > 2$.

For $K = 2$, we can compare the two methods more precisely. Consider the asymptotic regime in which arm 1 has the highest selection probability. In this setting, SAMBA updates the probability of arm 2 according to

$$\mathbb{E}_t^{\mathrm{SAMBA}}[p_{2,t+1}] = p_{2,t} - \alpha\Delta \cdot p_{2,t}^2 .$$

Meanwhile, under SGB we have

$$\begin{aligned}
\mathbb{E}_t^{\mathrm{SGB}}[p_{2,t+1}] &\approx p_{2,t} \cdot \mathbb{E}_t[e^{-2\delta\theta_t}] \\
&\approx p_{2,t} \cdot (1 - 2p_{2,t}(1 - p_{2,t})\eta(\Delta - \eta C_\eta)) \\
&= p_{2,t} - 2\eta(\Delta - \eta C_\eta) \cdot p_{2,t}^2(1 - p_{2,t}) ,
\end{aligned}$$

using the same steps as in the second to last equation in the proof of Thm. 1. Hence, in the asymptotic regime where $p_{2,t} \ll 1$, the expected update of SGB and SAMBA become approximately equivalent if the learning rates are related by $\alpha\Delta = 2\eta(\Delta - \eta C_\eta)$.

We can now also compare the behavior of the two policies in the transient regime where $p_{2,t} \geq p_{1,t}$. In that case, the analysis of both policies is based on properties of $p_{1,t}^{-1}$. It is therefore natural to use this quantity as a basis for comparison in this regime. For SGB , we showed that

$$\mathbb{E}_t^{\mathsf{SGB}}\left[\frac{1}{p_{1,t+1}}\right] \leq \frac{1}{p_{1,t}} - 2\eta(\Delta - \eta C_\eta)(1 - p_{1,t})^2 \,,$$

where the inequality is nearly tight when $\eta$ is small. For SAMBA, rather than relying on the computations of [29] which are specific to Bernoulli distributions, we propose a similar approximation to the one used for SGB, for rewards supported on $[-1, 1]$. Assuming $p_{2,t} > p_{1,t}$ and small $\alpha$, we obtain:

$$
\begin{aligned}
\mathbb{E}_t^{\mathsf{SAMBA}}\left[\frac{1}{p_{1,t+1}}\right] &= \frac{1}{p_{1,t}} \cdot \mathbb{E}_t\left[\frac{p_{1,t}}{1 + \alpha r_{1,t}} + \frac{p_{2,t}}{1 - \alpha\frac{p_{1,t}}{p_{2,t}}r_{2,t}}\right] \\
&\approx \frac{1}{p_{1,t}} \cdot \left(p_{1,t}(1 - \alpha\mu_1 + \alpha^2\mathbb{E}[r_{1,t}^2]) + p_{2,t} + \alpha p_{1,t}\mu_2 + \alpha^2\frac{p_{1,t}^2}{p_{2,t}}\mathbb{E}[r_{2,t}^2]\right) \\
&= \frac{1}{p_{1,t}} - \alpha\Delta + \alpha^2\left(\mathbb{E}[r_{1,t}^2] + \frac{p_{1,t}}{p_{2,t}}\mathbb{E}[r_{2,t}^2]\right) \\
&\leq \frac{1}{p_{1,t}} - \alpha(\Delta - 2\alpha) \,.
\end{aligned}
$$

This yields the sufficient condition $\alpha < \frac{\Delta}{2}$ to ensure that $p_{1,t}^{-1}$ is a super-martingale when $p_{2,t} > p_{1,t}$, which is a stricter requirement than the one from [29] for Bernoulli rewards. Note the above computations can be made exact by using a constant $D_\alpha$ analogous to $C_\eta$ used for second-order approximation of the exponential in the analysis of SGB .

The main insight from this analysis, however, is that the term $(1 - p_{1,t})^2$, which appears naturally in the update dynamics of SGB , does not appear in the approximation for SAMBA. Consequently, the elegant connection we established between $\mathbb{E}[p_0^{-1} - p_T^{-1}]$ and the regret for SGB does not seem to hold for SAMBA: the two policies seem to admit different pre-convergence behavior, even for $K = 2$.

We can finally remark that, although limited to Bernoulli distributions, the analysis of [29] yields logarithmic regret for SAMBA under the condition $\alpha < \Delta$. Notably, this requirement **does not depend on** $K$. We leave for future work the investigation of whether it is preferable—both theoretically and empirically—to maintain the update rule of SGB, where the parameters of all arms are updated at each step, or to adopt a more selective approach as in SAMBA, where non-leading arms cannot influence their respective updates. In particular, the intuitive explanation provided in Appendix D.3 regarding the necessary dependence of $\eta$ on $K$ for SGB may no longer apply if updates are implemented through a mechanism akin to that of SAMBA, where a leading arm is treated differently as the others.

# C Proof of Theorem 2: polynomial regret when $\eta \gtrsim \Delta$

**Roadmap** Below, we present the proof of Theorem 2, which re-uses the proof sketch of Section 2, but adding precise references to the technical lemmas needed for the proof. These results, Lemma 6 and Lemma 7, are presented in the following sections of this appendix. Furthermore, the proof of Lemma 7 is quite technical and itself supported by several intermediate results, that we detail.

We start by restating the theorem.

**Theorem 2** (Polynomial regret). *Fix* $\Delta \in (0, 1)$*, and consider the instance* $\nu = (\nu_k)_{k\in[K]}$ *with* $\nu_1 = Rad(\Delta)$ *and* $\nu_2 = \cdots = \nu_K = \delta_0$*. If the learning rate of SGB satisfies*

$$\eta > \lambda_\Delta \coloneqq \frac{K-1}{K}\log\left(1 + \frac{2\Delta}{1-\Delta}\right),$$

*then its regret on the instance* $\nu$ *is lower bounded as follows,*

$$\forall\varepsilon > 0, \qquad \mathcal{R}_T = \widetilde{\Omega}\left(T^{1-(1+\varepsilon)\lambda_\Delta/\eta}\right),$$

*where* $\widetilde{\Omega}$ *hides polylogarithmic terms in* $T$ *and constants depending on* $\eta$ *and* $\Delta$*.*

*Proof.* We follow the sketch of proof presented in Section 2, adding the detailed computations and references to the supporting results.

Step 1: let $\widetilde{p}_{n+1}$ be the the sampling probability of arm 1 after its $n$ first pulls, and $\mathcal{S}$ be the event that $\widetilde{p}_{n+1} \leq \frac{1}{2T}$ for some value of $n$ that we will fix later. Then, under $\mathcal{S}$ the probability that arm 1 is never pulled again after its $n$ first selections is larger than $\left(1 - \frac{1}{2T}\right)^T \geq \frac{1}{2}$, in which case the regret is larger than $T - n$. We hence first obtain that

$$R_T(\nu) \geq \Delta \cdot \mathbb{P}(\mathcal{S}) \cdot \frac{(T - n)_+}{2} .$$

Step 2: We lower bound $\mathbb{P}(\mathcal{S})$ for a well-chosen value of $n$, by identifying a scenario that leads (deterministically) to $\widetilde{p}_{n+1} \leq \frac{1}{2T}$. Let $n_0, n_1$ be integers to be specified later, we consider the following repartition of $n$ pulls in two phases,

- **Phase 0 - very unlucky start:** for its first $n_0$ pulls, arm 1 collects only $-1$ rewards. We denote this event by $\mathcal{S}_0$.

- **Phase 1 - failed recovery:** the following $n_1$ pulls of arm 1 have an empirical mean satisfying $\widehat{\mu}_{n_1} \lesssim -\Delta$ and the number of $+1$s received is never more than the number of $-1$s throughout this phase. We denote this event by $\mathcal{S}_1$.

Since $\nu_1$ is Rademacher of mean $\Delta$, it is direct that $\mathbb{P}(\mathcal{S}_0) = \left[\frac{1-\Delta}{2}\right]^{n_0}$. For phase 1, let $(\widetilde{R}_t)_{t \geq 1}$ denote the stream of rewards collected by arm 1 (in the order they are received), and $\widehat{\mu}_{n_1} = \frac{1}{n_1} \sum_{t=n_0+1}^{n_0+n_1} \widetilde{R}_t$ be the empirical mean of the rewards during phase 1. Let $S_{n_1} = \sum_{t=n_0+1}^{n_0+n_1} \mathbb{1}\{\widetilde{R}_t = 1\}$ be the number of ones received during this phase. We first define

$$\alpha := \frac{2}{\Delta}\left(\frac{1}{2} - \frac{\left\lceil \frac{1-\Delta}{2} \cdot n_1 \right\rceil}{n_1}\right) \in \left[1 - \frac{2}{\Delta n_1}, 1\right],$$

so that $\frac{1-\alpha\Delta}{2}$ is an integer greater than $\frac{1-\Delta}{2} \cdot n_1$ with $\alpha$ that is as close to 1 as possible.

From Lemma 6, we first obtain that

$$\mathbb{P}\left(\widehat{\mu}_{n_1} = -\alpha\Delta\right) = \mathbb{P}\left(S_{n_1} = \frac{1 - \alpha\Delta}{2} \cdot n_1\right) \geq e^{-1/6}\sqrt{\frac{2}{n_1\pi}} \exp\left(-n_1 \cdot \Delta \log\left(1 + \frac{2\Delta}{1-\Delta}\right)\right).$$

Then, conditioned on this event, Bertrand's Ballot theorem [64, 65] provides that the running number of $-1$ always exceeds the running number of $+1$ with probability

$$\frac{(n_1 - S_{n_1}) - S_{n_1}}{n_1} = \frac{n_1 - 2S_{n_1}}{n_1} = -\widehat{\mu}_{n_1} = \alpha\Delta,$$

which finally gives that the probability of $\mathcal{S}_1$ is lower bounded by

$$\mathbb{P}(\mathcal{S}_1) \geq \alpha\Delta \cdot e^{-1/6}\sqrt{\frac{1}{2n_1\pi}} \exp\left(-n_1 \cdot \Delta \log\left(1 + \frac{2\Delta}{1-\Delta}\right)\right) .$$

Then, it remains to show that $\mathcal{S}_0 \cap \mathcal{S}_1 \subset \mathcal{S}$ for some values of $n_0$ and $n_1$. By Lemma 7, this is guaranteed for $n = n_0 + n_1$ by fixing

$$n_0 = \left\lceil \frac{1}{\eta} \cdot \frac{K}{K-1} \cdot \log\left(\frac{1}{\varepsilon \cdot (K-1)}\right) \right\rceil \vee \left\lceil \frac{1}{\eta}\left(\frac{K}{K-1}\right)^2 \log\left(\frac{K^3 n_1}{(K-1)^2}\right) \right\rceil,$$

$$\text{and} \quad n_1 = \left\lceil \frac{K-1}{K} \cdot \frac{\log(2T/(K-1))}{\eta\Delta(1-\varepsilon)} + \frac{2}{\Delta} \right\rceil.$$

By independence of the two sequences of rewards involved, it holds that $\mathbb{P}(\mathcal{S}) \geq \mathbb{P}(\mathcal{S}_0) \cdot \mathbb{P}(\mathcal{S}_1)$. To obtain the scaling of the result, we combine the above results and use that $n_0$ is tuned as a function of $n_1$. Hence, we can state that there exists a function $C_{n_1}$, with (inverse) polynomial scaling in $n_1$, such that

$$\mathbb{P}(\mathcal{S}) \geq C_{n_1} \cdot e^{-n_1 \cdot \Delta \log\left(1 + \frac{2\Delta}{1-\Delta}\right)},$$

so that replacing $n_1$ by its value and ignoring the $C_{n_1}$ term, which we hide with the $\gtrsim$ sign, leads to

$$\mathbb{P}(\mathcal{S}) \gtrsim e^{-n_1 \cdot \Delta \log\left(1 + \frac{2\Delta}{1-\Delta}\right)}$$

$$\geq e^{-\left(1 + \frac{2}{\Delta} + \frac{K-1}{K} \frac{1+\varepsilon}{\eta\Delta} \log \frac{2T}{K-1}\right) \cdot \Delta \log\left(1 + \frac{2\Delta}{1-\Delta}\right)}$$

$$= e^{-(\Delta+2)\log\left(1 + \frac{2\Delta}{1-\Delta}\right)} \cdot e^{-\frac{K-1}{K} \frac{1+\varepsilon}{\eta} \log\left(1 + \frac{2\Delta}{1-\Delta}\right) \cdot \log \frac{2T}{K-1}}$$

$$= e^{-(\Delta+2)\log\left(1 + \frac{2\Delta}{1-\Delta}\right)} \cdot \left(\frac{2}{K-1}\right)^{-\frac{K-1}{K} \frac{1+\varepsilon}{\eta} \log\left(1 + \frac{2\Delta}{1-\Delta}\right)} \cdot T^{-\frac{K-1}{K} \frac{1+\varepsilon}{\eta} \log\left(1 + \frac{2\Delta}{1-\Delta}\right)} .$$

For completeness, we detail the poly-logarithmic terms coming from the expression of $C_{n_1}$ below,

$$C_{n_1} := \left(\frac{1-\Delta}{2}\right)^{n_0} \cdot \alpha\Delta \cdot e^{-1/6} \sqrt{\frac{2}{n_1\pi}}$$

$$\geq \alpha\Delta \cdot e^{-1/6} \cdot \sqrt{\frac{2}{n_1\pi}} \cdot \left(\frac{1-\Delta}{2}\right) \cdot \left(\frac{K^3 n_1}{(K-1)^2}\right)^{\frac{1}{\eta}\left(\frac{K}{K-1}\right)^2 \log\left(\frac{1-\Delta}{2}\right)}$$

$$= \alpha\Delta \cdot \left(\frac{1-\Delta}{2}\right) \cdot e^{-1/6} \cdot \sqrt{\frac{2}{\pi}} \cdot \left(\frac{K^3}{(K-1)^2}\right)^{\frac{1}{\eta}\left(\frac{K}{K-1}\right)^2 \log\left(\frac{1-\Delta}{2}\right)} \cdot n_1^{-\frac{1}{2} + \frac{1}{\eta}\left(\frac{K}{K-1}\right)^2 \log\left(\frac{1-\Delta}{2}\right)} .$$

We can further detail $C_{n_1}$ by replacing $n_1$ by its value. For simplicity, we capture all the problem-dependent factors in a new constant $c_{\eta,\Delta}$, and finally obtain that

$$\exists c_{\eta,\Delta} > 0 : \quad C_{n_1} \geq c_{\eta,\Delta} \cdot (\log(T))^{-\frac{1}{2} + \frac{1}{\eta}\left(\frac{K}{K-1}\right)^2 \log\left(\frac{1-\Delta}{2}\right)} .$$

Step 3: Combining the results from the first two steps, we obtain that

$$R_T(\nu) \geq \Delta \cdot \mathbb{P}(\mathcal{S}) \cdot \frac{(T - n_0 - n_1)_+}{2} \gtrsim T^{1 - \frac{K-1}{K} \frac{1+\varepsilon}{\eta} \log\left(1 + \frac{2\Delta}{1-\Delta}\right)},$$

where the omitted constants depending on $\eta$ and $\Delta$ and polylog factors in $T$ can be recovered from the above lower bound on $\mathbb{P}(\mathcal{S})$. $\qquad\square$

## C.1 Anti-concentration of the empirical mean of Rademacher variables

**Lemma 6.** *Let $n$ be a sample size and $S_n$ be the number of ones received from $n$ independent pulls of a Rademacher variable of mean $\Delta \geq \frac{2}{n}$. Fix $\alpha = \frac{2}{\Delta}\left(\frac{1}{2} - \frac{1}{n} \cdot \left\lceil \frac{1-\Delta}{2} \cdot n \right\rceil\right) \in \left[1 - \frac{2}{\Delta n}, 1\right]$, then*

$$\mathbb{P}\left(S_n = \frac{1 - \alpha\Delta}{2} \cdot n\right) \geq e^{-1/6} \sqrt{\frac{2}{n\pi}} \exp\left(-n \cdot \Delta \log\left(1 + \frac{2\Delta}{1-\Delta}\right)\right).$$

*Proof.* Using the notation $p := \frac{1+\Delta}{2}$, we use the following standard formula,

$$\mathbb{P}(S_n = s) = \binom{n}{s} \cdot p^s (1-p)^{n-s}$$

$$= \binom{n}{s} e^{s \log(p) + (n-s)\log(1-p)}$$

$$= \binom{n}{s} e^{n\left(\frac{s}{n} \log\left(\frac{p}{s/n}\right) + \frac{n-s}{n} \log\left(\frac{1-p}{(n-s)/n}\right)\right) - nH\left(\frac{s}{n}\right)}$$

$$= \binom{n}{s} e^{-nH\left(\frac{s}{n}\right)} \cdot e^{-n\mathrm{kl}\left(\frac{s}{n}, p\right)},$$

where $H(x) = -x \log x - (1-x)\log(1-x)$ is the Shannon entropy, and $\mathrm{kl}$ the Bernoulli KL-divergence. With $\frac{s}{n} = \frac{1-\alpha\Delta}{2}$ and $\alpha \leq 1$, we have

$$\mathrm{kl}\left(\frac{s}{n}, p\right) \leq \mathrm{kl}\left(\frac{1-\Delta}{2}, \frac{1+\Delta}{2}\right) = \frac{1-\Delta}{2} \log \frac{1-\Delta}{1+\Delta} + \frac{1+\Delta}{2} \log \frac{1+\Delta}{1-\Delta}$$

$$= \Delta \log\left(\frac{1+\Delta}{1-\Delta}\right) = \Delta \log\left(1 + \frac{2\Delta}{1-\Delta}\right).$$

Sterling's approximation for the factorial [66] gives for any $n \geq 1$

$$\sqrt{2\pi n}\left(\frac{n}{e}\right)^n < n! < \sqrt{2\pi n}\left(\frac{n}{e}\right)^n e^{1/12}.$$

Thus, for $c \in (0,1)$ such that $cn$ is integer, we obtain that

$$\binom{n}{cn} = \frac{n!}{(cn)!((1-c)n)!}$$

$$\geq \frac{\sqrt{2\pi n} \cdot n^n e^{-n}}{\left(e^{1/12}\sqrt{2\pi cn} \cdot c^{cn} n^{cn} e^{-cn}\right) \cdot \left(e^{1/12}\sqrt{2\pi(1-c)n} \cdot (1-c)^{(1-c)n} n^{(1-c)n} e^{-(1-c)n}\right)}$$

$$= \frac{e^{-1/6}}{\sqrt{2\pi c(1-c)n}} \cdot \left(\frac{n}{e}\right)^{n-cn-(1-c)n} \cdot \frac{1}{c^{cn} \cdot (1-c)^{(1-c)n}}$$

$$= \frac{e^{-1/6}}{\sqrt{2\pi c(1-c)n}} \cdot e^{nH(c)} .$$

Remarking that $\sqrt{2c(1-c)} \leq 1/\sqrt{2}$ for any $c \in (0,1)$, we finally obtain with $cn = s$:

$$\mathbb{P}(S_n = s) \geq e^{-1/6}\sqrt{\frac{2}{n\pi}} \cdot e^{-n\mathrm{kl}\left(\frac{1-\alpha\Delta}{2}, \frac{1+\Delta}{2}\right)} = e^{-1/6}\sqrt{\frac{2}{n\pi}} \exp\left(-n \cdot \Delta \log\left(1 + \frac{2\Delta}{1-\Delta}\right)\right).$$

$\square$

## C.2   Sufficient conditions on $n_0$ and $n_1$ for $\mathcal{S}_0 \cap \mathcal{S}_1 \subset \mathcal{S}$

**Lemma 7.** *Fix any $\varepsilon \in (0,1)$, and define $n_0, n_1$ as follows:*

$$n_0 = \left\lceil \frac{1}{\eta} \cdot \frac{K}{K-1} \cdot \log\left(\frac{1}{\varepsilon \cdot (K-1)}\right) \right\rceil \vee \left\lceil \frac{1}{\eta}\left(\frac{K}{K-1}\right)^2 \log\left(\frac{K^3 n_1}{(K-1)^2}\right) \right\rceil,$$

$$n_1 = \left\lceil \frac{K-1}{K} \cdot \frac{\log(2T/(K-1))}{\eta\Delta(1-\varepsilon)} + \frac{2}{\Delta} \right\rceil.$$

*Then under $\mathcal{S}_0 \cap \mathcal{S}_1$, it holds that $\widetilde{p}_{n_0+n_1+1} \leq \frac{1}{2T}$.*

*Proof.* Using the same notation as $\widetilde{p}_{n+1}$, let $\widetilde{\theta}_{n+1}$ be the the SGB parameter of arm 1 after its $n$ first pulls.

First, at the end of phase 0 the policy has received $n_0$ rewards of $-1$ from arm 1. Therefore, since $1 - \widetilde{p}_n \geq 1 - \frac{1}{K} = \frac{K-1}{K}$ for all $n \leq n_0$, we have that

$$\widetilde{\theta}_{n_0+1} = -\eta \sum_{n=1}^{n_0}(1 - \widetilde{p}_n) \leq -\frac{K-1}{K} \cdot \eta n_0. \tag{16}$$

Next, we show that the combination of this property and the events occurring under $\mathcal{S}_1$ are sufficient to prove the desired result. Indeed, we recall that under $\mathcal{S}_1$, the sequence of rewards in phase 1, $\{\widetilde{R}_t\}_{t=n_0+1}^{n_0+n_1}$, satisfy the following two conditions:

- The total number of $+1$ rewards should be less than $\frac{1-\alpha\Delta}{2} \cdot n_1$ and the total number of $-1$ rewards should be more than $\frac{1+\alpha\Delta}{2} \cdot n_1$; i.e. $\sum_{t=n_0+1}^{n_0+n_1} \widetilde{R}_t \leq -n_1\alpha\Delta$.

- There should never be more $+1$ rewards than $-1$ rewards observed at the current time step within the phase; i.e. $\sum_{t=n_0+1}^{n_0+k} \widetilde{R}_t \leq 0$ for all $k \leq n_1$.

The rest of the proof is based on characterizing the sequence among all sequences satisfying the above conditions that leads to the largest value for $\widetilde{\theta}_{n_0+n_1+1}$. We call this the *maximizing* sequence. Then, we show that even for that sequence it holds that $\widetilde{p}_{n_0+n_1+1} \leq \frac{1}{2T}$, from which the result of the lemma follows. We detail below the lemmas that lead us to this result.

In Lemma 8, we give a general construction for the maximising sequence that applies to reward sequences satisfying the above conditions for arbitrary initial parameter. Then, in Corollary 2, we show that under the occurence of phase 0, the maximising sequence of rewards with initial parameter $\widetilde{\theta}_{n_0+1}$ takes a simple form: a first period alternating $-1$ and $+1$ rewards followed by a second period with only $-1$ rewards. From this simple form, we can upper-bound $\widetilde{\theta}_{n_0+n_1+1}$ for the maximising sequence (also Corollary 2). Then, in Lemma 9, we show that the proposed tuning for $n_0$ and $n_1$ is sufficient to get that the probability $\widetilde{p}_{n_0+n_1+1}$ is smaller than $\frac{1}{2T}$.

$\square$

From a high-level perspective, we highlight that the phase $\mathcal{S}_0$ serves to mitigate the impact of the order in which rewards are received in the SGB updates, by ensuring that $\widetilde{\theta}_{n_0+t}$ remains "sufficiently" negative throughout the entire phase 1 (i.e. for $t \leq n_1$), while the tuning $n_1 = \mathcal{O}(\log(T))$ permits that a probability of $\frac{1}{2T}$ is achieved even in the identified maximizing scenario.

### C.2.1  Maximizing reward sequence and probability under $\mathcal{S}_0 \cap \mathcal{S}_1$

In this section, we consider reward sequences under $\mathcal{S}_1$ and characterize the maximizing sequence for any arbitrary parameter at the start of phase 1.

**Definition 1** (Constrained trajectories). *Let $\mathcal{Z} \subset \{-1,1\}^{\mathbb{N}}$ be the subset of trajectories of rewards $-1$ and $+1$ that satisfy the conditions imposed under $\mathcal{S}_1$. Let $S_n$ be a finite trajectory of $n$ rewards $\{R_1, \ldots, R_n\} \in \mathcal{Z}$, the feasibility indices $I_+^{\mathcal{Z}}$ and $I_-^{\mathcal{Z}}$ for the $n+1$-th reward are the functions satisfying:*

$$I_+^{\mathcal{Z}}(S_n) = \mathbb{1}(\{R_1, \ldots, R_n, +1\} \in \mathcal{Z}), \text{ and}$$
$$I_-^{\mathcal{Z}}(S_n) = \mathbb{1}(\{R_1, \ldots, R_n, -1\} \in \mathcal{Z}) .$$

*In particular, for $n < n_1$, $I_+^{\mathcal{Z}}(S_n) = 1$ if*

- *the number of $+1$ rewards in $S_n$ is less than $\frac{1-\alpha\Delta}{2} \cdot n_1$.*

- *the number of $+1$ rewards in $S_n$ is less than the number of $-1$ rewards in $S_n$.*

*And $I_-^{\mathcal{Z}}(S_n) = 1$ if the number of $-1$ rewards in $S_n$ is less than $\frac{1+\alpha\Delta}{2} \cdot n_1$.*

In words, the feasibility indices formalize whether a reward of $+1$ or $-1$ can complete or not a sequence of rewards that should verify the constraints imposed under $\mathcal{S}_1$. We now prove the following characterization of the maximising reward sequence.

**Lemma 8.** *Fix $\widetilde{\theta}_1 \in \mathbb{R}^K$. Consider $\{\widetilde{R}_t^M\}_{t=1}^{n_1} \in \mathcal{Z}$ defined as follows with $S_t^M = \{\widetilde{R}_1^M, ..., \widetilde{R}_t^M\}$:*

$$\widetilde{R}_t^M = \begin{cases} -1, & \text{if } \{\widetilde{\theta}_t^M > \frac{K-1}{K}\log(K-1) \text{ and } I_-^{\mathcal{Z}}(S_{t-1}^M) = 1\} \text{ or } I_+^{\mathcal{Z}}(S_{t-1}^M) = 0 \\ +1, & \text{if } \{\widetilde{\theta}_t^M < \frac{K-1}{K}\log(K-1) \text{ and } I_+^{\mathcal{Z}}(S_{t-1}^M) = 1\} \text{ or } I_-^{\mathcal{Z}}(S_{t-1}^M) = 0 \\ +1 \text{ or } -1, & \text{otherwise (the case where } \widetilde{\theta}_t^M = \frac{K-1}{K}\log(K-1) \text{ and either is fine).} \end{cases}$$

*Then $\{\widetilde{R}_t^M\}_{t=1}^{n_1}$ is the maximizing sequence for the SGB parameter of arm 1, i.e. it holds that $\widetilde{\theta}_{n_1+1}^M \geq \widetilde{\theta}_{n_1+1}$, for $\widetilde{\theta}_{n_1+1}^M$ obtained from $\{\widetilde{R}_t^M\}_{t=1}^{n_1}$ and any $\widetilde{\theta}_{n_1+1}$ obtained from a trajectory $\{\widetilde{R}_t\}_{t=1}^{n_1} \in \mathcal{Z}$, both starting from $\widetilde{\theta}_1$.*

*Proof of Lemma 8.* Assume $\{\widetilde{R}_t^M\}_{t=1}^{n_1}$ is not the sequence that maximizes $\widetilde{\theta}_{n_1+1}$. Denote the maximum by $\widetilde{\theta}_{n_1+1}^W$ with reward sequence $\{\widetilde{R}_t^W\}_{t=1}^{n_1} \in \mathcal{Z}$ and corresponding $\widetilde{\theta}_t^W$ / $S_t^W$. Since $\{\widetilde{R}_t^W\}_{t=1}^{n_1}$ is different from $\{\widetilde{R}_t^M\}_{t=1}^{n_1}$, there exists some $n$ such that one of the conditions in the construction of $\widetilde{R}_n^M$ is violated, so either

1. $\widetilde{R}_n^W = +1$, $\widetilde{\theta}_n^W > \frac{K-1}{K}\log(K-1)$ and $I_-^{\mathcal{Z}}(S_{n-1}^W) = 1$.

2. $\widetilde{R}_n^W = -1$, $\widetilde{\theta}_n^W < \frac{K-1}{K}\log(K-1)$ and $I_+^{\mathcal{Z}}(S_{n-1}^W) = 1$.

3. $\widetilde{R}_n^W = +1$ and $I_+^{\mathcal{Z}}(S_{n-1}^W) = 0$.

4. $\widetilde{R}_n^W = -1$ and $I_-^{\mathcal{Z}}(S_{n-1}^W) = 0$.

The last two violations cannot occur since $\{\widetilde{R}_t^W\}_{t=1}^{n_1} \in \mathcal{Z}$. Denote $m$ the largest such $n$. Then handling the first two cases separately, we have

1. $\widetilde{R}_m^W = +1$, $\widetilde{\theta}_m^W > \frac{K-1}{K}\log(K-1)$ and $I_-^{\mathcal{Z}}(S_{m-1}^W) = 1$. We must have $\widetilde{R}_{m+1}^W = -1$. Suppose not: if $\widetilde{R}_{m+1}^W = +1$, then since $I_-^{\mathcal{Z}}(S_m^W) = 1$ and $\widetilde{\theta}_{m+1} \geq \widetilde{\theta}_m > \frac{K-1}{K}\log(K-1)$, $m$ would not be the largest such that condition 1. is satisfied. Now consider an alternative sequence $\widehat{R}_n^W$ in $\mathcal{Z}$ defined as follows:

$$\widehat{R}_n^W = \begin{cases} \widetilde{R}_n^W, & \text{if } n \neq m, m+1 \\ -1, & \text{if } n = m \\ +1, & \text{if } n = m+1 \end{cases}$$

Let $\widehat{\theta}_n^W$ correspond to the sequence of parameters with rewards $\widehat{R}_n^W$. By Lemma 10:

$$\widetilde{\theta}_{n+2}^W < \widehat{\theta}_{n+2}^W.$$

Since $\widehat{R}_n^W = \widetilde{R}_n^W$ for all $n > m+1$, we end up with $\widehat{\theta}_{n_1+1}^W > \widetilde{\theta}_{n_1+1}^W$ contradicting that $\widetilde{\theta}_{n_1+1}^W$ is the maximum.

2. $\widetilde{R}_m^W = -1$, $\widetilde{\theta}_m^W < \frac{K-1}{K}\log(K-1)$ and $I_+^{\mathcal{Z}}(S_{m-1}^W) = 1$. We must have $\widetilde{R}_{m+1}^W = +1$. Suppose not: if $\widetilde{R}_{m+1}^W = -1$, then since $I_+^{\mathcal{Z}}(S_m^W) = 1$ and $\widetilde{\theta}_{m+1} \leq \widetilde{\theta}_m < \frac{K-1}{K}\log(K-1)$, $m$ would not be the largest such that condition 2. is satisfied. Now consider an alternative sequence $\widehat{R}_n^W$ in $\mathcal{Z}$ defined as follows:

$$\widehat{R}_n^W = \begin{cases} \widetilde{R}_n^W, & \text{if } n \neq m, m+1 \\ +1, & \text{if } n = m \\ -1, & \text{if } n = m+1 \end{cases}$$

Let $\widehat{\theta}_n^W$ correspond to the sequence of parameters with rewards $\widehat{R}_n^W$. By Lemma 10:

$$\widetilde{\theta}_{n+2}^W < \widehat{\theta}_{n+2}^W.$$

Since $\widehat{R}_n^W = \widetilde{R}_n^W$ for all $n > m+1$, we end up with $\widehat{\theta}_{n_1+1}^W > \widetilde{\theta}_{n_1+1}^W$ contradicting that $\widetilde{\theta}_{n_1+1}^W$ is the maximum.

In both cases, we get a contradiction, therefore $\{\widetilde{R}_t^W\}_{t=1}^{n_1}$ cannot be the maximizing sequence. Since this holds for any sequence different from $\{\widetilde{R}_t^M\}_{t=1}^{n_1}$, $\{\widetilde{R}_t^M\}_{t=1}^{n_1}$ is the maximizing sequence. We refer to the end of this section for the statement of Lemma 10 and its proof. $\qquad\square$

The next corollary characterizes the maximal possible value of parameter $\widetilde{\theta}_{n_0+n_1+1}$, under event $\mathcal{S}_0 \cap \mathcal{S}_1$ by applying Lemma 8 under the occurrence of the event $\mathcal{S}_0$.

**Corollary 2.** *Assume $\widetilde{\theta}_1 \leq -\frac{K-1}{K} \cdot \eta n_0$ and $n_0 \geq \frac{1}{\eta}\left(\frac{K}{K-1}\right)^2 \log\left(\frac{K^3 n_1}{(K-1)^2}\right)$. Then it holds that*

$$\widetilde{\theta}_{n_1+1} \leq \widetilde{\theta}_{n_1+1}^M \leq -\eta\alpha\Delta n_1 \cdot \left(1 - \frac{1}{K-1}e^{-\frac{K-1}{K}\eta n_0}\right)$$

*with $\widetilde{\theta}_{n_1+1}^M$ being the parameter obtained with the following reward sequence*

$$\widetilde{R}_{nt}^M = \begin{cases} (-1)^t, & \text{if } t \leq (1-\alpha\Delta) \cdot n_1, \\ -1, & \text{otherwise.} \end{cases}$$

The above corollary highlights the role of phase 0 in ensuring that $\widetilde{\theta}_1$ is small enough so that $\widetilde{\theta}_t^M < \frac{K-1}{K}\log(K-1)$ for all $t$ in the trajectory despite the asymmetry of the SGB update.

*Proof of Corollary 2.* By Definition 1, $I_+^{\mathcal{Z}}(S_{t-1}^M) = 0$ if either

- the number of $+1$ rewards up to time $t$ of phase 1 is $\frac{1-\alpha\Delta}{2} \cdot n_1$

- the same number of $+1$ and $-1$ rewards have been observed up to time $t$ of phase 1.

We assumed that $\widetilde{\theta}_1 \leq -\frac{K-1}{K} \cdot \eta n_0 < 0 < \frac{K-1}{K}\log(K-1)$. From Lemma 8, we have that as long as $\widetilde{\theta}_t < \frac{K-1}{K}\log(K-1)$ and $I_+^{\mathcal{Z}}(S_{t-1}^M) = 1$, then $\widetilde{R}_t^M = +1$. However, since $I_+^{\mathcal{Z}}(S_{t-1}^M) = 0$ when the same number of $+1$ and $-1$ rewards have been observed up to time $t$ of phase 1, this leads to an alternation of $-1$ and $+1$: $\widetilde{R}_t^M = (-1)^t$.

This alternation of $-1$ and $+1$ rewards continues until either the parameter $\widetilde{\theta}_t$ becomes greater than $\frac{K-1}{K}\log(K-1)$ or receiving rewards of $+1$ stops being possible ($I_+^{\mathcal{Z}}(S_{t-1}^M) = 0$) because the number of $+1$ rewards up to time $t$ of phase 1 is $\frac{1-\alpha\Delta}{2} \cdot n_1$ (this will occur before $t = n_1$). The former could happen because of the asymmetry of the SGB update. However, we now show that if $n_0$ is large enough / phase 0 is long enough so that $\widetilde{\theta}_1$ is small enough then the asymmetry can be handled so that $\theta_t$ remains negative.

*Intermediate claim:* If $n_0 \geq \frac{1}{\eta}\left(\frac{K}{K-1}\right)^2 \log\left(\frac{K^3 n_1}{(K-1)^2}\right)$, then $\widetilde{\theta}_{2n+1}^M \leq \frac{K-1}{K}\widetilde{\theta}_1 < 0$ for all $n \leq \frac{1-\alpha\Delta}{2} \cdot n_1$.

We prove the intermediate claim by induction. Recall that the SGB update is $\widetilde{\theta}_{t+1} = \widetilde{\theta}_t + \eta(1-\widetilde{p}_t) \cdot \widetilde{R}_t$. A positive reward increases the parameter value and a negative reward decreases it.

Base case $n = 0$ follows from $\widetilde{\theta}_1 \leq 0 \implies \widetilde{\theta}_1 \leq \frac{K-1}{K}\widetilde{\theta}_1$.

Inductive step: assume true for all $k \leq n < \frac{1-\alpha\Delta}{2} \cdot n_1$. Then $\widetilde{R}_t^M = (-1)^t$ for all $t \leq 2(n+1)$ (the alternating sequence has not stopped since $n < \frac{1-\alpha\Delta}{2} \cdot n_1$ so not all $+1$ rewards have been observed and the parameter has remained negative throughout). We have:

$$\widetilde{\theta}_{2(n+1)+1}^M = \widetilde{\theta}_1 + \sum_{t=1}^{2(n+1)} (1 - \widetilde{p}_t^M) \cdot \widetilde{R}_t^M$$

$$= \widetilde{\theta}_1 + \eta \sum_{k=1}^{n+1} \left\{ (1 - \widetilde{p}_{2k}^M) - (1 - \widetilde{p}_{2k-1}^M) \right\}$$

$$= \widetilde{\theta}_1 + \eta \sum_{k=1}^{n+1} \left\{ \widetilde{p}_{2k-1}^M - \widetilde{p}_{2k}^M \right\}.$$

Note that $\widetilde{\theta}_{2k}^M = \widetilde{\theta}_{2k-1}^M - \eta(1 - \widetilde{p}_{2k-1}^M) < \widetilde{\theta}_{2k}^M$ since $\widetilde{R}_{2k-1}^M = -1$. Therefore the sum appearing in the above expression is positive. It stems from the asymmetry of the SGB update. The presence of phase 0 makes $\widetilde{\theta}_1$ small enough such that this asymmetry is not too significant keeping $\widetilde{\theta}_{2(n+1)+1}^M$ of

the same order as $\widetilde{\theta}_1$. Specifically, we have

$$
\begin{aligned}
\widetilde{p}_{2k-1}^M - \widetilde{p}_{2k}^M &= \frac{1}{1 + (K-1) \cdot e^{-\frac{K}{K-1}\widetilde{\theta}_{2k-1}^M}} - \frac{1}{1 + (K-1) \cdot e^{-\frac{K}{K-1}\widetilde{\theta}_{2k}^M}} \\
&= (K-1) \cdot \frac{e^{-\frac{K}{K-1}\widetilde{\theta}_{2k}^M} - e^{-\frac{K}{K-1}\widetilde{\theta}_{2k-1}^M}}{(1 + (K-1)e^{-\frac{K}{K-1}\widetilde{\theta}_{2k-1}^M})(1 + (K-1)e^{-\frac{K}{K-1}\widetilde{\theta}_{2k}^M})} \\
&= \frac{(K-1)e^{-\frac{K}{K-1}\widetilde{\theta}_{2k-1}^M}}{1 + (K-1)e^{-\frac{K}{K-1}\widetilde{\theta}_{2k-1}^M}} \cdot \frac{1}{1 + (K-1)e^{-\frac{K}{K-1}\widetilde{\theta}_{2k}^M}} \cdot \left(e^{\frac{K}{K-1}(\widetilde{\theta}_{2k-1}^M - \widetilde{\theta}_{2k}^M)} - 1\right) \\
&\leq \frac{1}{K-1}\exp\left(\frac{K}{K-1}\widetilde{\theta}_{2k-1}^M\right) \\
&\leq \exp\left(\widetilde{\theta}_1\right) \quad \text{by the inductive hypothesis}
\end{aligned}
$$

$$
\implies \widetilde{\theta}_{2(n+1)+1}^M \leq \widetilde{\theta}_1 + \eta(n+1)e^{\widetilde{\theta}_1} \leq \frac{K-1}{K}\widetilde{\theta}_1.
$$

The last inequality follows from

$$
\begin{aligned}
n_0 \geq \frac{1}{\eta}\left(\frac{K}{K-1}\right)^2 \log\left(\frac{K^3 n_1}{(K-1)^2}\right) &\implies \frac{K^3}{(K-1)^2}n_1 \leq n_0 e^{\eta n_0 (K-1)^2/K^2} \\
&\implies \eta n_1 \leq \frac{1}{K}\left(n_0 \eta \frac{(K-1)^2}{K^2}\right)e^{-\widetilde{\theta}_1} \quad \text{using } \widetilde{\theta}_1 \leq -\left(\frac{K-1}{K}\right)^2 \eta n_0 \\
&\implies \eta n_1 e^{\widetilde{\theta}_1} \leq -\frac{1}{K}\widetilde{\theta}_1 \quad \text{using again } \widetilde{\theta}_1 \leq -\left(\frac{K-1}{K}\right)^2 \eta n_0 \\
&\implies \eta(n+1)e^{\widetilde{\theta}_1} \leq -\frac{1}{K}\widetilde{\theta}_1,
\end{aligned}
$$

using that $n \leq \frac{1-\alpha\Delta}{2} \cdot n_1 < \frac{n_1}{2} \leq n_1 - 1$. This completes the proof of the intermediate claim.

The claim implies that throughout the alternating sequence of $-1$ and $+1$ rewards, the parameter $\widetilde{\theta}_t$ remains negative and that observing all the $+1$ rewards causes the alternating sequence to terminate. We have that $\widetilde{R}_t^M = (-1)^t$ lasts until we have observed at most $\frac{1-\alpha\Delta}{2} \cdot n_1$ rewards of $+1$ after which $I_+^Z(S_{t-1}^M) = 0$ for all remaining rounds and therefore the remaining rewards in the maximizing sequence are all $-1$.

When the alternation ends, by the above claim we have that after receiving $t_0 = (1-\alpha\Delta) \cdot n_1$ samples, it holds that $\widetilde{\theta}_{2n+1}^M \leq \frac{K-1}{K}\widetilde{\theta}_1 \leq -\left(\frac{K-1}{K}\right)^2 \eta n_0$. There are $n_1 - t_0 = \alpha\Delta n_1$ rounds left in the phase, all of which have a reward of $-1$. Hence we have

$$
\widetilde{p}_{t_0+t}^M \leq \frac{1}{K-1}e^{\frac{K}{K-1}\widetilde{\theta}_{t_0+t}^M} \leq \frac{1}{K-1}e^{\frac{K}{K-1}\widetilde{\theta}_{t_0}^M} \leq \frac{1}{K-1}e^{-\frac{K-1}{K}\eta n_0}.
$$

Fixing $\delta_0 = \frac{1}{K-1}e^{-\frac{K-1}{K}\eta n_0}$, we have:

$$
\begin{aligned}
\widetilde{\theta}_{n_1+1}^M = \theta_{t_0+(n_1-t_0)+1}^M &= \widetilde{\theta}_{t_0+1}^M - \eta \sum_{k=t_0+1}^{n_1}(1 - \widetilde{p}_k^M) \\
&\leq -\eta(n_1 - t_0)(1 - \delta_0) \\
&= -\eta\alpha\Delta n_1(1 - \delta_0) \\
\implies \widetilde{\theta}_{n_1+1} &\leq -\eta\alpha\Delta n_1(1 - \delta_0).
\end{aligned}
$$

$\square$

With the previous results at hand, we can finally obtain the sufficient condition on $n_0$ and $n_1$ to guarantee that $\mathcal{S}_0 \cap \mathcal{S}_1 \subset \mathcal{S}$.

**Lemma 9.** *Fix any $\varepsilon \in (0, 1)$, and define $n_0, n_1$ as follows:*

$$n_0 = \left\lceil \frac{1}{\eta} \cdot \frac{K}{K-1} \cdot \log\left(\frac{1}{\varepsilon \cdot (K-1)}\right) \right\rceil \vee \left\lceil \frac{1}{\eta}\left(\frac{K}{K-1}\right)^2 \log\left(\frac{K^3 n_1}{(K-1)^2}\right) \right\rceil,$$

$$n_1 = \left\lceil \frac{K-1}{K} \cdot \frac{\log(2T/(K-1))}{\eta\Delta(1-\varepsilon)} + \frac{2}{\Delta} \right\rceil.$$

*Then under $\mathcal{S}_0 \cap \mathcal{S}_1$, it holds that $\widetilde{p}_{n_0+n_1+1} \le \frac{1}{2T}$.*

*Proof.* Under $\mathcal{S}_0$, we have that $\widetilde{\theta}_{n_0+1} \le -\frac{K-1}{K} \cdot \eta n_0$ and by construction,

$$n_0 \ge \frac{1}{\eta} \cdot \left(\frac{K}{K-1}\right)^2 \cdot \log\left(\frac{K^3 n_1}{(K-1)^2}\right).$$

Therefore, we apply Corollary 2 on the rewards in phase 1, which gives

$$
\begin{aligned}
\widetilde{\theta}_{n_0+n_1+1} &\le -\eta\alpha\Delta n_1 \left(1 - \frac{1}{K-1}e^{-\frac{K-1}{K}\eta n_0}\right) \\
&\le -\eta\alpha\Delta(1-\varepsilon)n_1 \quad \text{since } n_0 \ge \frac{1}{\eta} \cdot \frac{K}{K-1} \cdot \log\left(\frac{1}{\varepsilon \cdot (K-1)}\right) \\
&\le -\eta\Delta(1-\varepsilon)n_1 + 2\eta(1-\varepsilon) \quad \text{since } \alpha > 1 - \frac{2}{\Delta n_1} \\
&\le -\frac{K}{K-1}\log\left(\frac{2T}{K-1}\right) \quad \text{since } n_1 \ge \frac{K-1}{K} \cdot \frac{\log(2T/(K-1))}{\eta\Delta(1-\varepsilon)} + \frac{2}{\Delta} \\
\implies \widetilde{p}_{n_0+n_1+1} &= \frac{e^{\widetilde{\theta}_{n_0+n_1+1}}}{e^{\widetilde{\theta}_{n_0+n_1+1}} + (K-1)e^{-\frac{\widetilde{\theta}_{n_0+n_1+1}}{K-1}}} \le \frac{e^{\frac{K}{K-1}\cdot\widetilde{\theta}_{n_0+n_1+1}}}{(K-1)} \le \frac{1}{2T}.
\end{aligned}
$$

$\square$

We conclude this section with Lemma 10, which is the last technical result supporting the proof of Lemma 7

**Lemma 10.** *Consider an arbitrary parameter $\theta_n$ for* SGB *at the $n$-th pull of arm 1 (and the parameters of all other arms are the same and equal to $-\frac{1}{K-1}\theta_n$). Let $R_k \in \{-1, 1\}$ be the reward for the $k$-th pull. Let $\theta_{n+2}$ correspond to the case where $R_n = 1, R_{n+1} = -1$ and $\widetilde{\theta}_{n+2}$ correspond to the case where $R_n = -1, R_{n+1} = 1$.*

- *If $\theta_n > \frac{K-1}{K}\log(K-1)$, $\theta_{n+2} < \widetilde{\theta}_{n+2}$*

- *If $\theta_n = \frac{K-1}{K}\log(K-1)$, $\widetilde{\theta}_{n+2} = \theta_{n+2}$*

- *If $\theta_n < \frac{K-1}{K}\log(K-1)$, $\theta_{n+2} > \widetilde{\theta}_{n+2}$*

*Proof of Lemma 10.*

$$\widetilde{\theta}_{n+2} = \theta_n - \eta(1-p_n) + \eta(1-\widetilde{p}_{n+1}) = \theta_n - \eta(1-p_n) + \eta\frac{K-1}{K-1+\exp(\frac{K}{K-1}\widetilde{\theta}_{n+1})}$$

$$= \theta_n - \eta(1-p_n) + \eta\frac{K-1}{K-1+\exp(\frac{K}{K-1}\theta_n - \frac{K}{K-1}\eta(1-p_n))}$$

$$\theta_{n+2} = \theta_n + \eta(1-p_n) - \eta(1-p_{n+1}) = \theta_n + \eta(1-p_n) - \eta\frac{K-1}{K-1+\exp(\frac{K}{K-1}\theta_{n+1})}$$

$$= \theta_n + \eta(1-p_n) - \eta\frac{K-1}{K-1+\exp(\frac{K}{K-1}\theta_n + \frac{K}{K-1}\eta(1-p_n))}$$

$$\implies \theta_{n+2} - \widetilde{\theta}_{n+2} = 2\eta(1-p_n) - \eta\frac{K-1}{K-1+\exp(\frac{K}{K-1}\theta_n + \frac{K}{K-1}\eta(1-p_n))}$$

$$- \eta\frac{K-1}{K-1+\exp(\frac{K}{K-1}\theta_n - \frac{K}{K-1}\eta(1-p_n))}.$$

Using that $1-p_n = \frac{K-1}{K-1+\exp(\frac{K}{K-1}\theta_n)}$, consider the function

$$f(x) = \frac{2\eta(K-1)}{K-1+\exp(x)} - \frac{\eta(K-1)}{K-1+\exp(x + \frac{K}{K-1}\frac{\eta(K-1)}{K-1+\exp(x)})}$$

$$- \frac{\eta(K-1)}{K-1+\exp(x - \frac{K}{K-1}\frac{\eta(K-1)}{K-1+\exp(x)})}.$$

Then denoting $Y = \exp(\frac{K}{K-1}\frac{\eta(K-1)}{K-1+\exp(x)})$ and $Z = K-1$,

$$f(x) > 0 \iff \frac{2}{K-1+\exp(x)} > \frac{1}{K-1+\exp(x + \frac{K}{K-1}\frac{\eta(K-1)}{K-1+\exp(x)})}$$

$$+ \frac{1}{K-1+\exp(x - \frac{K}{K-1}\frac{\eta(K-1)}{K-1+\exp(x)})}$$

$$\iff \frac{2}{Z+\exp(x)} > \frac{1}{Z+Y\exp(x)} + \frac{1}{Z+\frac{1}{Y}\exp(x)}$$

$$\iff 2Z^2 + 2ZYe^x + \frac{2Z}{Y}e^x + 2e^{2x} > 2Z^2 + ZYe^x + \frac{Z}{Y}e^x + 2Ze^x + Ye^{2x} + \frac{1}{Y}e^{2x}$$

$$\iff ZYe^x + \frac{Z}{Y}e^x + 2e^{2x} > 2Ze^x + Ye^{2x} + \frac{1}{Y}e^{2x}$$

$$\iff ZY^2 + Z + 2Ye^x > 2ZY + Y^2e^x + e^x$$

$$\iff (Y^2 - 2Y + 1)(Z - e^x) > 0$$

$$\iff (Y-1)^2(Z - e^x) > 0$$

$$\iff x < \log Z = \log(K-1).$$

Similarly, $f(x) < 0 \iff x > \log(K-1)$ and $f(x) = 0 \iff x = \log(K-1)$. Since $f(\frac{K}{K-1}\theta_n) = \theta_{n+2} - \widetilde{\theta}_{n+2}$, we have the stated result. $\qquad\square$

# D   Supporting results for the proof of Theorem 3

This section contains the classical lower bound results in bandits (appendix D.1) and the proof of Lemma 1 (appendix D.2). The proof of Theorem 3 in the main paper is entirely based on the results presented in this section.

## D.1   Details on asymptotic lower bounds in bandits with bounded-support

We denote by $\mathrm{KL}(\cdot, \cdot)$ the Kullback-Leibler divergence between two distributions. We first state an adaptation of the classical asymptotic lower bound in MAB, first proved by Lai and Robbins

[36] for parametric families of distributions, and later extended by Burnetas and Katehakis [37] for non-parametric families. The following result is inspired by the elegant formulation and proof of this lower bound, that can be found in Emilie Kaufmann's thesis [67, Theorem 1.2].

**Theorem 6** (Asymptotic problem-dependent regret lower bound, adapted from Theorem 1.2 of [67])**.** *Let $\nu \in \mathcal{F}^K$ be a MAB instance with distributions from the same family of distributions $\mathcal{F}$. Let $\alpha \in (0, 1)$, and assume that a policy $\pi$ satisfies $\mathcal{R}_T = \mathcal{O}(T^\alpha)$ for any instance $\nu \in \mathcal{F}^K$. Then, the number of pulls of any sub-optimal arm $k$ (such that $\Delta_k > 0$) under the policy $\pi$ must satisfy*

$$\liminf_{T \to \infty} \frac{\mathbb{E}[N_k(T)]}{\log(T)} \geq \frac{1 - \alpha}{\mathcal{K}_{\mathrm{inf}}^{\mathcal{F}}(\nu_k, \mu^\star)} \ , \quad \mathcal{K}_{\mathrm{inf}}^{\mathcal{F}}(\nu_k, \mu^\star) = \inf_{\nu' \in \mathcal{F}} \{\mathrm{KL}(\nu_k, \nu') : \mathbb{E}_{X \sim \nu'}[X] > \mu^\star\} \ , \quad (17)$$

*for any sub-optimal arm $k$, and using the notation $\mu^\star = \max_{j \in [K]} \mu_j$.*

The above theorem holds for any generic family $\mathcal{F}$, but we recall that in this paper we use the notation $\mathcal{F}$ for the family of distributions supported on $[-1, 1]$. Furthermore, in Theorem 3 we consider even more specifically the family $\mathcal{F}_\Delta^K$, for some $\Delta \in (0, 1)$, of bandit instances with bounded rewards (with support $[-1, 1]$) and minimum gap larger than $\Delta$. To prove the theorem, we derive the following result.

**Lemma 11** ($\mathcal{K}_{\mathrm{inf}}^{\mathcal{F}_\Delta^K}$ for the Dirac distributions of Lemma 1)**.**

$$\forall \Delta \in (0, 1) : \quad \mathcal{K}_{\mathrm{inf}}^{\mathcal{F}_\Delta^K}(\delta_{-\Delta}, 0) = \log\left(1 + \frac{2\Delta}{1 - \Delta}\right)$$

*Proof.* We first obtain from Lemma 12 below that $\mathcal{K}_{\mathrm{inf}}^{\mathcal{F}_\Delta^K}(\delta_{-\Delta}, 0) = \mathcal{K}_{\mathrm{inf}}^{\mathcal{F}}(\delta_{-\Delta}, \Delta)$. The result then follows from Lemma 13. $\square$

We first prove the relation between KL-inf quantity associated $\mathcal{F}_\Delta^K$ and the one associated with $\mathcal{F}$.

**Lemma 12** (Modified $\mathcal{K}_{\mathrm{inf}}$ with knowledge of minimum gap)**.** *Fix $\Delta \in (0, 1)$, and let $\mathcal{F}_\Delta^K$ denote the class of $K$-armed bandit instances with distributions bounded in $[-1, 1]$ and with a minimal non-zero gap of $\Delta$. Then, for any $\nu_k \in \mathcal{F}$ satisfying $\Delta_k \geq \Delta$, it holds that*

$$\mathcal{K}_{\mathrm{inf}}^{\mathcal{F}_\Delta^K}(\nu_k, \mu^\star) = \mathcal{K}_{\mathrm{inf}}^{\mathcal{F}}(\nu_k, \mu^\star + \Delta).$$

*Proof.* The quantity $\mathcal{K}_{\mathrm{inf}}^{\mathcal{F}_\Delta^K}(\nu_k, \mu^\star)$ reflects how far the distribution $\nu_k$ is from an alternative distribution $\nu'_k$ such that arm $k$ would be optimal under this reward distribution. By assuming that the minimum gap is $\Delta$, it is necessary that the average of the distribution $\nu'_k$ is at least $\mu^\star + \Delta$. Since the support of the distribution has to remain in $[-1, 1]$, the identity is direct. $\square$

We then prove the remaining supporting result for the proof of Lemma 11.

**Lemma 13** ($\mathcal{K}_{\mathrm{inf}}^{\mathcal{F}}$ for bounded distributions)**.** *Let $\mu_0 \in [-1, 1]$ and $\mu_1 \in [-1, 1]$ satisfying $\mu_1 \geq \mu_0$. Then,*

$$\mathcal{K}_{\mathrm{inf}}^{\mathcal{F}}(\delta_{\mu_0}, \mu_1) = \log\left(\frac{1 - \mu_0}{1 - \mu_1}\right).$$

*Proof.* To turn the Dirac distribution of mean $\mu_0$ into a distribution of mean $\mu_1$ at minimum (KL-)cost while keeping its support in $[-1, 1]$, the most efficient way is to transfer as little mass as possible in $+1$. We thus solve

$$p \cdot \mu_0 + (1 - p) = \mu_1 \Longrightarrow p = \frac{1 - \mu_1}{1 - \mu_0} \ ,$$

which is the probability mass remaining in $\mu_0$ after the transfer. This gives the result. $\square$

We now formulate some remarks, before recalling some known properties of KL-inf divergences for bounded distributions.

**Remark 1.** *If the support $[-R, R]$ for $R > 0$, and using the notation $\mathcal{F}_{[-R,R]}$ for this family of distributions, we obtain*

$$\mathcal{K}_{\inf}^{\mathcal{F}_{[-R,R]}}(\delta_{\mu_0}, \mu_1) = \frac{R - \mu_0}{R - \mu_1},$$

*so the thresholds of Theorem 3 are trivially multiplied by $R$, since $\log(1 + \Delta)$ becomes $\log(1 + \Delta/R)$ in the lower bound.*

**Remark 2.** *From Lemma 12 we see that $\mathcal{K}_{\inf}^{\mathcal{F}_\Delta^K}(., \mu^\star)$ is infinite if $\mu^\star \geq 1 - \Delta$. This reflects that, in that case, the regret does not have to be logarithmic. For instance, by adapting the analysis from [68] we could verify that preceding a UCB policy by a greedy step that would play any arm with empirical mean larger than $1 - \Delta$ (instead of computing the UCB) would yield constant regret in the case where the best arm admits an expectation larger than $1 - \Delta$.*

*Since this fact is not particularly insightful to understand the theoretical behavior of $\mathcal{SGB}$, and the interesting scenarios arise when $\Delta$ is small, we chose to ignore this trick in the paper.*

**Summary of known properties of $\mathcal{K}_{\inf}^{\mathcal{F}}$**    It is well known that for bounded distributions the quantity $\mathcal{K}_{\inf}^{\mathcal{F}}(\nu_k, \mu^\star)$ can be expressed as the solution of an optimization problem, but is also related to the KL-divergence of Bernoulli distributions and to $\Delta_k^2$. We now elaborate on these results.

Most existing results in the literature are derived for bounded distributions of support $[0, 1]$ ([50, 30, 69], among many others). We denote this family by $\bar{\mathcal{F}}$. Since the KL divergence is invariant by rescaling, it is clear that $\mathcal{K}_{\inf}^{\mathcal{F}}$ and $\mathcal{K}_{\inf}^{\bar{\mathcal{F}}}$ are related, more precisely

$$\forall (\nu, \mu) \in \mathcal{F} \times [-1, 1], \ \mathcal{K}_{\inf}^{\mathcal{F}}(\nu, \mu) = \mathcal{K}_{\inf}^{\bar{\mathcal{F}}}(\bar{\nu}, \bar{\mu}),$$

with $\bar{\mu} = \frac{1+\mu}{2}$ is the distribution of $\frac{1+X}{2}$ for a random variable $X \sim \nu$, and similarly for $\bar{\nu}$. Honda and Takemura [50] derived the following expression,

$$\forall (\bar{\nu}, \bar{\mu}) \in \bar{\mathcal{F}} \times [0, 1], \quad \mathcal{K}_{\inf}^{\bar{\mathcal{F}}}(\bar{\nu}, \bar{\mu}) = \sup_{\lambda \in [0,1]} \mathbb{E}_{X \sim \bar{\nu}} \left[ \log \left( 1 - \lambda \frac{X - \bar{\mu}}{1 - \bar{\mu}} \right) \right] .$$

We further denote by $\mathrm{kl}$ the Bernoulli divergence, that can be expressed as

$$\forall (\mu_0, \mu_1) \in [0, 1], \quad \mathrm{kl}(\mu_0, \mu_1) = \mu_0 \log \left( \frac{\mu_0}{\mu_1} \right) + (1 - \mu_0) \log \left( \frac{1 - \mu_0}{1 - \mu_1} \right) .$$

Then, for any distribution $\nu_0 \in \bar{\mathcal{F}}$ of mean $\mu_0$ and $\mu_1 \in [\mu_0, 1]$ it is established that

$$\mathcal{K}_{\inf}^{\bar{\mathcal{F}}}(\nu_0, \mu_1) \geq \mathrm{kl}(\mu_0, \mu_1) \geq 2(\mu_1 - \mu_0)^2$$

We refer to [69] for proofs and detailed discussions on the tightness of the inequalities. In particular, the inequalities are tight for Bernoulli instances with expectation near $1/2$ and small gaps. This translates into the fact that, **on $\mathcal{F}$, the most difficult distributions are Rademacher with means near $0$.**

Hence, for problems from $\mathcal{F}^K$ the regret of uniformly efficient policies is expected to scale in the worst case, as a function of the gap $\Delta$, with $2(K-1) \cdot \frac{\log(T)}{\Delta}$, which becomes $\frac{(K-1)}{2\Delta} \cdot \log(T)$ if $\Delta$ is assumed to be known, i.e. $\nu \in \mathcal{F}_\Delta^K$. This is nearly matched for an instance with $K - 1$ sub-optimal Rademacher arms of mean $0$ (and so $\mu_1 = \Delta$). We see that the results we obtained in Theorem 1 and Theorem 4 verify this property, and are thus coherent with the lower bound.

To be more precise, the asymptotic scaling of the regret upper bound of Theorem 1 is $\frac{\log(T)}{2\eta}$ for small enough learning rates, while the lower bound on the instance $\nu_1 = \mathrm{Rad}(\Delta)$, $\nu_2 = \mathrm{Rad}(0)$ is $\frac{2\Delta \log(T)}{-\log(1-4\Delta^2)}$, so the two bounds can be arbitrarily close for $\eta = \Delta e^{-2\Delta}$ (satisfying the condition of the theorem) and $\Delta \to 0$.

## D.2   Proof of Lemma 1

We start by restating the result presented in the main paper.

**Lemma 1** (Regret upper bound on an easy instance). *Let $\nu \in \mathcal{F}^K$ be a MAB defied by $\nu_1 = \delta_0$ and $\nu_2 = \cdots = \nu_K = \delta_{-\Delta}$, for some $\Delta > 0$. Then, for **any** learning rate $\eta$, SGB satisfies*

$$\forall \varepsilon \in (0,1), \ \mathcal{R}_T^{SGB} \leq \frac{1 + \log\left(1 + (K-1)T\eta\Delta\right)}{(1-\varepsilon)\eta} + \frac{K^2}{\varepsilon} \cdot \left(\Delta + \frac{1}{\eta}\log\left(\frac{K}{\varepsilon}\right)\right).$$

*Proof.* Since the probabilities are only updated when a sub-optimal arm is played, we introduce the notation $(\widetilde{p}_{k,N})_{N \in \mathbb{N}}$ to denote the sampling probabilities at the $N$-th pull of a sub-optimal arm. We use notation $N_t = \sum_{k=2}^K N_k(t)$, where we recall that, for $k \in [K]$, $N_k(t) = \sum_{s=1}^t \mathbb{1}(A_s = k)$. Then, we can first remark that

$$\forall t \geq 1, \ \theta_{1,t+1} = \theta_{1,t} + p_{1,t} \cdot \eta\Delta\mathbb{1}(A_t \neq 1) \implies \theta_{1,t} = \eta\Delta \cdot \sum_{n=1}^{N_t} \widetilde{p}_{1,n},$$

On the other hand, the parameters of sub-optimal arms satisfy

$$\begin{aligned}
\forall k \geq 2, \ \theta_{k,t+1} &= \theta_{k,t} - \eta(1 - p_{k,t})\Delta\mathbb{1}(A_t = k) + \eta p_{k,t}\Delta\mathbb{1}(A_t \neq \{k,1\}) \\
&= \theta_{k,t} - \eta\Delta \cdot ((1 - p_{k,t})\mathbb{1}(A_t = k) - p_{k,t}\mathbb{1}(A_t \neq \{k,1\})) \\
&= \theta_{k,t} - \eta\Delta \cdot (\mathbb{1}(A_t = k) - p_{k,t}\mathbb{1}(A_t \neq 1)) \\
&\implies \theta_{k,t} = -\eta\Delta N_k(t) + \eta\Delta \sum_{n=1}^{N_t} \widetilde{p}_{k,n} \ .
\end{aligned}$$

Thus, we obtain that

$$\forall k \in \{2, \ldots, K\}, \forall t \geq 1, \quad \theta_{1,t} - \theta_{k,t} = \eta\Delta N_k(t) + \eta\Delta \sum_{n=1}^{N_t}(\widetilde{p}_{1,n} - \widetilde{p}_{k,n}). \tag{18}$$

We additionally prove that $\widetilde{p}_{1,n} \geq \max_{k \geq 2} \widetilde{p}_{k,n}$ by induction: this is true at initialization because all probabilities are equal. Then if the property holds for some time $t \geq 1$, it then holds that

$$\max_{k=\{2,\ldots,K\}} \delta\theta_{k,t} \leq \Delta \cdot \max_{k \neq 1} p_{k,t}\mathbb{1}(A_t \neq 1) \leq \Delta \cdot p_{1,t}\mathbb{1}(A_t \neq 1) = \delta\theta_{1,t} \ ,$$

so $\theta_{1,t+1} = \max_{j \in [K]} \theta_{j,t+1}$, therefore $p_{1,t+1}$ remains the maximum sampling probability at the next step. We deduce that

$$\forall t \geq 1, \forall k \geq 2 : \ p_{k,t} \leq \frac{e^{\theta_{k,t}}}{e^{\theta_{1,t}}} = e^{-\eta\Delta N_k(t) - \eta\Delta \sum_{s=1}^{t-1}(p_{1,s} - p_{k,s})\mathbb{1}(A_s \neq 1)} \leq e^{-\eta\Delta N_k(t)} \ . \tag{19}$$

Consider $\mathcal{T}_k = \{t \geq 1 : p_{k,t} \geq \varepsilon/K\}$. From the last bound of (19), we first obtain that

$$N_k(t) \geq N_k^\varepsilon := \left\lceil \frac{1}{\eta\Delta} \cdot \log\left(\frac{K}{\varepsilon}\right) \right\rceil \implies \forall s \geq t : \ p_{k,s} \leq \frac{\varepsilon}{K} \ .$$

Then, $\mathbb{E}[|\mathcal{T}_k|]$ is trivially upper bounded by $N_k^\varepsilon \cdot \frac{K}{\varepsilon}$, by conditional independence of the number of steps between two successive pulls of arm $k$. Thus, it holds that

$$\mathbb{E}[|\mathcal{T}_k|] \leq \frac{K}{\varepsilon\,\eta\Delta} \cdot \log\left(\frac{K}{\varepsilon}\right) + \frac{K}{\varepsilon}$$

Now, considering $\mathcal{T} = \cup_{k=2}^K \mathcal{T}_k$, we can simply write that

$$\mathbb{E}[|\mathcal{T}|] \leq \sum_{k=2}^K \mathbb{E}[|\mathcal{T}_k|] \leq \frac{K(K-1)}{\varepsilon\,\eta\Delta}\log\left(\frac{K}{\varepsilon}\right) + \frac{K(K-1)}{\varepsilon} \ .$$

We can now consider the number of sub-optimal pulls incurred for $t \notin \mathcal{T}$. We can get back to using Eq. (19), but now exploiting the penultimate inequality with

$$p_{k,t} \leq e^{-\eta\Delta(N_k(t) + \sum_{s=1}^{t-1}(p_{1,s} - p_{k,s})\mathbb{1}(A_s \neq 1))} \leq e^{-\eta\Delta(N_k(t) + \sum_{s=1}^{t-1}(1-\varepsilon)\mathbb{1}(A_s \neq 1, s \notin \mathcal{T}))},$$

since by design $t \notin \mathcal{T} \Rightarrow p_{1,t} - p_{k,t} = 1 - \sum_{k'=2}^{K} p_{k',t} - p_{k,t} \geq 1 - \varepsilon$. We denote by $\overline{\mathcal{T}}$ the complementary of $\mathcal{T}$ (so $\mathcal{T} \cup \overline{\mathcal{T}} = [T]$ and $\mathcal{T} \cap \overline{\mathcal{T}} = \emptyset$), and by $\overline{N}_t$ the number of sub-optimal pulls within this range, so $\overline{N}_t = \sum_{s=1}^{T} \mathbb{1}(A_s \neq 1, s \notin \mathcal{T})$. To obtain the result, we just ignore the $N_k(t)$ term in the upper bound, and use a very standard proof scheme in bandits. We first define $N_T^\varepsilon = \frac{\log(1+(K-1)\eta\Delta T)}{(1-\varepsilon)\eta\Delta}$, and obtain that

$$
\mathbb{E}\left[\sum_{t=1, t \notin \mathcal{T}}^{T} \mathbb{1}(A_t \neq 1)\right] = \mathbb{E}[\overline{N}_T]
$$

$$
\leq \mathbb{E}\left[\sum_{t \in \overline{\mathcal{T}} \cap [T]} \mathbb{1}\left(A_t \neq 1, \overline{N}_t \leq N_T^\varepsilon\right) + \sum_{t \in \overline{\mathcal{T}} \cap [T]} \mathbb{1}\left(A_t \neq 1, \overline{N}_t > N_T^\varepsilon\right)\right]
$$

$$
\leq N_T^\varepsilon + T(K-1)e^{-\eta\Delta(1-\varepsilon)N_T^\varepsilon}
$$

$$
\leq \frac{\log(1 + T(K-1)\eta\Delta)}{(1-\varepsilon)\eta\Delta} + \frac{(K-1)T}{1 + T(K-1)\eta\Delta}
$$

$$
\leq \frac{1 + \log(1 + T(K-1)\eta\Delta)}{(1-\varepsilon)\eta\Delta} \,,
$$

which concludes the proof since

$$
\mathcal{R}_T(\nu) = \Delta\mathbb{E}\left[\sum_{t=1}^{T} \mathbb{1}(A_t \neq 1)\right] = \Delta\mathbb{E}\left[\sum_{t=1, t \notin \mathcal{T}}^{T} \mathbb{1}(A_t \neq 1) + \sum_{t=1, t \in \mathcal{T}}^{T} \mathbb{1}(A_t \neq 1)\right]
$$

$$
\leq \frac{1 + \log(1 + T(K-1)\eta\Delta)}{(1-\varepsilon)\eta} + \Delta\mathbb{E}[|\mathcal{T}|]
$$

$$
\leq \frac{1 + \log(1 + T(K-1)\eta\Delta)}{(1-\varepsilon)\eta} + \frac{K^2}{\varepsilon}\left(\Delta + \frac{1}{\eta}\log\left(\frac{K}{\varepsilon}\right)\right).
$$

$\square$

**Possible improvement** In the last steps of the proof of Lemma 1, we used Eq. (19) and ignored for simplicity the $N_k(t)$ term in the following bound

$$
p_{k,t} \leq e^{-\eta\Delta(N_k(t) + \sum_{s=1}^{t}(1-\varepsilon)\mathbb{1}(A_s \neq 1, s \notin \mathcal{T}))} \,.
$$

Intuitively, the symmetry between the arms and the fact that the rewards are deterministic strongly suggest that it should hold that $N_k(t) \approx \frac{\overline{N}_t}{K-1}$ with high probability, at least when $\overline{N}_t$ becomes large enough. Hence, this suggest that we could decrease $\overline{N}_T^\varepsilon$ by a factor $\frac{K-1}{K}$, and obtain a bound

$$
\mathcal{R}_T \leq \frac{1}{1-\varepsilon} \cdot \frac{K-1}{K} \cdot \frac{\log(T)}{\eta} + A_\varepsilon \,,
$$

for some constant $A_\varepsilon$. Proving it for a general number of arms seems intricate. Furthermore, for large $K$ the difference between the two results is minor –and our result already gives the inverse linear dependency in $K$– so we leave further investigation for future works. However, we now prove this improved bound for $K = 2$ to formally justify this intuition, and support the proof of Theorem 3 in the case $K = 2$. We can further remark that the improvement is not only on the $1/2$ factor in front of the logarithmic term, but also is the dependency in $\varepsilon$, which becomes $\log\left(\frac{1}{\varepsilon}\right)$ instead of $\varepsilon^{-1}$.

**Lemma 14** (Improved upper bound on the easy instance for $K = 2$). *Let $\nu \in \mathcal{F}^K$ be a MAB defined by $\nu_1 = \delta_0$ and $\nu_2 = \delta_{-\Delta}$, for some $\Delta > 0$. Then, for **any** learning rate $\eta$, SGB satisfies*

$$
\forall \varepsilon \in (0,1), \quad \mathcal{R}_T \leq \frac{\log(1 + 2T\varepsilon(1-\varepsilon)\eta\Delta) + 1}{2(1-\varepsilon)\eta} + \frac{\log\left(\frac{1}{\varepsilon}\right)}{\eta}.
$$

*Proof.* The improved result comes from the fact that for $K = 2$, $\theta_{1,t} = -\theta_{2,t}$ at all steps (Eq. (9)), which simplifies several arguments. Let $\widetilde{\theta}_N$ denote the parameter of arm 1 after $N$ pulls of arms 2.

Using that $\inf_{t \geq 1} p_{1,t} = p_{1,1} = 1/2$, we can now prove that a deterministic number of samples $N_\varepsilon$ suffices to reach $\widetilde{p}_{1,N} \geq 1 - \varepsilon$. Indeed, it holds that

$$1 - \widetilde{p}_{N_\varepsilon} \leq e^{-2\widetilde{\theta}_{N_\varepsilon}}, \quad \text{and} \quad \widetilde{\theta}_{N_\varepsilon} \geq \frac{\eta \Delta N_\varepsilon}{2} \ ,$$

from which we deduce that choosing $N_\varepsilon = \frac{\log\left(\frac{1}{\varepsilon}\right)}{\eta \Delta}$ is sufficient to guarantee that $\inf_{N \geq N_\varepsilon} \widetilde{p}_N \geq 1 - \varepsilon$. Let us now consider a number $\overline{N}$ of selections of arm 2 after its $N_\varepsilon$ first pulls. For $N = N_\varepsilon + \overline{N} \geq N_\varepsilon$, it then holds that

$$\widetilde{\theta}_N \geq \widetilde{\theta}_{N_\varepsilon} + \eta \Delta (1 - \varepsilon) \overline{N} \ ,$$

which yields $1 - \widetilde{p}_N \leq e^{-2\widetilde{\theta}_N} \leq \varepsilon \cdot e^{-2\eta \Delta (1 - \varepsilon) \overline{N}}$ by definition of $N_\varepsilon$. Using the same last steps as for the proof of Lemma 1, and choosing $\overline{N} = \frac{\log(1 + 2T\varepsilon(1-\varepsilon)\eta\Delta)}{2(1-\varepsilon)\eta\Delta}$, we then obtain that

$$\mathbb{E}[N_2(T)] \leq \mathbb{E}\left[\sum_{t=1}^{T} \mathbb{1}\left(A_t = 2, N_2(t) \leq N_\varepsilon + \overline{N}\right) + \sum_{t=1}^{T} \mathbb{1}\left(A_t = 2, N_2(t) > N_\varepsilon + \overline{N}\right)\right]$$

$$\leq N_\varepsilon + \overline{N} + T\varepsilon \cdot e^{-2\eta\Delta(1-\varepsilon)\overline{N}}$$

$$\leq \frac{\log\left(\frac{1}{\varepsilon}\right)}{\eta\Delta} + \frac{\log(1 + 2T\varepsilon(1-\varepsilon)\eta\Delta)}{2(1-\varepsilon)\eta\Delta} + \frac{\varepsilon \cdot T}{1 + 2T\varepsilon(1-\varepsilon)\eta\Delta}$$

$$\leq \frac{\log(1 + 2T\varepsilon(1-\varepsilon)\eta\Delta) + 1}{2(1-\varepsilon)\eta\Delta} + \frac{\log\left(\frac{1}{\varepsilon}\right)}{\eta\Delta}$$

which gives the result. $\qquad\qquad\qquad\qquad\qquad\qquad\qquad\qquad\qquad\qquad\qquad\qquad\qquad\qquad\square$

### D.3  Conjectures on the critical threshold for $K \geq 3$

We recall that, by combining Theorem 2 and 3, we proved that $\eta \lesssim \Delta \wedge K^{-1}$ is a necessary condition to guarantee logarithmic regret for SGB. We also hinted in the main paper (Section 3 and experiments of Section 4) that we believe that for general $K$-armed problems $\eta$ should scale with $\Delta/K$, and more precisely that the critical learning rate is $\eta = \frac{2\Delta}{K}$. In this section, we formalize this conjecture, and present detailed justifications.

**Conjecture 1.** *For $\Delta > 0$, consider the instance $\nu \in \mathcal{F}_\Delta^K$ defined by $\nu_1 = \delta_0$ and $\nu_2 = \cdots = \nu_K = Rad(-\Delta)$. Then, it holds that*

$$\exists \eta_0 > \frac{2\Delta}{K} \ s.t. \ , \forall \eta \in (0, \eta_0) : \ \mathcal{R}_T \leq \frac{K-1}{K} \cdot \frac{\log(T)}{\eta} + o(\log(T)) \ . \tag{20}$$

*and, as a consequence, SGB can be uniformly efficient on all instances from $\mathcal{F}_\Delta^K$ **only if** $\eta \leq \frac{2\Delta}{K}$.*

The empirical results presented in Section G.3 support this conjecture, and even suggest that the logarithmic bound on the post-convergence term is the same **for all problems** $\nu \in \mathcal{F}_\Delta^K$ (Conjecture 3), although considering only the instance described in the conjecture is sufficient to discuss the critical threshold.

We now formalize how Eq. (20) suffices to establish that the critical learning rate is smaller than $\frac{2\Delta}{K}$, and describe the technical results that would be necessary to prove the conjecture.

*Intuition.* Let us assume that Eq. (20) holds. Then, the deduction that $\eta \leq \frac{2\Delta}{K}$ is necessary for SGB to achieve logarithmic regret follows from the same proof steps as Theorem 3: if $\Delta$ is small enough, then the lower bound of Lai and Robbins [36] (Theorem 6 in Appendix D.1) establishes that the asymptotic regret of SGB cannot be better than $(K - 1) \cdot \frac{\log(T)}{2\Delta}$ if it is uniformly efficient on all instances. By contradiction of the lower bound, this proves the last statement of the conjecture, since some parameters $\eta \in \left[\frac{2\Delta}{K}, \eta_0\right]$, which is non-empty, would violate the lower bound for $\Delta$ small enough.

Hence, to prove the conjecture it suffices to prove Eq. (20), akin to how Lemma 1 supported the proof of Theorem 3. More precisely, from Equation (7) and Lemma 2 we believe that it can be proved that, for an appropriate range of learning rates, the following holds:

$$\mathbb{E}[\theta_{1,T}] \leq \frac{K-1}{K} \cdot \frac{\log(T)}{\eta} + o(\log(T)) \quad \text{and} \quad \mathbb{E}\left[\sum_{t=1}^{T}(1-p_{1,t})^2\right] = o(\log(T)) .$$

We assume the first result (logarithmic scaling) because we have a strong intuition that the post-convergence term corresponding to a fixed instance, with random rewards, should admit the same asymptotic scaling as its deterministic counterpart $(\nu_k)_{k\in[K]} = (\delta_{\mu_k})_{k\in[K]}$, that we studied in Lemma 1 in the case considered in the conjecture. We furthermore include a $\frac{K-1}{K}$ factor to conform our intuition that this is a possible improvement of the upper bound of Lemma 1, as discussed in the main paper and Appendix D.2.

Let us now discuss the upper bound on the failure regret. By introducing $\eta_0 > 0$, we anticipate that (in contrast to Lemma 1), it might be necessary to restrict the learning to a bounded range to prevent large deviations for the parameters $(\theta_{k,t})_{k\geq 2}$. Indeed, since rewards of sub-optimal arms are random, it is possible that $\theta_{k,t}$ becomes arbitrarily large for some $k \geq 2$. However, since this increase would necessarily cause $p_{2,t}$ to be large, we would expect arm 2 to be pulled frequently, and thus $\theta_{2,t}$ to decrease in reasonable time. In simpler terms, the instance chosen to state the conjecture makes it unlikely that the failure regret of SGB is large: arm 1 cannot deliver bad rewards, and any over-performance of sub-optimal arms will lead to fast correction, likely causing relatively small regret, probably upper bounded by a constant (thus $o(\log(T))$).

$\square$

Conjecture 1 offers a direction to derive the critical threshold for $\eta$ that is very similar to the proof of Theorem 3, in which we derived the necessary condition with scaling $K^{-1}$.

Then next conjecture, on the contrary, mirrors the intuition behind the proof of Theorem 2, by proposing a candidate difficult instance for SGB, for which scaling of the learning rate below the critical threshold would be necessary.

**Conjecture 2** (Candidate difficult instance). *Let $K \geq 3$, and $\nu \in \mathcal{F}_\Delta^K$ be the bandit instance defined by $\nu_1 = Rad(\Delta)$, $\nu_2 = \delta_0$, and $\nu_3 = \cdots = \nu_K = \delta_{-\mu}$ for any $\mu \in (0,1)$. Then, $\eta < \frac{2\Delta}{(K-1)(1-\Delta)}$ is necessary to guarantee logarithmic regret on such instance for any number of arms $K$.*

*Intuition.* As discussed in the main paper after stating Theorem 3, the specificity of SGB compared to more standard bandit policies is the fact that adding some very sub-optimal arm is not "neutral" with respect to better-performing arms, which we illustrate by understanding some scenarios that can happen with this choice of instance.

We follow the proof of Theorem 2 to provide a detailed intuition to support the conjecture. We recall that, in the proof of Theorem 2, we identified a scenario under which the regret of SGB is linear, and proved that this scenario happens with a probability that is too large to avoid polynomial regret when $\eta > \Delta + \mathcal{O}(\Delta^2)$. Let us consider a similar scenario.

Phase 0: unlucky start Assume that for its $n_0$ first pulls arm 1 receives a reward $-1$, and that these pulls happen during a (random) number of steps $t_0$. By tuning $n_0$, we can make $\theta_{2,t_0+1} - \theta_{1,t_0+1}$ arbitrarily large, and as a consequence, $p_{2,t_0+1} - p_{1,t_0+1}$ can be made very close to 1. The pulls of arms $\{3, \ldots, K\}$ can only make the gap between the two parameters grow faster, since it is clear that $p_{2,t} \geq p_{1,t}$ throughout this phase.

Phase 1: failed recovery, with a twist: We are then tempted to consider the same scenario as in the proof of Theorem 2, in which in a second phase arm 1 continues to receive rewards that are sufficiently bad to push its probability below $1/(2T)$. By following the same proof scheme, we could prove that the fact that the arms $k \geq 3$ are now very sub-optimal can only make this scenario happen *faster*: indeed, if the rewards in phase 1 are sufficiently bad that the gap $p_{2,t} - p_{1,t} \approx 1$ from the end of phase 0 can be maintained throughout phase 1, then

$$\forall t \geq t_0 + 1, \quad A_t \geq 3 \Longrightarrow \theta_{2,t+1} - \theta_{1,t+1} = \theta_{2,t} - \theta_{1,t} + \eta\mu \cdot \underbrace{(p_{2,t} - p_{1,t})}_{\approx 1} .$$

Hence, the probability of arm 1 decreases not only because of its unlucky pulls, but also each time a "very bad" arm is drawn. This is a new phenomenon compared to the instance studied in the proof of Theorem 2. We believe that this is the reason why the learning rate should depend on $\Delta/K$, as we explain in the following. In the proof of Theorem 2, we showed that after the $n_1$ pulls of arm 1 in phase 1 it held that

$$\theta_{1,t_0+t_1+1} \approx \theta_{1,t_0+1} - \eta\Delta n_1 , \text{ and } \forall k \geq 2 : \theta_{k,t_0+t_1+1} = -\frac{\theta_{1,t_0+t_1+1}}{K-1}$$

which led us to choose $n_1 \approx \frac{K-1}{K} \cdot \frac{\log(2T)}{\eta\Delta}$, using the linear dependency between parameters. In the new instance considered here, we believe that the following holds instead,

$$\theta_{1,t_0+t_1+1} \approx \theta_{1,t_0+1} - \eta\Delta n_1 - \eta\mu(\overline{n}_{t_0+t_1} - \overline{n}_{t_0}) , \text{ with } \overline{n}_t = \sum_{s=1}^{t} \mathbb{1}(A_s \geq 3) .$$

Hence, intuitively we can expect that a lower value of $n_1$ (compared to the theorem) could lead to $p_{1,t_0+t_1+1} \leq \frac{1}{2T}$ with large-enough probability.

The following detailed intuition about this phenomenon is speculative, but we think that from that point we can argue the following: since arm 1 appears "at best" as a distribution with gap $\Delta$, while the others appear as distributions with gap $-\mu < 0$, then it should hold that $p_{1,t}/p_{k,t} \asymp \frac{\mu}{\Delta}$ throughout the duration of phase 1. We formally this in Conjecture 3 in Appendix G.3, as a new conjecture supported by experiments. As a consequence, with non-negligible probability, it should hold that $\overline{n}_{t_0+t_1} - \overline{n}_{t_0} \approx (K-2) \cdot \frac{\Delta}{\mu} \cdot n_1$, by summing over the sample sizes of the $K-2$ bad deterministic arms. Under such scenario, we would then obtain that

$$\theta_{1,t_0+t_1+1} - \theta_{2,t_0+t_1+1} \approx \theta_{1,t_0+1} - \theta_{2,t_0+1} - \eta\Delta(K-1)n_1 ,$$

from which the desired tuning of $n_1$ would become $n_1 \approx \frac{\log(2T)}{(K-1)\eta\Delta}$.

By plugging into the last steps of the proof of Theorem 2, we would obtain that a lower bound on the probability of linear regret is of order $e^{-\frac{2\Delta}{(K-1)\eta(1-\Delta)}\log(2T)}$, which gives a lower bound on the regret of scaling $T^{1-\frac{2\Delta}{\eta(K-1)(1-\Delta)}}$, polynomial if $\eta > \frac{2\Delta}{(K-1)(1-\Delta)}$. We believe that these arguments motivate the conjecture, although a proper formalization would involve many technicalities (akin to the proof of Theorem 2, but with more involved arguments), that we leave for future work. In particular, in order to prove the results presented in this paper we manly had to prove upper bounds of the form $p_{k,t} \lesssim \frac{1}{\eta\Delta_k t}$, while for this new result it would be necessary to further prove that $p_{k,t} \gtrsim \frac{1}{\eta\Delta_k t}$ with some reasonable probability, to use that $\mu(\overline{n}_{t_0+t_1} - \overline{n}_{t_0}) \approx \Delta n_1$ during phase 1 is a likely scenario. $\qquad\square$

We remark that the upper bound on the critical rate suggested by Conjecture 1 is smaller than the upper bound suggested by Conjecture 2, but the two are relatively close when $\Delta$ is small and $K$ is large. Proving either one of the two results presents independent challenges, and successfully proving either would constitute a significant improvement in the theoretical understanding of SGB.

## E   Proof of Theorem 4

We start by restating the theorem, before detailing its proof.

**Theorem 4** (Logarithmic regret for identical gaps). *Let $\nu \in \mathcal{F}^K$ be a MAB instance satisfying $\Delta_2 = \cdots = \Delta_K = \Delta$, for some $\Delta \in (0,1)$. Then, the regret of SGB tuned with a learning rate $\eta$ satisfying $\eta C_\eta < \frac{2\Delta}{K+2}$ is upper bounded as follows,*

$$\mathcal{R}_T^{SGB}(\nu) \leq \frac{2}{\eta} \cdot \mathbb{E}[\theta_{1,T+1}] + \Gamma_\nu \leq \frac{2(K-1)}{\eta} \cdot \log(T) + \Gamma_\nu , \text{ for some constant } \Gamma_\nu > 0.$$

*Proof.* Instead of using Equation (7) directly, we use Proposition 3 (Appendix F.1) to convert it into an alternative expression depending on the probabilities $\sum_{t=1}^{T} \mathbb{P}(p_{1,t} < 1/2)$, that is more adapted to

the arguments presented in the rest of the proof. Combined with Lemma 2, we obtain a first regret upper bound

$$\mathcal{R}_T \leq 2(K-1) \cdot \left( \frac{\log(T)}{\eta} + 4 \right) + 2\Delta \sum_{t=1}^{T} \mathbb{P}\left( p_{1,t} < \frac{1}{2} \right),$$

which gives the logarithmic term of the theorem, and we can remark the constant $\Gamma_\nu$ should be an upper bound of $8(K-1) + \sum_{t=1}^{T} \mathbb{P}(p_{1,t} < 1/2)$. It remains to prove that the tuning of $\eta$ guarantees that the latter sum is finite. To do that, we use a proof inspired by the analysis of SAMBA [29]. Indeed, similarly to the proof of Theorem 1, we are going to prove that $\mathbb{E}[p_t^{-1}]$ is decreasing, under a condition on $\eta$ determined by the initial stage $p_{1,1} = \frac{1}{K}$. Using Lemma 17 with $H_t = 0$, thanks to the identical gaps, we obtain that, for all steps $t \geq 1$,

$$\mathbb{E}_t \left[ \frac{1}{p_{t+1}} \right] \leq \frac{1}{p_t} - \eta(\Delta - \eta C_\eta) \frac{K}{K-1}(1-p_t)^2 + \underbrace{\frac{\eta^2 C_\eta}{2} \cdot \frac{\sum_{k=2}^{K} p_{k,t}^2(1-p_{k,t}) - p_t(1-p_t)^2}{p_t}}_{W_t} .$$

(21)

Comparing with our analysis for two-armed bandits (Th. 1), the term $W_t$ causes additional difficulty. To overcome it, we simply use that $\sum_{k=2}^{K} p_{k,t}^2(1-p_{k,t}) \leq \frac{1-p_t}{4} \leq \frac{1}{4}$, so we can simplify the recursion as follows,

$$\mathbb{E}_t \left[ \frac{1}{p_{t+1}} \right] \leq \frac{1}{p_t} \left( 1 + \frac{\eta^2 C_\eta}{8} \right) - \eta(\Delta - \eta C_\eta) \frac{K}{K-1}(1-p_t)^2,$$

which we can now use to complete the proof of the theorem.

Step 1: reaching $p_t = \frac{1}{2}$ in finite time. we now consider the collection of time steps $\mathcal{T}_0$ starting in $t = 0$, and ending at the stopping time $\tau_0$ corresponding to the first time $p_t \geq \frac{1}{2}$ holds. We use that within $\mathcal{T}_0$, when $p_t < 1/2$, it holds that $\frac{K}{K-1}(1-p_t)^2 \geq \frac{1}{4}$. Thus, for $t \leq \tau_0$ it holds that

$$\mathbb{E}_t \left[ \frac{1}{p_{t+1}} \right] \leq \frac{1}{p_t} \left( 1 + \frac{\eta^2 C_\eta}{8} \right) - \frac{\eta}{4}(\Delta - \eta C_\eta) = \frac{1}{p_t} + \frac{\eta}{8} \cdot \left( \eta C_\eta \left( \frac{1}{p_t} + 2 \right) - 2\Delta \right) . \quad (22)$$

We then use that $p_1^{-1} = K$ to obtain that if $\eta C_\eta < \frac{2\Delta}{K+2}$, then it holds that $\mathbb{E}\left[ \frac{1}{p_2} \right] < \frac{1}{p_1} - c$, for the constant $c = 2\Delta - \eta C_\eta(K+2) > 0$. We deduce that $\mathbb{E}[p_{t \wedge \tau_0}^{-1}] \leq p_1^{-1} - ct$: by induction, if $\mathbb{E}_t[p_t^{-1}] \leq p_1^{-1} = K$ then, after taking expectations, the right hand term of the above equation is smaller than $\eta C_\eta(K+2) - 2\Delta \leq -c$, for some constant $c > 0$. Hence, by using similar proof steps as Proposition 1 of [29], we obtain that $\mathbb{E}[t \wedge \tau_0] \leq \frac{1}{cp_1} = \frac{K}{2\Delta - \eta e^\eta(K+2)}$.

Step 2: subsequent transient phases. We now consider additional transient phases: after reaching $p_{\tau_0} \geq \frac{1}{2}$, there is a probability that $p_{\tau_0 + t} < \frac{1}{2}$ happens again for some $t > 0$. This starts a new transient phase $\mathcal{T}_1$, starting at some step $s_1 > \tau_0$, with initial probability $p_{s_1} \geq \frac{e^{-2\eta}}{2}$, which is due to the step-size and the fact the previous iterate is above the threshold, i.e., $p_{s_1-1} > \frac{1}{2}$. Similarly, we can define potential subsequent transient phases $(\mathcal{T}_j)_{j \in \mathbb{N}}$. For each phase, since the update formula (22) only depends on the probability of arm 1, we obtain with the same arguments as for $\mathcal{T}_0$, that

$$\forall j \geq 1, \ \mathbb{E}[\tau_j - s_j | s_j < +\infty] \leq \frac{2e^{2\eta}}{2\Delta - \eta C_\eta(K+2)} .$$

Step 3: finite expected number of transient phases. For any learning rate $\eta$, there exists a $\rho \in [0,1)$ that, if $p_t \geq 1/2$ holds, then $\inf_{s \geq 1} p_{t+s} \geq 1/2$. In words, after the probability of the best arm reaches $1/2$ there is a positive probability that it remains above $1/2$ forever: $\forall j \geq 1$, $\mathbb{P}(s_{j+1} = +\infty | \tau_j < +\infty) \geq \rho$. By conditional independence of the phases, it is then direct to see that the expected number of transient phases is upper bounded by $\rho^{-1}$, which leads too

$$\sum_{t=1}^{+\infty} \mathbb{P}\left( p_t < \frac{1}{2} \right) \leq \frac{K}{2\Delta - \eta C_\eta(K+2)} + \frac{1}{\rho} \cdot \frac{2e^{2\eta}}{2\Delta - \eta C_\eta(K+2)},$$

which is a problem-dependent constant with explicit dependency in $K, \eta, \Delta, \rho$. We now prove that the –non-explicit– constant $\rho$ is positive.

**Step 4: proving that $\rho > 0$.** We first prove the bound for the first transient stage, which we then extend to an arbitrary stage by the Markov property. Denote $q_t = 1 - p_t$ and let $q_{\tau_0}$ be the first iterate for which we know that $q_{\tau_0} \leq \frac{1}{2}$, i.e., $p_{\tau_0} \geq \frac{1}{2}$. Assume there is a non-zero probability $p' > 0$ that the subsequent iterate $q_{\tau_0+1}$ is strictly bounded away from $\frac{1}{2}$, that is, $q_{\tau_0+1} \geq c < \frac{1}{2}$.

We denote $\sigma_0$ to be the first time when $q_t \geq \frac{1}{2}$ after the time $\tau_0$, that is: $\sigma_0 := \min\{t \geq \tau_0 : q_t \geq \frac{1}{2}\}$. Consider $q_{t \wedge \sigma_0}$ for $t \geq \tau_0$, which, by the developments in (22), is a supermartingale. As a result we can sufficiently upper bound $\mathbb{E}[q_{t \wedge \sigma_0} | q_{\tau_0+1} < c]$ using the Optional Stopping Theorem as

$$
\begin{aligned}
c > \mathbb{E}\left[q_{\tau_0+1}|q_{\tau_0+1} < c\right] &\geq \mathbb{E}\left[q_{t \wedge \sigma_0}|q_{\tau_0+1} < c\right] \\
&= \mathbb{E}\left[q_{\sigma_0}\mathbb{I}[t \geq \sigma_{\tau_0}]|q_{\tau_0+1} < c\right] + \mathbb{E}\left[q_t\mathbb{I}[t < \sigma_0]|q_{\tau_0+1} < c\right] \\
&\geq \frac{1}{2}\mathbb{P}\left(\sigma_0 \leq t|\sigma_{\tau_0+1} < c\right) + \mathbb{E}\left[q_t\mathbb{I}[t < \sigma_0]|q_{\tau_0+1} < c\right] \\
&\xrightarrow{t \to \infty} \frac{1}{2}\mathbb{P}\left(\sigma_0 < \infty|q_{\tau_0+1} < c\right),
\end{aligned}
$$

where in the second line we split the expectation of random trajectories that hit $\sigma_{\tau_0}$ before $t$ and those that do not, in the third line we use the lower bound $q_{\sigma_0} \geq \frac{1}{2}$, and in the final line, we apply the Dominated Convergence Theorem and use that the second term remains a submartingale for all $t$ and converges to $0$ as $t \to \infty$. Therefore we have

$$
\mathbb{P}\left(\sigma_0 = \infty|q_{\tau_0+1} < c\right) > 1 - 2c > 0.
$$

To factor out the assumption that $q_{\tau_0+1} < c$, we get

$$
\begin{aligned}
\mathbb{P}\left(\sigma_0 = \infty|q_{\tau_0+1} \leq \frac{1}{2}\right) &= (1 - p') + p'\mathbb{P}\left(\sigma_0 = \infty|q_{\tau_0+1} < c\right) \\
&\leq 1 - (1 - \delta)p' = p'(1 - 2c) := \rho > 0.
\end{aligned}
$$

The argument can be extended to all stages using the Markov property. By the Markov property we have

$$
\mathbb{P}\left(\sigma_k < \infty|\tau_{k-1} < \infty\right) \geq 1 - \rho > 0.
$$

We can decompose the events as

$$
\{\sigma_{k-1} < \infty\} \cap \{\tau_{k-1} - \sigma_{k-1} < \infty\} = \{\tau_k < \infty\},
$$

and, since $\mathbb{P}(\tau_k - \sigma_{k-1} < \infty|\sigma_{k-1} < \infty) = 1$ by the developments in (22) and the bound on the stepsize, we get

$$
\mathbb{P}\left(\sigma_k < \infty|\sigma_{k-1} < \infty\right) = \mathbb{P}\left(\sigma_k < \infty|\tau_k < \infty\right) \geq 1 - \rho.
$$

This concludes the proof that $\rho > 0$, and thus that the constant $\Gamma_\nu$ of the theorem is finite. $\qquad\square$

We finally state and prove a counterpart of Corollary 1 in the setting of arms with identical gaps, showing that horizon-dependent tuning can also provide a robust alternative to the knowledge of $\Delta$ in this setting.

**Lemma 15** ($\sqrt{T}$ regret with $\eta = 1/\sqrt{T}$). *Assume that $\eta = 1/\sqrt{T}$, and that the bandit instance $\nu \in \mathcal{F}^K$ satisfies $\Delta_2 = \cdots = \Delta_K$ as in Theorem 4. Then the regret of SGB is upper bounded by $\mathcal{R}_T(\nu) \lesssim K\log(T)\sqrt{T}$ on such instance, where only absolute constants are hidden.*

*Proof.* Again, if the gap $\Delta$ is small enough so that $\eta C_\eta \geq \frac{\Delta}{K+2}$ then we can upper bound the regret by $\Delta T$, which yields $\mathcal{R}_T(\nu) \lesssim K\sqrt{T}$. In the alternative case, we can use the bound of the theorem. The logarithmic term gives a bound scaling with $K\log(T)\sqrt{T}$, as desired. This could be sufficient if we aimed for a problem-dependent bound, by leaving the constant $\Gamma_\nu$. However, it is relatively

easy to change the analysis to obtain a problem-independent bound scaling with $\sqrt{T}$. Indeed, we first observe from (21) that $\forall t \in [T], \mathbb{E}\left[\frac{1}{p_t}\right] \leq K \cdot \left(1 + \frac{\eta^2 e^\eta}{8}\right)^T \lesssim K$, hence by Corollary 3 it holds that

$$
\begin{aligned}
\Delta \cdot \mathbb{E}\left[\sum_{t=1}^T (1-p_t)^2\right] &\leq \Delta \cdot \frac{K-1}{\eta(\Delta - \eta C_\eta)} + \frac{\eta^2}{2} \Delta \cdot \sum_{t=1}^T \mathbb{E}\left[\frac{W_t}{p_t}\right] \\
&\leq \Delta \cdot \frac{K-1}{\eta\left(\Delta - \frac{\Delta}{K+2}\right)} + \frac{\eta^2}{2} \sum_{t=1}^T \mathbb{E}\left[\frac{1}{p_t}\right] \\
&\lesssim \frac{K}{\eta} + K \lesssim K\sqrt{T} \,,
\end{aligned}
$$

which finalizes the proof. $\qquad\square$

## F  Additional technical results and proofs

In this section we present the detailed computations of the tools presented in Section 3, for the regret analysis of SGB when there are more than two arms, with additional discussions presenting some intuitions that we did not include in the main paper for space reasons.

### F.1  Alternative regret bound from Equation (7)

In this appendix we present an alternative regret bound that can be derived from Equation (7), and that we use in the proof of Theorem 4.

**Proposition 3** (Generic regret upper bound for SGB). *For **any** learning rate $\eta > 0$ and for any $\varepsilon \in (0,1)$, defining $C_\varepsilon = \frac{\max_{k \in [K]} \Delta_k}{(1-\varepsilon)\Delta}$, SGB satisfies the following regret bound,*

$$
\forall \nu \in \mathcal{F}^K, \quad \mathcal{R}_T^{SGB} \leq C_\varepsilon \cdot \left(\frac{\mathbb{E}[\theta_{1,T}]}{\eta} + \Delta \cdot \sum_{t=1}^T \mathbb{P}\left(p_{1,t} < 1 - \varepsilon\right)\right) \,.
$$

*Proof.* We start from (7), and consider bounding the *failure regret* in the right-hand term. For any $\varepsilon \in (0,1)$, it holds that

$$
\begin{aligned}
\mathbb{E}\left[\sum_{t=1}^T (1-p_{1,t})^2\right] &\leq \mathbb{E}\left[\sum_{t=1}^T (1-p_{1,t})^2 \mathbb{1}(p_{1,t} < 1-\varepsilon)\right] + \mathbb{E}\left[\sum_{t=1}^T (1-p_{1,t})^2 \mathbb{1}(p_{1,t} \geq 1-\varepsilon)\right] \\
&\leq \mathbb{E}\left[\sum_{t=1}^T \mathbb{1}(p_{1,t} < 1-\varepsilon)\right] + \varepsilon \cdot \mathbb{E}\left[\sum_{t=1}^T (1-p_{1,t}) \mathbb{1}(p_{1,t} \geq 1-\varepsilon)\right] \\
&\leq \sum_{t=1}^T \mathbb{P}\left(p_{1,t} < 1-\varepsilon\right) + \varepsilon \cdot \mathbb{E}\left[\sum_{t=1}^T (1-p_{1,t})\right] \,,
\end{aligned}
$$

By recognizing the regret in the last term and putting it together in (7) we get a recursive relation

$$
\forall \nu \in \mathcal{F}^K, \quad \mathcal{R}_T^{SGB} \leq \overline{\mathcal{R}}_T := \frac{\Delta_{\max}}{\eta\Delta} \mathbb{E}[\theta_{1,T}] + \Delta_{\max} \cdot \sum_{t=1}^T \mathbb{P}\left(p_{1,t} < 1-\varepsilon\right) + \varepsilon \cdot \overline{\mathcal{R}}_T \,,
$$

which by reorganizing the terms provides

$$
\mathcal{R}_T^{SGB} \leq \overline{\mathcal{R}}_T \leq \frac{\Delta_{\max}}{(1-\varepsilon)\eta\Delta} \cdot \mathbb{E}[\theta_{1,T}] + \frac{\Delta_{\max}}{1-\varepsilon} \cdot \sum_{t=1}^T \mathbb{P}\left(p_{1,t} < 1-\varepsilon\right) \,.
$$

The proof is completed by factorizing $C_\varepsilon$ out of both terms. $\qquad\square$

**Additional insights from Proposition 3** We recall that, with the same notations, Equation (7) provides

$$\forall \nu \in \mathcal{F}^K, \quad \mathcal{R}_T^{\text{SGB}}(\nu) \leq C_0 \cdot \left( \frac{\mathbb{E}[\theta_{1,T}]}{\eta} + \Delta \cdot \mathbb{E}\left[ \sum_{t=1}^{T} (1 - p_{1,t})^2 \right] \right),$$

with both regret bounds valid for any value of $\eta > 0$, and $\varepsilon \in (0,1)$ in Proposition 3. Hence, proving logarithmic regret for SGB under a specific choice of $\eta$ can be equivalently done by either proving that $\mathbb{E}\left[ \sum_{t=1}^{T} (1 - p_{1,t})^2 \right] = o(\log(T))$, as we did in the proof of Theorem 1, or that

$$\mathbb{E}[T_\varepsilon^\eta] = o(\log(T)), \quad \text{for} \quad T_\varepsilon^\eta := \{t \in [T] : p_{1,t} \leq 1 - \varepsilon\} .$$

In words, $\mathbb{E}[T_\varepsilon^\eta]$ is simply the expected number of time steps for which the probability of the best arm is below the threshold $1 - \varepsilon$, within a trajectory of duration $T$.

Then, we believe that the following result is interesting to interpret the regret of SGB in the polynomial regime. We use the notation $A \asymp B$ to express that both $A \lesssim B$ and $B \lesssim A$ hold.

**Corollary 3.** *Assume that $\mathcal{R}_T^{SGB} = \Omega(T^\alpha)$ for some instance $\nu \in \mathcal{F}^K$ and some $\alpha > 0$. Then, it holds that*

$$\mathcal{R}_T^{SGB} \asymp \mathbb{E}[T_\varepsilon^\eta] \asymp \mathbb{E}\left[ \sum_{t=1}^{T} (1 - p_{1,t})^2 \right] .$$

*Proof.* From Lemma 2, we know that $\mathbb{E}[\theta_{1,T}]$ is always logarithmic, and can thus be ignored when comparing the bounds when the regret is polynomial. Then, the fact that the regret is upper bounded by each term comes from Proposition 3 and Eq. (7). It remains to show that the converse holds. We first directly obtain that

$$\mathcal{R}_T^{\text{SGB}} \geq \Delta \mathbb{E}\left[ \sum_{t=1}^{T} (1 - p_{1,t}) \right] \geq \Delta \mathbb{E}\left[ \sum_{t=1}^{T} (1 - p_{1,t})^2 \right],$$

and then that

$$\mathcal{R}_T^{\text{SGB}}(\nu) \geq \Delta \mathbb{E}\left[ \sum_{t=1}^{T} (1 - p_{1,t}) \right] \geq \Delta \varepsilon \mathbb{E}\left[ \sum_{t=1}^{T} \mathbb{1}(p_{1,t} \leq 1 - \varepsilon) \right],$$

which gives the result since the regret thus admits the same scaling in $T$ as both quantities. $\qquad\square$

We interpret this result as follows: when the regret of SGB is not logarithmic, then it scales with the number of times $1 - p_{1,t}$ is smaller than any fixed threshold. This gives a a way to interpret the term $\mathbb{E}\left[ \sum_{t=1}^{T} (1 - p_{1,t})^2 \right]$ as the *failure regret* introduced in the main paper.

**Non-unique optimal arm** We now show that Proposition 3 and Lemma 2 can be adapted when several arms are optimal, however we kept the assumption that one arm is optimal throughout the paper to simplify the presentation.

If the optimal arm is not unique, we can recover an expression similar to (7) by grouping the $K^\star$ optimal arms together. Indeed, defining $\theta_{\star,t} = \sum_{j=1}^{K} \theta_{j,t} \mathbb{1}(\Delta_j = 0)$ and $p_{\star,t} = \sum_{j=1}^{K} p_{j,t} \mathbb{1}(\Delta_j = 0)$ we have that

$$\mathcal{R}_T \leq \overline{\mathcal{R}}_T := \max_{k \in [K]} \Delta_k \cdot \mathbb{E}\left[ \sum_{t=1}^{T} (1 - p_{\star,t}) \right],$$

and also that

$$\begin{aligned}
\mathbb{E}[\delta\theta_{\star,t}] &= \eta \sum_{j=1}^{K^\star} p_{j,t} \sum_{k=1}^{K} p_{k,t} \Delta_{j,k} \\
&\geq \sum_{j=1}^{K^\star} p_{j,t} (1 - p_{\star,t}) \Delta \\
&= p_{\star,t} (1 - p_{\star,t}) \Delta ,
\end{aligned}$$

so we can adapt Equation (7) by replacing $\theta_{1,t}$ by $\theta_{\star,t}$ and $p_{1,t}$ by $p_{\star,t}$. Then, for the proof of Lemma 2 we only need to consider the event $\mathbb{E}\left[\max_{k>K^\star} \theta_{k,t}\right]$, and remark that under this event and $\theta_{\star,t} \geq \gamma_t$ it holds that

$$1 - p_{\star,t} \leq \frac{(K - K^\star)}{\sum_{j=1}^{K^\star} e^{\theta_{j,t}} + (K - K^\star)} \leq \frac{(K - K^\star)}{\sum_{j=1}^{K^\star} e^{\theta_{j,t}}}$$

$$\leq \frac{K - K^\star}{K^\star} \cdot e^{-\frac{\theta_{\star,t}}{K^\star}} \leq \frac{K - K^\star}{K^\star} \cdot e^{-\frac{\gamma_T}{K^\star}} \ ,$$

so we can adapt the result of Lemma 2 by replacing $\gamma_T = \log(1 + (K - 1)\eta T)$ by $\gamma_T = K^\star \log\left(1 + 2\eta \frac{K-K^\star}{K^\star} T\right)$.

**Remark 3.** *It is actually quite intuitive that the number of optimal arms appear in the regret bound, since their respective parameters will increase slower by splitting the pulls among them. We believe that our lower bound $\sum_{j=1}^{K^\star} e^{\theta_{j,t}} \geq K^\star e^{\frac{\theta_{\star,t}}{K^\star}}$, that generates this multiplicative factor, is tight in most plausible scenario if $T$ is large enough. For instance, if the optimal arms share the same distribution we might the parameters of all optimal arms to be relatively close to each other.*

### F.2 Proof of Lemma 2 and additional discussions

We restate and prove the lemma.

**Lemma 2.** *For any horizon $T \geq 1$ and for **any learning rate** $\eta$, parameter $\theta_{1,T+1}$ of SGB satisfies*

$$\mathbb{E}[\theta_{1,T+1}] \leq \log\left(1 + \left(\eta\Delta + \frac{\eta^2}{2}e^\eta\right)(K - 1) \cdot M_T \cdot T\right) \wedge (K - 1) \cdot (\log(T) + 4\eta) \ ,$$

*Proof.* We propose two independent proofs for the two components of the upper bound.

First bound: we derive this first bound, depending on $M = \sup_{k \in [K]} \sup_{t \geq 1} \mathbb{E}\left[e^{\theta_{k,t}}\right]$, from Jensen inequality. The proof follows the same steps as the proof of Theorem 1. For any $\lambda > 0$, we obtain that

$$\mathbb{E}[\theta_T] = \frac{1}{\lambda}\mathbb{E}\left[\log\left(e^{\lambda\theta_T}\right)\right] \leq \frac{1}{\lambda}\log\left(\mathbb{E}\left[e^{\lambda\theta_T}\right]\right) \ ,$$

We now study $\mathbb{E}\left[e^{\lambda\theta_T}\right]$, using that $e^{\lambda\theta_{t+1}} = e^{\lambda\theta_t + \lambda\delta\theta_t}$. As in the proof of Theorem 1, we use approximations of the exponential to get a recursion formula. More precisely, here we use that for any $q > 0$ and bounded reward $r$ of expectation $\mu$ it holds that

$$\mathbb{E}[e^{qr}] = 1 + q\mu + \sum_{n=2}^{+\infty} \frac{q^n}{n!}\mathbb{E}[r^n] \leq 1 + q\mu + q^2 \cdot \sum_{n=2}^{+\infty} \frac{q^{n-2}}{n!} \leq 1 + q\mu + \frac{q^2}{2} \cdot e^q,$$

that we apply in the following computations with notation $a_t = \lambda\eta(1 - p_t)$ and $b_t = \lambda\eta p_t$,

$$\mathbb{E}_t[e^{\lambda\theta_{t+1}}] = e^{\lambda\theta_t} \cdot \mathbb{E}_t[e^{\lambda\delta\theta_t}] \leq e^{\lambda\theta_t} \cdot \left(p_t\mathbb{E}_t[e^{a_t r_t}|A_t = 1] + (1 - p_t)\mathbb{E}[e^{-b_t r_t}|A_t \neq 1]\right)$$

$$\leq e^{\lambda\theta_t} \cdot \left(1 + p_t \cdot \left(a_t\mu_1 + a_t^2 \cdot e^{a_t}/2\right) + (1 - p_t) \cdot \left(-b_t \max_{k \geq 2}\mu_k + b_t^2 \cdot e^{b_t}/2\right)\right)$$

$$\leq e^{\lambda\theta_t} \cdot \left(1 + p_t(1 - p_t) \cdot \left(\lambda\eta\Delta + (\lambda\eta)^2 e^{\lambda\eta}/2\right)\right) \ ,$$

where in the last line we replaced $a_t, b_t$ by their values and used that $\lambda\eta(p_t \vee 1 - p_t) \leq \lambda\eta$. We can further write that

$$e^{\lambda\theta_t}p_t(1 - p_t) = \frac{e^{(1+\lambda)\theta_t} \cdot \sum_{k=2}^K e^{\theta_{k,t}}}{(e^{\theta_t} + \sum_{k=2}^K e^{\theta_{k,t}})^2} = p_t^2 \cdot e^{(\lambda-1)\theta_{1,t}} \cdot \sum_{k=2}^K e^{\theta_{k,t}} \ .$$

For $K = 2$, choosing $\lambda = 2$ was convenient because it made this expression exactly equal to $p_t^2$. In general, we unfortunately cannot make this simplification. For that reason, we choose $\lambda = 1$ to

eliminate the term depending on $\theta_{1,t}$. Hence, we just obtain that

$$\mathbb{E}[e^{\theta_{1,T}}] \leq 1 + \left(\eta\Delta + \frac{\eta^2}{2}e^\eta\right) \cdot \sum_{t=1}^{T-1} \mathbb{E}\left[\sum_{k=2}^{K} e^{\theta_{k,t}}\right]$$

$$\leq 1 + \left(\eta\Delta + \frac{\eta^2}{2}e^\eta\right) \cdot \sup_{k\geq 2}\sup_{t\geq 1} \mathbb{E}\left[e^{\theta_{k,t}}\right] \cdot (K-1) \cdot T \, ,$$

which leads to the first part of the result of the lemma.

Second bound: the main intuition of the proof is that, when $\theta_{1,T}$ becomes large, it must be at the expense of another parameter, that must be very small. We formalize this with the following intermediate result.

**Lemma 16.** *For any time horizon $T \geq 1$, it holds that*

$$\mathbb{E}\left[\min_{k\in[K]} \theta_{k,T}\right] \geq -\log(T) - 4\eta \, .$$

Before proving this result, we show that it immediately leads to the second bound of Lemma 2: by Equation (9) it holds that $\theta_{1,T} = -\sum_{k=2}^{K} \theta_{k,T} \leq -(K-1)\min_{k\in[K]}\theta_{k,T}$, so

$$\mathbb{E}[\theta_{1,T}] \leq (K-1) \cdot \big(\log(T) + 4\eta\big) \, .$$

We now prove Lemma 16. Fix $\gamma_T := \log(T) + \eta$, and consider $\theta_t^- := -\min_{k\in[K]}\theta_{k,t} \geq 0$. Then,

$$\mathbb{E}[\theta_T^-] \leq \mathbb{E}[\theta_T^- \mathbb{1}(\theta_T^- \leq \gamma_T)] + \mathbb{E}[\theta_T^- \mathbb{1}(\theta_T^- > \gamma_T)]$$

$$\leq \gamma_T \mathbb{P}\left(\theta_T^- \leq \gamma_T\right) + \mathbb{E}[\theta_T^- \mathbb{1}(\theta_T^- > \gamma_T)]$$

We now use the following fact: if $\theta_T^- \geq \gamma_T$, then there exists a time step $u \leq T$ such that

$$\forall s \in \{u, \ldots, T\} : \theta_s^- \geq \gamma_T \text{ and } \theta_{u-1}^- < \gamma_T$$

Using that $\theta_u^- \in [\gamma_T, \gamma_T + \eta]$, we obtain that

$$\mathbb{E}[\theta_T^-] \leq \gamma_T \mathbb{P}\left(\theta_T^- \leq \gamma_T\right) + \mathbb{E}[\theta_u^- \mathbb{1}(\theta_T^- \geq \gamma_T)] + \mathbb{E}[(\theta_T^- - \theta_u^-)\mathbb{1}(\theta_T^- \geq \gamma_T)]$$

$$\leq \gamma_T \mathbb{P}\left(\theta_T^- \leq \gamma_T\right) + \mathbb{E}[(\gamma_T + \eta)\mathbb{1}(\theta_T^- \geq \gamma_T)] + \mathbb{E}[(\theta_T^- - \theta_u^-)\mathbb{1}(\theta_T^- \geq \gamma_T)]$$

$$\leq \gamma_T + \eta + \mathbb{E}[(\theta_T^- - \theta_u^-)\mathbb{1}(\theta_T^- \geq \gamma_T)]$$

$$\leq \gamma_T + \eta + \mathbb{E}\left[\left(\sum_{s=u}^{T-1} \delta\theta_s^-\right)\mathbb{1}(\forall s \geq u : \theta_s^- \geq \gamma_T)\right] \, .$$

From this step, we carefully handle the randomness of $u$ and its dependence with the event $E_u = \{\forall s \geq u : \theta_s^- \geq \gamma_T\}$ in the remaining expectation as follows,

$$\mathbb{E}\left[\left(\sum_{s=u}^{T-1} \delta\theta_s^-\right)\mathbb{1}(E_u)\right] \leq \mathbb{E}\left[\left(\sum_{s=u}^{T-1} (\delta\theta_s^-)_+\right)\mathbb{1}(E_u)\right]$$

$$\leq \mathbb{E}\left[\sum_{s=u}^{T-1} (\delta\theta_s^-)_+ \mathbb{1}(\theta_{s+1}^- \geq \gamma_T)\right]$$

$$\leq \mathbb{E}\left[\sum_{s=1}^{T-1} (\delta\theta_s^-)_+ \mathbb{1}(\theta_{s+1}^- \geq \gamma_T)\right]$$

$$\leq \mathbb{E}\left[\sum_{s=1}^{T-1} \mathbb{E}_s[(\delta\theta_s^-)_+]\mathbb{1}(\theta_{s+1}^- \geq \gamma_T)\right] \, .$$

Then, for any round $s \in [T]$, let $m_s = \operatorname{argmin}_{k\in[K]}\theta_{k,t}$ denote the index of the arm corresponding to the minimum parameter. Then, it either holds that $m_s = m_{s+1}$, and so $\delta\theta_s^- = -\delta\theta_{m_{s+1},s}$, or

that $m_s \neq m_{s+1}$, in which case $\theta_{m_s,s} \leq \theta_{m_{s+1},s}$ so $\delta\theta_s^- \leq -\delta\theta_{m_{s+1},s}$. Hence, in both cases we have that $(\delta\theta_s^-)_+ \leq |\delta\theta_{m_{s+1},s+1}|$. To conclude, it remains to remark that, by definition of $m_{s+1}$ and the fact that $\theta_{s+1}^{-1} \geq \gamma_T$, it must hold that $\theta_{m_{s+1},s} \leq -\gamma_T + \eta$, so $p_{m_{s+1},s} \leq e^{\theta_{m_{s+1},s}} \leq e^{-\gamma_T+\eta}$. Hence, we can finally obtain that

$$\mathbb{E}\left[\left(\sum_{s=u}^{T-1}\delta\theta_s^-\right)\mathbb{1}(E_u)\right] \leq \eta\mathbb{E}\left[\sum_{s=0}^{T-1} 2p_{m_{s+1},s}(1-p_{m_{s+1},s})\mathbb{1}(\theta_{m_{s+1},s+1} \geq \gamma_T)\right]$$

$$\leq 2\eta T e^{-\gamma_T+\eta} \leq 2\eta.$$

$\square$

## F.3    Regret upper bound for two arms with time-varying rate

We develop in this section the counterpart of Corollary 1 with a time-varying rate $\eta_t = \sqrt{\frac{\log(t)}{t}}$.

**Proposition 4.** *When $\Delta$ is unknown, the time-varying learning rate $\eta_t = \sqrt{\frac{\log(e\vee t)}{t}}$ yields a regret bound $\mathcal{R}_T \lesssim \log(T)^{3/2}\sqrt{T}$ for any two-armed bandit $\nu \in \mathcal{F}^2$, where the hidden constants are absolute.*

*Proof.* The first part of the analysis consists in adapting Eq. (7) with time varying learning rates. We remark that the sequence $(\eta_t)_{t\geq 1}$ is non-increasing, hence we can write that

$$\mathbb{E}[\theta_T] = \mathbb{E}\left[\sum_{t=1}^{T}\eta_t \cdot p_t(1-p_t)\right]\Delta \geq \Delta\eta_T \cdot \mathbb{E}\left[\sum_{t=1}^{T}p_t(1-p_t)\right],$$

so we can reuse Eq. (7) by replacing $\eta$ by $\eta_T$ in the first term, which first yields

$$\mathcal{R}_T \leq \frac{\mathbb{E}[\theta_T]}{\eta_T} + \Delta \cdot \mathbb{E}\left[\sum_{t=1}^{T}(1-p_t)^2\right].$$

Then, by following the same steps as the proof of Theorem 1, we obtain that

$$\forall t \geq 1,\ \mathbb{E}_t[e^{2\theta_{t+1}}] \leq e^{2\theta_t} + p_t^2 \cdot 2\left(\eta_t\Delta + \eta_t^2 C_{\eta_t}\right) \implies \mathbb{E}\left[e^{2\theta_T}\right] \leq 1 + 2\sum_{t=1}^{T-1}\left(\eta_t\Delta + \eta_t^2 C_\eta\right) \lesssim T,$$

since the sequence $(\eta_t)_{t\geq 1}$ can be upper bounded by constants. We further note that a more precise (e.g. $\sqrt{T}$) bound would not change the result because it is used inside a logarithm. We now consider the failure regret term. We first remark that, since the learning rate converges to 0, there exists a deterministic time $t_0$ satisfying $\sup_{t\geq t_0}\eta_t C_{\eta_t} \leq \frac{\Delta}{2}$. More precisely, it holds that $t_0 \lesssim \frac{\log(\Delta^{-1})}{\Delta^2}$. The last part of the proof consists in using the proof steps of Theorem 1 for $t \geq t_0$, and obtain that

$$\Delta \cdot \mathbb{E}\left[\sum_{t=1}^{T}(1-p_t)^2\right] \leq \Delta t_0 + \frac{\Delta \cdot \mathbb{E}[x_{t_0}]}{2\eta_T(\Delta - \sup_{t\geq t_0}\eta_t C_{\eta_t})} \lesssim \frac{\log(\Delta^{-1})}{\Delta} + \frac{\mathbb{E}[x_{t_0}]}{\eta_T}.$$

It remains to upper bound $\mathbb{E}[x_{t_0}]$. We recall from the proof of the theorem that the following relation holds, for **any** learning rate ,

$$\forall t \geq 1: \ \mathbb{E}_t[x_{t+1}] \leq x_t - 2\eta_t(1-p_t)^2 \cdot (\Delta - \eta_t C_{\eta_t}),$$

which holds for any arbitrary sequence of learning rate since the proof considers each time step separately. Hence, even in the worst case where $\eta_t$ is much larger than $\Delta$, we can use that

$$\mathbb{E}_t[x_{t+1}] - x_t \lesssim \eta_t^2, \quad \text{so} \quad \mathbb{E}[x_{t_0}] \lesssim \sum_{t=1}^{t_0}\eta_t^2 \leq \log(t_0) \cdot \sum_{t=1}^{t_0}\frac{1}{t_0} \lesssim \log(t_0)^2.$$

We then deduce from the fact that $\mathbb{E}_t[x_{t+1}] - x_t \lesssim \eta_t^2$ that $\mathbb{E}[x_{t+1}] \lesssim \log(t_0)$, which finally gives a regret bound of order

$$\mathcal{R}_T \lesssim \left( \frac{\log(T)}{\eta_T} + \frac{\log(\Delta^{-1})}{\Delta} + \frac{\log(\Delta^{-1})^2}{\eta_T} \right) \wedge \Delta T$$

$$\lesssim \sqrt{T \log(T)} + \log(T)^{3/2}\sqrt{T} + \frac{\log(\Delta^{-1})}{\Delta} \wedge \Delta T,$$

and yields the result by remarking that the gap-dependent term cannot itself be larger than $\sqrt{T \log(T)}$, so the dominant term of the bound is $\log(T)^{3/2}\sqrt{T}$, that comes from the term involving $\mathbb{E}[x_{t_0}]$ in the regret decomposition. $\qquad\square$

**Remark 4** (Improved logarithms for $\eta_t = 1/\sqrt{t}$). *We remark that the above regret bound can be slightly optimized by considering $\eta_t = 1/\sqrt{t}$, that gives*

$$\frac{\log(T)}{\eta_T} + \frac{\sum_{t=1}^{t_0} \eta_t^2}{\eta_T} \lesssim \log(T) \cdot \sqrt{T},$$

*which becomes the scaling of the upper bound. However, we chose to study more precisely $\eta_t = \sqrt{\frac{\log(t \vee e)}{t}}$ in this section and Appendix G.1 because it directly compares to the optimal horizon-dependent rate of Corollary 1.*

### F.4 Proof of Lemma 3

The proof of Lemma 3 is a direct consequence of the following technical result, after using the same telescopic arguments as in the proof of Theorem 1, and that $\sum_{k=2}^{K} p_{k,t}^2 \geq \frac{(1-p_{1,t})^2}{K-1}$.

**Lemma 17.** *Define $C_\eta$ as in Theorem 1. Then, for any $\nu \in \mathcal{F}^K$ and $t \geq 1$, it holds that*

$$\mathbb{E}_t\left[ \frac{1}{p_{1,t+1}} \right] \leq \frac{1}{p_{1,t}} - \eta G_t \cdot (\Delta - \eta C_\eta) + \frac{\eta H_t + \frac{\eta^2}{2} W_t}{p_{1,t}},$$

$$\text{with} \quad \begin{cases} G_t = (1 - p_{1,t})^2 + \sum_{k=2}^{K} p_{k,t}^2 \\ H_t = \sum_{k=2}^{K} p_{k,t}^2 \left( \mathbb{E}_t[\Delta_{A_t}] - (1 - p_{1,t})\Delta_k \right) \\ W_t = \sum_{k=2}^{K} p_{k,t}^2 (1 - p_{k,t}) - p_{1,t}(1 - p_{1,t})^2 \end{cases}.$$

*In addition, $H_t = 0$ if $\Delta_2 = \cdots = \Delta_K = \Delta$, and $W_t = 0$ if $K = 2$.*

*Proof.* We define $x_t = \frac{1}{p_{1,t}}$, and use (13) to obtain the following recursion,

$$x_{t+1} = x_t \cdot \sum_{k=1}^{K} p_{k,t} \cdot e^{\eta \delta\theta_{k,t} - \eta \delta\theta_{1,t}}.$$

By definition of $C_\eta$, for $x \in [-2, 2]$ it holds that $e^{\eta x} \leq 1 + \eta x + C_\eta \cdot \frac{\eta^2 x^2}{2}$. By linearity of the expectation, we deduce that

$$\mathbb{E}_t[x_{t+1}] \leq x_t \cdot \left( 1 + \eta \sum_{k=1}^{K} p_{k,t} \cdot \mathbb{E}_t\left[ \delta\theta_{k,t} - \delta\theta_{1,t} \right] + \frac{\eta^2 C_\eta}{2} \sum_{k=1}^{K} p_{k,t} \cdot \mathbb{E}_t\left[ (\delta\theta_{k,t} - \delta\theta_{1,t})^2 \right] \right).$$

We investigate the first-order term of the approximation. Using (5), and that $\delta\theta_{1,t} = -\sum_{k=2}^{K} \delta\theta_{k,t}$ (Eq. (9)) we obtain that

$$\sum_{k=2}^{K} p_{k,t} \cdot \mathbb{E}_t\left[ \delta\theta_{k,t} - \delta\theta_{1,t} \right] = -\sum_{k=2}^{K} (1 - p_{1,t} + p_{k,t}) \cdot p_{k,t} \cdot \sum_{j \neq k} p_{j,t}(\Delta_k - \Delta_j) := -E_t. \quad (23)$$

We now further analyze the newly defined term $E_t$, that we express as a function of $\mathbb{E}_t[\Delta_{A_t}] = \sum_{j=1}^{K} p_j \Delta_j$. We can naturally decompose $E_t$ into a positive and a negative term. First,

$$\sum_{k=2}^{K}(1 - p_{1,t} + p_{k,t}) \cdot p_{k,t} \cdot \sum_{j=1}^{K} p_{j,t}\Delta_j = \left((1 - p_{1,t})^2 + \sum_{k=2}^{K} p_{k,t}^2\right)\mathbb{E}_t[\Delta_{A_t}] = G_t \cdot \mathbb{E}_t[\Delta_{A_t}] ,$$

with $G_t$ as defined in the statement of the lemma. We then obtain that

$$\sum_{k=2}^{K}(1 - p_{1,t} + p_{k,t}) \cdot p_{k,t} \cdot \sum_{j=1}^{K} p_{j,t}\Delta_k = \sum_{k=2}^{K}(1 - p_{1,t} + p_{k,t}) \cdot p_{k,t} \cdot \Delta_k$$

$$= (1 - p_{1,t})\mathbb{E}_t[\Delta_{A_t}] + \sum_{k=2}^{K} p_{k,t}^2 \Delta_k .$$

So, by subtracting the two we obtain that

$$E_t = (1 - p_{1,t}) \cdot p_{1,t} \cdot \mathbb{E}_t[\Delta_{A_t}] + \sum_{k=2}^{K} p_{k,t}^2 \left(\Delta_k - \mathbb{E}_t[\Delta_{A_t}]\right) \tag{24}$$

$$= (1 - p_{1,t}) \cdot p_{1,t} \cdot \mathbb{E}_t[\Delta_{A_t}] + p_{1,t} \cdot \sum_{k=2}^{K} p_{k,t}^2 \Delta_k + \underbrace{\sum_{k=2}^{K} p_{k,t}^2 \left((1 - p_{1,t})\Delta_k - \mathbb{E}_t[\Delta_{A_t}]\right)}_{-H_t} \tag{25}$$

$$\geq p_{1,t} \cdot \Delta \cdot \underbrace{\left((1 - p_{1,t})^2 + \sum_{k=2}^{K} p_{k,t}^2\right)}_{G_t} - H_t . \tag{26}$$

This writing allows to obtain that $H_t = 0$ if the gaps are identical, because in that case it holds that $\mathbb{E}_t[\Delta_{A_t}] = (1 - p_{1,t})\Delta$.

Let us now consider the term

$$V_t := \sum_{k=1}^{K} p_{k,t} \cdot \mathbb{E}_t\left[(\delta\theta_{k,t} - \delta\theta_{1,t})^2\right] .$$

To analyze the value of the term inside the expectation, it suffices to consider three cases:

$$(\delta\theta_{k,t} - \delta\theta_{1,t})^2 = \begin{cases} (1 - p_{k,t} + p_{1,t})^2 \mathbb{E}[r_t^2 | A_t = k] & \text{if } A_t = k , \\ (1 - p_{1,t} + p_{k,t})^2 \mathbb{E}[r_t^2 | A_t = 1] & \text{if } A_t = 1 , \\ (p_{1,t} - p_{k,t})^2 \mathbb{E}[r_t^2 | A_t \neq 1, k] & \text{otherwise.} \end{cases}$$

$$\forall k \in \{2, \ldots, K\} : \quad \mathbb{E}_t\left[(\delta\theta_{k,t} - \delta\theta_{1,t})^2\right] \leq p_{k,t} \cdot (1 - p_{k,t} + p_{1,t})^2 + p_{1,t} \cdot (1 - p_{1,t} + p_{k,t})^2$$
$$+ (1 - p_{1,t} - p_{k,t}) \cdot (p_{1,t} - p_{k,t})^2 .$$

Note that this expression is exact for Rademacher distributions, for which $r_t^2 = 1$ always holds. Let us use the notation $z = p_{1,t} - p_{k,t}$ to simplify some terms, and again drop the $t$ subscript to simplify the notation. Replacing the terms, we get

$$\mathbb{E}\left[(\delta\theta_k - \delta\theta_1)^2\right] \leq p_k \cdot (1 + z)^2 + p_1 \cdot (1 - z)^2 + (1 - p_1 - p_k) \cdot z^2$$
$$= p_k \cdot (1 + 2z + z^2) + p_1 \cdot (1 - 2z + z^2) + z^2 - (p_1 + p_k) \cdot z^2$$
$$= p_k \cdot (1 + 2z) + p_1 \cdot (1 - 2z) + z^2$$
$$= p_k + p_1 - 2z^2 + z^2$$
$$= p_k + p_1 - z^2 .$$

From this step we can continue by plugging back the value of $z$, and finally obtain that

$$\sum_{k=2}^{K} p_k \cdot \mathbb{E}\left[(\delta\theta_k - \delta\theta_1)^2\right] \leq \sum_{k=2}^{K} p_k \cdot \left(p_k + p_1 - (p_1 - p_k)^2\right) \tag{27}$$

$$= \sum_{k=2}^{K} p_k \cdot \left(p_k(1 - p_k) + p_1(1 - p_1) + 2p_1 p_k\right) \tag{28}$$

$$= \sum_{k=2}^{K} p_k^2(1 - p_k) + p_1 \cdot (1 - p_1)^2 + 2p_1 \sum_{k=2}^{K} p_k^2 \tag{29}$$

$$= 2p_1 \cdot \left((1 - p_1)^2 + \sum_{k=2}^{K} p_k^2\right) + \underbrace{\sum_{k=2}^{K} p_k^2(1 - p_k) - p_1 \cdot (1 - p_1)^2}_{W} , \tag{30}$$

where in the last line we made appear a term that can be expressed similarly to $E_t$ in the above computations. Finally, we get,

$$\eta^2 V_t \leq 2\eta^2 p_{1,t} \cdot \left((1 - p_{1,t})^2 + \sum_{k=2}^{K} p_{k,t}^2\right) + \eta^2 W_t = 2\eta^2 p_{1,t} G_t + \eta^2 W_t,$$

with $W_t = \sum_{k=2}^{K} p_{k,t}^2(1 - p_{k,t}) - p_{1,t} \cdot (1 - p_{1,t})^2$. This concludes the proof, since it is immediate that $W_t = 0$ for $K = 2$. $\qquad\square$

### F.4.1   Properties of the components of Lemma 17

In this section, we investigate structural properties of the terms $H_t$ and $W_t$, which appear in Lemma 17 in the setting $K > 2$. These terms complicate the analysis of the failure regret compared to the simpler case $K = 2$. We believe that understanding their behavior is a crucial step toward extending the theoretical analysis of SGB to multi-armed settings.

In particular, we show that $H_t$ and $W_t$ may exhibit non-trivial behavior even when $K = 3$, suggesting that bounding them uniformly can be challenging. Nevertheless, we also identify several natural scenarios where these terms are either small or negative, which may help control their contribution in a refined analysis.

Property 1: $H_t$ can be non-negative

Consider a setting with $K = 3$ arms where $\Delta_3 > \Delta_2$. Then, for any round $t$, dropping the subscript in the sampling probabilities for notational simplicity, we obtain:

$$H_t = p_2^2(p_2\Delta_2 + p_3\Delta_3 - (p_2 + p_3)\Delta_2) + p_3^2(p_2\Delta_2 + p_3\Delta_3 - (p_2 + p_3)\Delta_3)$$
$$= p_2^2 p_3(\Delta_3 - \Delta_2) + p_3^2 p_2(\Delta_2 - \Delta_3)$$
$$= p_2 p_3(\Delta_3 - \Delta_2)(p_2 - p_3) ,$$

which is non-negative if and only if $p_2 \geq p_3$.

In the regret bound of Lemma 3, the above result converts into a term

$$S_T^H := \sum_{t=1}^{T} \mathbb{E}[H_t'] := \sum_{t=1}^{T} \mathbb{E}\left[\frac{p_{2,t} p_{3,t}}{p_{1,t}}(p_{2,t} - p_{3,t})\right] \cdot (\Delta_3 - \Delta_2) .$$

There are several intuitive reasons to believe that this term can be controlled under reasonable tuning of the learning rate. First, we interpret the fact that $p_{2,t} \leq p_{3,t}$ yields $H_t' < 0$ as follows: if the current state of SGB is "very wrong", in the sense that arm 3 (the worst arm) is currently treated as the best arm by the policy, some self-correction will occur, i.e. this state is unstable. This effect is even amplified if $p_{1,t}$ is very small. Moreover, for the case where $p_{2,t} > p_{3,t}$, then $H_t'$ is positive but scales with $p_{2,t}^2 \cdot p_{3,t}/p_{1,t}$. Even in scenarios for which $p_{2,t} \approx 1$ we might nonetheless expect $p_{3,t}$ to have decreased "faster" than $p_{1,t}$, especially if $\Delta_3 - \Delta_2$ is significant. In any case, a scenario where

*both* $p_{2,t}$ and $p_{3,t}$ are significantly larger than $p_{1,t}$ appears quite unlikely, and we thus believe that $H'_t$ should be (at least) bounded with high probability. This is also supported by the observation that $H_t = 0$ at initialization (where $p_2 = p_3 = 1/3$).

Property 2: Generalization for $K$ arms

For $K$ arms, assuming $\Delta_2 \leq \cdots \leq \Delta_K$, we obtain:

$$
\begin{aligned}
H_t &= \sum_{k=2}^{K} p_k^2 \left( \sum_{j=2}^{K} p_j(\Delta_j - \Delta_k) \right) \\
&= \sum_{k=2}^{K} p_k^2 \left( \sum_{j>k} p_j(\Delta_j - \Delta_k) - \sum_{j=2}^{k} p_j(\Delta_k - \Delta_j) \right) \\
&= \sum_{k=2}^{K} p_k^2 \sum_{j>k} p_j(\Delta_j - \Delta_k) - \sum_{j=2}^{K} \sum_{k>j} p_k^2 p_j(\Delta_k - \Delta_j) \\
&= \sum_{k=2}^{K} p_k^2 \sum_{j>k} p_j(\Delta_j - \Delta_k) - \sum_{k=2}^{K} \sum_{j>k} p_j^2 p_k(\Delta_j - \Delta_k) \\
&= \sum_{k=2}^{K} \sum_{j>k} (\Delta_j - \Delta_k) p_k p_j (p_k - p_j) \,,
\end{aligned}
$$

which we believe might be useful for future analysis of SGB starting from Lemma 3.

Property 3: $W_t$ can be non-negative

We again drop the subscript $t$ for the probabilities, and consider $W_t = \sum_{k=2}^{K} p_k^2(1-p_k) - p_1(1-p_1)^2$. Let us consider again $K = 3$, and use the notation $\widetilde{W}_t = \sum_{k=2}^{K} p_k^2(1 - p_k)$. Then, using that $p_3 = 1 - p_1 - p_2$ we re-write $\widetilde{W}_t$ as follows,

$$
\begin{aligned}
\widetilde{W}_t &= p_2^2(1 - p_2) + p_3^2(1 - p_3) = p_2^2(1 - p_2) + (1 - p_1 - p_2)^2(p_1 + p_2) \\
&= p_1(1 - p_1)^2 + p_2^2(1 - p_2) + p_2(1 - p_1)^2 - 2p_2 p_1(1 - p_1) - 2p_2^2(1 - p_1) + p_2^2 p_1 + p_2^3 \\
&= p_1(1 - p_1)^2 + p_2^2 + p_2(1 - p_1)(1 - 3p_1) - 2p_2^2 + 3p_2^2 p_1 \\
&= p_1(1 - p_1)^2 - p_2^2(1 - 3p_1) + p_2(1 - p_1)(1 - 3p_1) \\
&= p_1(1 - p_1)^2 + p_2(1 - 3p_1)(1 - p_2 - p_1) \,.
\end{aligned}
$$

Hence, we obtain the simple property

$$
K = 3 \implies W_t = p_{2,t} p_{3,t}(1 - 3p_{1,t}), \quad \text{so} \quad W_t \geq 0 \Leftrightarrow p_{1,t} \leq 1/3.
$$

Furthermore, it is easy to see that for $p_1 < \frac{1}{3}$, the maximum of $W_t$ is achieved when $p_2 = p_3 = \frac{1-p_1}{2}$, and so:

$$
W_t \leq (1 - p_{1,t})^2 \cdot \frac{1 - 3p_{1,t}}{4} \,.
$$

Note that this allocation is always feasible for $K > 3$, which provides at least a lower bound on the maximum possible value of $W_t$ in general.

Property 4: $p_{1,t} \geq \frac{1}{3} \implies W_t \leq 0$

We proved this property for $K = 3$, in fact showing that the relation is an equivalence in that case, and now extend it for $K > 3$. Let us assume that $p_{1,t} \geq \frac{1}{3}$. Then, under the constraint $\sum_{k=2}^{K} p_k = 1 - p_1$, the function $p_k^2(1 - p_k)$ is increasing over all admissible values of $p_k$. By symmetry, we deduce that the maximizer of $W$ should be achieved when $m$ of the suboptimal probabilities are equal to $\frac{1-p_1}{m}$ and the remaining ones are zero, for some $m \in [K - 1]$. For a fixed $m$, this yields:

$$
W = \frac{(1 - p_1)^2}{m^2} \left( 1 - \frac{1 - p_1}{m} \right) - p_1(1 - p_1)^2 \,.
$$

If $m$ were continuous, the maximizer would be $m = \frac{3}{2}(1 - p_1)$. Since $m$ must be discrete, it is either $m = 0$ or $m = 1$. Both cases yield $W = 0$.

This property shows that once SGB correctly identifies arm 1 as the best arm and assigns it a sufficiently large probability, the term $W_t$ ceases to be problematic and no longer contributes adversely to the dynamics of Lemma 17.

Property 5: $W_t = 0$ if the sampling probabilities are uniform

We conclude this section by showing that $p_{1,t} \geq \frac{1}{3}$ is sufficient but not necessary for $W_t$ to be small. In particular, the uniform probability vector $p = \left(\frac{1}{K}, \ldots, \frac{1}{K}\right)$ also yields

$$W_t = (K - 1)p_1^2(1 - p_1) - p_1(1 - p_1)^2 = p_1(1 - p_1)(Kp_1 - 1) = 0 \ .$$

This indicates that $W_t = 0$ at the uniform initialization, and thus should remain small in the very first steps of the algorithm. This observation supports the idea that the early-phase behavior of SGB is not negatively impacted by $W_t$. In fact, we even conjecture that, under properly small tuning of the learning rate $\eta$, it might be possible to prove that $\mathbb{E}\left[\frac{W_t}{p_{1,t}}\right] \leq 0$ for all steps $t \geq 1$.

### F.5 Regret upper bounds for two arms under alternative assumptions on rewards

In this section we discuss how the regret upper bound of Theorem 1, and in particular the necessary condition on $\eta$, can be adapted for unbounded rewards. Indeed, the boundedness of rewards is only used to derive Equation (8), where we use a second order approximation of the moment-generating function of rewards. By considering alternative assumptions, we can modify this equation and propagate it in the rest of the proof.

**Sub-Gaussian distributions** Consider the classical example of a $\sigma^2$-sub-Gaussian distribution $\nu$ of expectation $\mu$. By definition, it satisfies the following condition,

$$\forall q \in \mathbb{R}, \ \mathbb{E}_{r \sim \nu}\left[e^{qr}\right] \leq e^{q\mu + \frac{q^2\sigma^2}{2}} \ .$$

To adapt our results, we can thus perform a second-order approximation on this inequality, and not directly on $e^{qr}$. Assume for instance that $|\mu + \frac{\sigma^2}{2}| \leq 1$, then we can write that

$$\forall q \in [-1, 1], \ \mathbb{E}_{r \sim \nu}\left[e^{qr}\right] \leq e^{q\mu + \frac{q^2\sigma^2}{2}} \leq 1 + q\mu + \frac{q^2\sigma^2}{2} + \left(q\mu + \frac{q^2\sigma^2}{2}\right)^2 + \mathcal{O}(q^3)$$

$$= 1 + q\mu + q^2\left(\mu^2 + \frac{\sigma^2}{2}\right) + \mathcal{O}(q^3) \ .$$

In the rest of the example, we neglect the $\mathcal{O}(q^3)$ term for simplicity of the computations, and present how the problem quantities appear in the results with this simplification. To use the above result, we further assume that $2\eta \leq 1$. Then, it is easy to verify that we can follow the proof steps of the theorem, and essentially replace $C_\eta$ (which is $\approx 1$ in the theorem) by $\mu_1^2 + \frac{\sigma^2}{2}$. This yields

$$\mathbb{E}[\theta_{T+1}] \leq \frac{1}{2}\log\left(1 + 2\left(\eta\Delta + \eta^2\left(\mu_1^2 + \frac{\sigma^2}{2}\right)\right)T\right)$$

for the logarithmic term, and the condition on $\eta$ to guarantee that $\mathbb{E}[(1 - p_t)^2] = \mathcal{O}(1)$ becomes

$$\eta \leq \frac{2\Delta}{\sigma^2 + 2\max_{k=1,\ldots,K} \mu_k^2} \ .$$

This result shows that SGB can work with unbounded sub-Gaussian rewards, if properly tuned according to some additional knowledge on the variance proxy and the range of the reward expectations.

**Bernstein condition** This assumption is also standard (see Chapter 2.8 of [33]). Assume that all reward distributions satisfy the following moment conditions,

$$\exists B \geq 0: \quad \forall k \in [K], \ \forall m \geq 2, \ \mathbb{E}_{r \sim \nu}[|r|^m] \leq \frac{1}{2}m!B^{m-2}\mathbb{E}[r^2] \ .$$

This condition is quite handy for the analysis of SGB, since we can now adapt Eq. (8) as follows,

$$\forall q \in \mathbb{R}: \quad \mathbb{E}_{r\sim\nu}[e^{qr}] = 1 + q\mu + \sum_{m=2}^{+\infty} \frac{q^m}{m!} \mathbb{E}[r^m]$$

$$\leq 1 + q\mu + \frac{q^2\mathbb{E}[r^2]}{2} \sum_{m=2}^{+\infty} (qB)^{m-2}$$

$$\leq 1 + q\mu + \frac{q^2\mathbb{E}[r^2]}{2} \cdot \frac{1}{1-qB},$$

if we further assume that $qB < 1$. From this result, if we ignore the $\frac{1}{1-qB}$ term for simplicity (that is, $\eta B \ll 1$) we can then follow the proof steps of Theorem 1 and obtain a sufficient condition $\eta < \frac{\Delta}{s^2}$ for logarithmic regret, with $s^2 = \max_{k=1,\dots,K} \mathbb{E}_{r\sim\nu_k}[r^2]$.

# G   Supplementary experiments

In this section, we present additional experiments that support the theoretical findings of the paper and complement the empirical results from Section 4. We provide the code in the supplementary material accompanying the paper for reproducibility[3].

For simplicity, all experiments use Rademacher distributions unless stated otherwise. These distributions are particularly well-suited for our purposes, as they maximize variance for a given mean under the constraint that rewards lie in $[-1, 1]$. Intuitively, they represent the most "challenging" case within this setting.

Before presenting our results, we clarify a key terminological choice: throughout the paper, the term regret refers to the quantity defined *in expectation*, as in Eq. (1). In this section, we report empirical estimates of this quantity based on multiple runs of the algorithm. Given $M$ independent runs under a given policy, the *empirical regret* is computed as

$$\widehat{\mathcal{R}}_T^M = \frac{1}{M} \sum_{m=1}^M R_T^m := \frac{1}{M} \sum_{m=1}^M \sum_{k=2}^K \Delta_k N_k^m(T) \,, \tag{31}$$

where $N_k^m(T)$ denotes the number of pulls of arm $k \in [K]$ up to horizon $T$ for the $m$-th run of the policy.

For any $M \in \mathbb{N}$, it directly follows that $\mathcal{R}_T = \mathbb{E}\left[\widehat{R}_T^M\right]$. In this section, as well as in Section 4 in the main paper, the term regret (or *average regret*) refers to this empirical estimate unless otherwise stated. Finally, references to the "regret" of specific trajectories refer to the values of $(R_T^m)_{m\in\mathcal{M}}$ for a subset of runs indexed by $\mathcal{M}$, while references to the *distribution of regret* pertain to the empirical distribution of the values $(R_T^m)_{m\in[M]}$.

## G.1   Time-varying learning rate

We start by illustrating the results presented in Corollary 1 and Appendix F.3, establishing the theoretical guarantees of SGB with time-dependent learning rate. In each setup, we run SGB tuned with:

- the oracle gap-dependent tuning $\eta = \frac{2\Delta}{K}$,

- a horizon-dependent gap-free tuning $\eta = \sqrt{\frac{K\log(T)}{T}}$, which requires to know $T$ in advance,

- an anytime tuning $\eta = \rho \wedge \sqrt{\frac{K\log(t)}{t}}$, for some clipping parameter $\rho$ that we discuss.

We consider horizon $T = 10^4$, for which it holds that $\sqrt{\frac{\log(T)}{T}} = 0.03$.

---

[3]will be turned into a Github link after publication.

**Two-armed bandits** We evaluate two configurations, both featuring a sub-optimal arm with mean 0. In the first setup, we set $\Delta = 0.05$ to explore a regime where the initial anytime learning rate significantly exceeds the oracle tuning. In contrast, the second setup uses $\Delta = 0.5$ to examine the opposite scenario, where the learning rate decays quickly relative to the gap. This allows us to highlight the algorithm's sensitivity to the relationship between the decaying learning rate and the problem difficulty.

The results, presented in Figure 3, lead to two key observations. First, the horizon-dependent tuning yields sublinear regret in both setups, supporting the theoretical guarantee of Corollary 1. Second, the experiment with $\Delta = 0.05$ highlights the critical role of the clipping parameter $\rho$ in stabilizing performance under the anytime rate: while the average regret under $\eta_t = 0.5 \wedge \sqrt{\log(t)/t}$ is sublinear, a significant fraction of trajectories suffer nearly linear regret. This effect is substantially mitigated by setting $\rho = 0.1$ instead of $\rho = 0.5$. Interestingly, this improvement arises despite the fact that $\eta_t$ drops below $0.1$ naturally (without clipping) after approximately 650 steps, suggesting that early iterations with an overly aggressive learning rate can severely compromise the algorithm's trajectory.

This observation echoes findings from prior work on the sensitivity of SGB to initialization [14, 15]. In particular, as suggested by Theorem 2, if SGB is run with a poor learning rate up to time $t_0$, then with non-negligible probability it can reach a state where $p_{1,t_0} \lesssim t_0^{-1}$, making recovery exceedingly slow.

**$K$-armed bandits** The insights presented for two-armed bandits extend to our third experiment with $K = 5$ and $\Delta = 0.25$, where the use of a time-varying learning rate again leads to high variance in regret across trajectories, an effect that only tighter clipping can avoid. In this setting, it takes approximately 4000 steps for the un-clipped learning rate to fall below the oracle tuning.

Overall, these experiments emphasize that phases of substantial **over-estimation** of the learning rate should be avoided at all costs to ensure reliable performance. If no information about $\Delta$ is available, tuning the learning rate using the final horizon $T$ appears more robust, and the case where $T$ is unknown can be handled using a standard doubling trick. More generally, any use of excessively large learning rates, even briefly, can result in high variance across runs. Nonetheless, these experimental findings remain consistent with the theoretical results established in this paper, as the regret (in the sense of Eq. (1)) is defined *in expectation*, and the sublinear guarantees for time-varying learning rates apply in that sense.

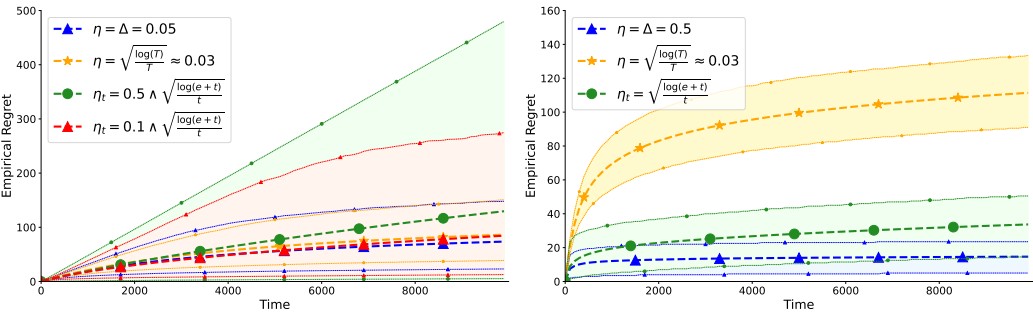

Figure 3: Average regret and $10 - 90\%$ percentiles up to horizon $T = 10^4$, $N = 5 \cdot 10^3$ trajectories on a bandit problem with means $\mu_1 = 0.05, \mu_2 = 0$ (Left), $\mu_1 = 0.5, \mu_2 = 0$ (Right).

## G.2 Comparison with standard bandit policies

We now benchmark the performance of SGB against several standard bandit policies. Motivated by the literature review in Appendix A.3, we include a range of algorithms that exemplify different frameworks for balancing exploration and exploitation:

- UCB, in its standard implementation [18, Figure 1].
- Thompson Sampling (TS), using a Beta prior/posterior setup [19, 46], and a uniform prior.
- MED, implemented with the Bernoulli KL-divergence [49, 53].

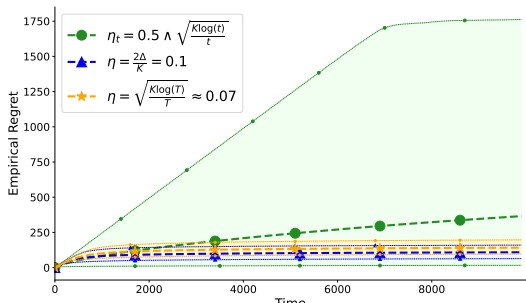

Figure 4: Average regret and $10 - 90\%$ percentiles up to horizon $T = 10^4$, $N = 5 \cdot 10^3$ trajectories on the bandit with means $\mu_1 = 0.25$, $\mu_2 = \cdots = \mu_5 = 0$.

- `SAMBA`, as defined by Walton and Denisov [29]. While less standard, we include it as an alternative application of the policy gradient principle to bandits.

Since each of these policies is designed for Bernoulli rewards, we rescale the Rademacher rewards to binary format before passing them to the algorithm: each reward $r$ is transformed to $r' = 0.5 \cdot (r + 1)$. We emphasize that this transformation is **not** applied to `SGB`, which directly receives the original Rademacher rewards, $r \in \{-1, 1\}$.

Among the selected baselines, both `MED` and `TS` are known to be asymptotically optimal in the considered settings, in the sense that their regret upper bounds match the Lai & Robbins lower bound [36]. For optimism-based algorithms, we chose `UCB` over `KL-UCB`. While `KL-UCB` offers better asymptotic guarantees, on par with `TS` and `MED`, it requires solving an optimization problem at every time step which significantly increases its runtime. Given the scale of our experimental evaluation, spanning many settings and runs, this choice is primarily motivated by practical considerations. Additionally, benchmarking `SGB` against other algorithms (like `UCB`) that are not explicitly tailored to the KL-divergence structure of the environment offers a complementary perspective on its empirical performance.

**Parameters tuning** To ensure a fair comparison between `SGB` and `SAMBA`, we followed a consistent rule for setting their respective learning rates. The experiments presented in Section 4 of the main paper indicate that, for both policies, the most effective strategy seems to consist in using the *critical learning rate predicted by theory* for each specific setup. For `SGB`, this leads us to choose $\eta = \frac{2\Delta}{K}$, which we conjecture to be the critical threshold for $K \geq 2$ (see Appendix D.3). For `SAMBA`, we set $\alpha = \frac{\Delta}{2}$. As discussed in Appendix B.2, Walton and Denisov [29] prove logarithmic regret for Bernoulli rewards when $\alpha = \frac{\Delta}{\mu_1 - \Delta} = \frac{\Delta}{\max_{k \geq 2} \mu_k}$. The factor $\frac{1}{2}$ in our tuning arises from the rescaling of rewards from the interval $[-1, 1]$ to $[0, 1]$, and we deliberately ignore the denominator to avoid introducing further assumptions on the mean values of the arms.

**Experiments** We consider four experimental setups designed to illustrate the comparative performance of `SGB` across a range of scenarios:

- In Figure 5 (Left), we consider a setting with $K = 5$ arms, all with means close to 0, and where each sub-optimal arm has the same gap $\Delta = 0.1$.
- In Figure 5 (Right), we consider a setting with $K = 10$ arms. The minimum gap is $0.2$, while the other sub-optimal arms have significantly larger gaps: $0.5$ for two arms and $1$ for five arms.
- In Figure 6 (Left), we consider $K = 5$ arms with very low rewards, with $\mu_1 = -0.8$.
- In Figure 6 (Right), we consider $K = 4$ arms with high rewards, with $\mu_1 = 0.96$, and the worst arm having mean $0.9$.

The first setup illustrates that, when the problem is fully aligned with the oracle information used by `SGB` (i.e., $\Delta$ is known and exactly matches the gap of **all** sub-optimal arms), the performance of `SGB` closely matches that of state-of-the-art policies such as Thompson Sampling. Although `TS` achieves

this without oracle knowledge, the experiment shows that, for well-specified problems, `SGB` can be highly competitive with proper tuning.

In contrast, the second setup highlights a limitation: when the sub-optimality gaps vary significantly, `SGB` becomes sub-optimal compared to optimized policies like `MED` and `TS`, which implement separate exploration mechanism for each arm. While it still performs reasonably well (e.g., better than `UCB`), the fixed learning rate results in over-exploration of significantly sub-optimal arms.

Finally, the last two experiments in Figure 6 demonstrate that although `SGB` outperforms `UCB` in scenarios with extreme means (close to the boundaries of the $[-1, 1]$ support), it cannot compete with approaches like `MED` and `TS`, which leverage KL-divergences or tailored posterior distributions to better adapt to the geometry of the problem.

In all setups, `SGB` compares favorably to `SAMBA` in terms of both average regret and the $10$–$90\%$ empirical regret range. We believe the parameter tuning of both methods is fair with respect to their theoretical guarantees and the level of oracle information used. A more detailed comparison with `SAMBA` is provided in Appendix G.4.

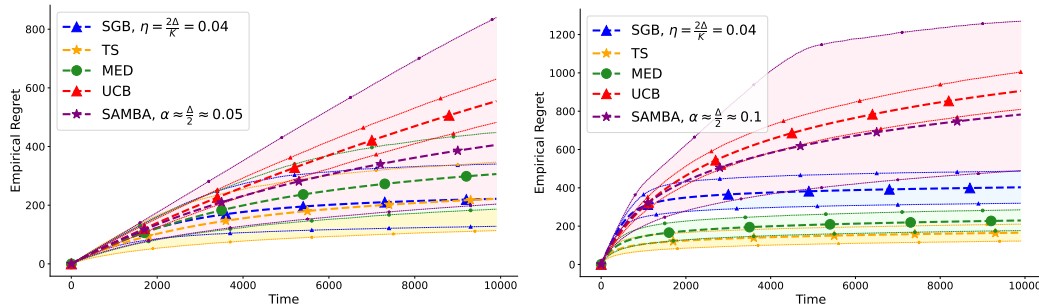

Figure 5: Average regret and $10 - 90\%$ percentiles up to horizon $T = 10^4$, $N = 5 \cdot 10^3$ trajectories on the bandit problems with averages $\mu_1 = 0.1, \mu_2 = \cdots = \mu_5 = 0$ (Left), and $\mu_1 = 0.5, \mu_2 = 0.3$, $\mu_3 = \cdots = \mu_8 = 0$, and $\mu_9 = \mu_{10} = -0.5$ (Right).

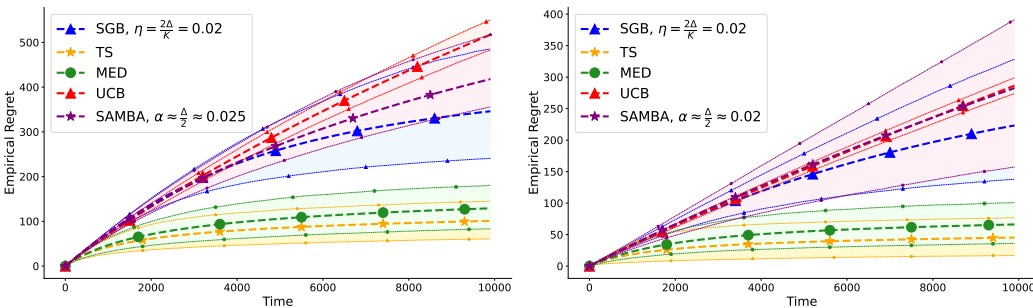

Figure 6: Average regret and $10 - 90\%$ percentiles up to horizon $T = 10^4$, $N = 5 \cdot 10^3$ trajectories on the bandit problems with averages $\mu_1 = -0.8, \mu_2 = -0.85, \mu_3 = \mu_4 = -0.9 = 0, \mu_5 = -0.95$ (Left), and $\mu_1 = 0.96, \mu_2 = \mu_3 = 0.92, \mu_4 = 0.9$ (Right).

### G.3   Asymptotic tightness of Theorem 1

We propose a set of experiments designed to verify the tightness of the $\log(T)$ factor in the regret upper bound of Theorem 1 for $K = 2$, and to support the conjecture that the bound of Lemma 2 for general $K$ might be tightened by a factor $\frac{K-1}{K}$.

**Two-armed bandits**   For the case $K = 2$, we consider a problem with means $\mu_1 = 0$ and $\mu_2 = -0.25$, so that $\Delta = 0.25$. We perform 1000 independent runs of `SGB` over a horizon $T = 10^5$, for three fixed learning rates: $\eta \in \{0.05, 0.1, 0.2\}$. Each choice of $\eta$ lies within the range of validity of Theorem 1. In Figure 7, we plot the empirical regret (computed as $t \mapsto \widehat{\mathcal{R}}_t^M$, see Eq. (31) with

$M = 10^3$) in logarithmic scale, along with the theoretical curves $t \mapsto \frac{\log(t)}{2\eta}$ corresponding to each learning rate.

To facilitate the comparison of the asymptotic slopes, we represent the $x$-axis in logarithmic scale, and each theoretical curve is vertically shifted to match the empirical regret at time $T$. That is, we plot

$$t \in [T] \mapsto \frac{\log(t)}{2\eta} + \widehat{\mathcal{R}}_T^M - \frac{\log(T)}{2\eta} \ .$$

The figure confirms the tightness of the bound in Theorem 1 for this particular setup: for all three values of $\eta$, the empirical regret closely follows the theoretical prediction as $T$ increases, thus supporting the sharpness of the logarithmic term in the regret bound.

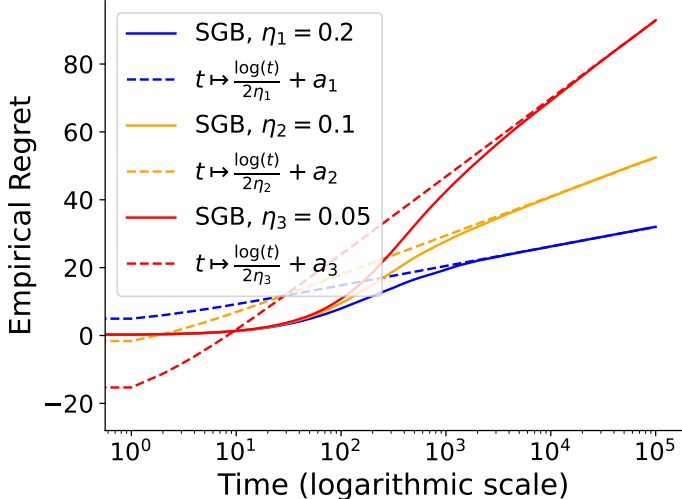

Figure 7: Empirical regret of SGB in logarithmic scale up to horizon $T = 10^5$, averaged over $N = 10^3$ independent trajectories on the two-armed bandit problem with means $\mu_1 = 0$ and $\mu_2 = -0.25$. Results for three learning rates $(\eta_i)_{i \in [3]} \in \{0.05, 0.1, 0.2\}$, where the offsets $(a_i)_{i \in [3]}$ are chosen so that each comparator curve matches the empirical regret at time $T$.

**$K$-armed bandits.** We now consider two experimental setups with $K > 2$. In the first, with $K = 4$ arms, we set $\mu_1 = 0$ and choose sub-optimal arms with mean $-0.4$, so that all arms share the same sub-optimality gap. In the second setup, with $K = 5$, we choose $\mu_1 = 0.9$, $\mu_2 = 0.65$, and $\mu_3 = \mu_4 = \mu_5 = -1$, so that the minimal gap is $\Delta = 0.25$, while some arms have significantly larger gaps.

The results, reported in Figure 8, show that in both cases the regret appears to scale asymptotically as $\frac{K-1}{K} \cdot \frac{\log(T)}{\eta}$. While this behavior can be expected in the first setup, where all sub-optimal arms are symmetric, the second case is more surprising. We refer to discussions in Appendix D.2 for additional insights. Based on these observations, we formulate the following conjecture.

**Conjecture 3.** *For any* bandit problem $\nu \in \mathcal{F}^K$, in the *post-convergence regime,* ***all sub-optimal arms*** $k \in [K]$ *are explored at a rate* $\frac{1}{K} \cdot \frac{1}{\eta \Delta_k t}$, *and thus contribute equally to the overall (asymptotic) regret, each by a term* $\frac{1}{\eta K} \log(T)$.

Note that this conjecture is stronger than the conjectures formulated in Appendix D.3: Conjecture 1 could be proved as a consequence of Conjecture 3, while the latter would considerably help in proving Conjecture 2, as discussed in Appendix D.3. Additionally, we can emphasize that in such Rademacher/bounded settings, optimized policies like TS and MED typically satisfy an asymptotic exploration rate of $\frac{1}{2\Delta_k^2 t}$ (up to some KL-divergence approximation), which matches the above rate if $\eta = \frac{2\Delta_k}{K}$. This is coherent with the result of the first experiment presented in Section G.2.

To further test the conjecture, we considered two additional setups, for which the results are reported in Figure 9. We considered instances with $K = 9$ and $K = 12$ respectively, with arms with various

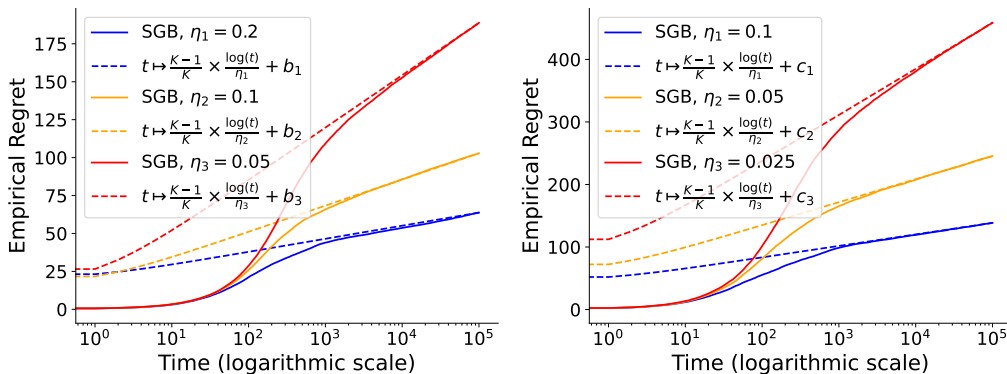

Figure 8: Empirical regret of SGB in logarithmic scale up to horizon $T = 10^5$, averaged over $N = 10^3$ independent trajectories on the bandit problems defined by $\mu_1 = 0$ and $\mu_2 = \mu_3 = \mu_4 = -0.4$ (Left) and $\mu_1 = 0.9$, $\mu_2 = 0.65$ and $\mu_3 = \mu_4 = \mu_5 = -1$ (Right). Results for three learning rates $(\eta_i)_{i \in [3]} \in \{\frac{\Delta}{2K}, \frac{\Delta}{K}, \frac{2\Delta}{K}\}$, where the offsets $(b_i)_{i \in [3]}$ and $(c_i)_{i \in [3]}$ are chosen so that each comparator curve matches the empirical regret at time $T$.

gaps. More precisely, the instances considered in the two experiments are parametrized by the following averages,

$$\mu_{\mathrm{xp.1}} = \{0.5, 0.2, 0.2, 0, 0, 0, -0.2, -0.2, -0.2\}, \quad \text{and}$$
$$\mu_{\mathrm{xp.2}} = \{0, -0.4, -0.4, -0.4, -0.5, -0.5, -0.5, -0.6, -0.6, -0.8, -0.8, -0.8\}.$$

We still observe the same asymptotic scaling as in previous experiments, which hints that the conjecture holds.

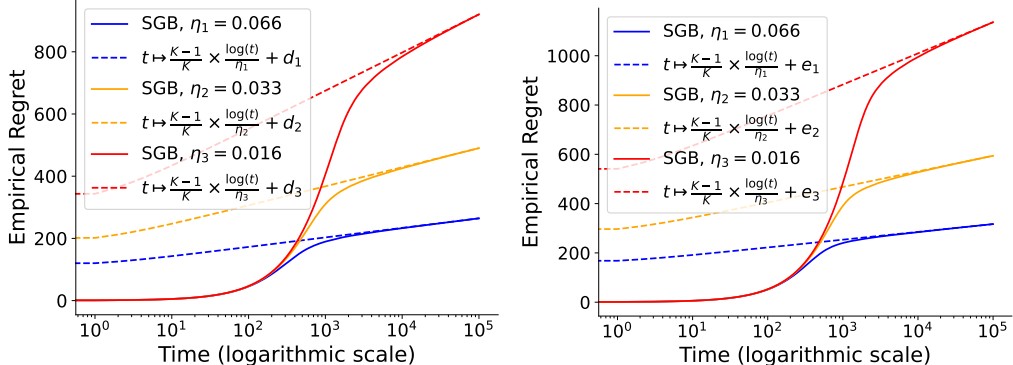

Figure 9: Empirical regret of SGB in logarithmic scale up to horizon $T = 10^5$, averaged over $N = 10^4$ independent trajectories on the bandit problems defined by averages $\mu_{\mathrm{xp.1}}$ (Left) and averages $\mu_{\mathrm{xp.2}}$ (Right). Results for three learning rates $(\eta_i)_{i \in [3]} \in \{\frac{\Delta}{2K}, \frac{\Delta}{K}, \frac{2\Delta}{K}\}$, where the offsets $(d_i)_{i \in [3]}$ and $(e_i)_{i \in [3]}$ are chosen so that each comparator curve matches the empirical regret at time $T$.

### G.4 Detailed comparison with SAMBA

We now isolate SAMBA and SGB, and propose experiments with the double objective of highlighting the differences between the two approaches, and of illustrating the discussion of Section 2 regarding the tuning of the learning rate in PG methods under additional moment conditions. We mainly focus on settings with $K = 2$, since the experiments presented in Appendix G.2 already show that SGB and SAMBA are not directly comparable for $K > 2$ in various scenarios.

In contrast with previous sections, where we considered Rademacher rewards, all problems in this subsection involve **Bernoulli rewards**. This model choice allows us to discuss the performance of

`SAMBA` directly in the setting under which it was originally studied in [29]. Contrarily to Rademacher rewards, this setting further allows us to design problems that exhibit interesting moment conditions, for instance by specifying low average rewards, as typically encountered in online advertising [34].

First, we compare `SAMBA` and `SGB` in configurations where their parameter tuning (denoted respectively by $\alpha$ and $\eta$) yields the same asymptotic regret bound. Indeed, while Theorem 1 gives a regret bound scaling as $\frac{\log(T)}{2\eta}$ for `SGB`, Theorem 1 of [29] gives a bound of $\frac{\log(T)}{\alpha}$ for `SAMBA`. Hence, in Figure 10, we present results for two experiments where we choose tunings $\alpha = \Delta$ and $\eta = \frac{\Delta}{2}$. While in both cases the asymptotic (logarithmic) scaling of the two policies appears to match—illustrated by parallel average regret curves in logarithmic scale—their pre-convergence behaviors differ significantly. Our results suggest that for these "equivalent" asymptotic tuning, `SAMBA` outperforms `SGB` in the settings considered.

However, while the parameter $\alpha$ is already maximized under no additional assumptions on $\max_{k \geq 2} \mu_k$, the learning rate $\eta$ of `SGB` could be increased further, up to $\Delta$. This illustrates that even if `SAMBA` and `SGB` share comparable post-convergence dynamics under appropriate choices of learning rates, as discussed from a theoretical perspective in Appendix B.2, their pre-convergence behaviors differ considerably. As such, the two policies may have distinct merits and limitations. In the figure, we retain `TS` from the benchmark policies introduced in Section G.2 to illustrate the performance of a state-of-the-art bandit policy on this problem.

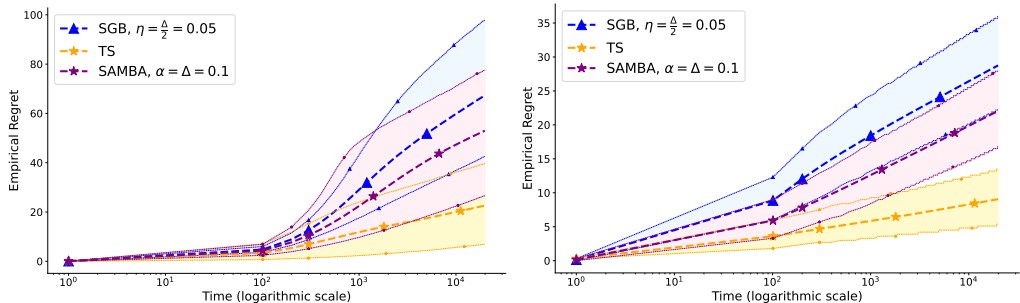

Figure 10: Empirical regret in logarithmic scale up to horizon $T = 2 \cdot 10^4$, averaged over $N = 10^4$ trajectories on the bandit problems defined by $\mu_1 = 0.6$ and $\mu_2 = 0.5$ (Left) and $\mu_1 = 0.4$, $\mu_2 = 0.1$ (Right). In both cases, `SGB` is tuned with $\eta = \frac{\Delta}{2}$ while `SAMBA` uses $\alpha = \Delta$.

Then, we illustrate how additional moment assumptions can allow `SAMBA` and `SGB` to perform well with learning rates larger than $\Delta$. We consider two problems with Bernoulli rewards with small average values. In a first experiment, with $K = 2$ arms, we use the same learning rate for both algorithms, $\eta = \alpha = \frac{\Delta}{\mu_1}$; in a second experiment with $K = 10$ arms and averages $\mu_1 = 0.05$ and $\mu_2 = \cdots = \mu_{10} = 0.03$, for which we still choose $\alpha = \frac{\Delta}{\mu_1}$ but adopt $\eta = \frac{2\Delta}{K\mu_1}$ in order to align with our analysis. We recall that the theoretical guarantees of `SAMBA` for this tuning of $\alpha$ hold for any number of arms $K$, thanks to a careful design of the policy that updates the leading arm differently from the others. We refer to Appendix B.2 for details. The results, presented in Figure 11, show that the additional oracle knowledge allows the PG algorithms to achieve a performance that can be comparable to Thompson Sampling. Notably, in the first experiment the regret statistics of `SGB` (average regret, 10–90% quantiles) closely match those of `TS`; and in the second experiment the average regret of both PG policies remain competitive, although `SAMBA` suffers larger variance across trajectories than `TS`. Nevertheless, we can further remark that in both cases the variance across trajectories is considerably smaller for `SGB` compared to `SAMBA`. Although the results do not suggest that PG methods should be preferred over `TS`—even under the additional assumptions we introduced—the fact that their performance is comparable to a policy that is optimized for the considered family of distributions is notable.

We then consider a third experiment with $K = 10$ arms, defined by the following means:

$$\mu_{\text{xp.3}} = \{0.1, 0.09, 0.09, 0.08, 0.08, 0.08, 0.05, 0.05, 0.05, 0.05\}, \quad \text{so } \frac{\Delta}{\mu_1} = 0.1,$$

In this setting, several sub-optimal arms have sub-optimality gaps up to five times larger than that of the second-best arm. As illustrated in Figure 12, and in line with the discussion of the second

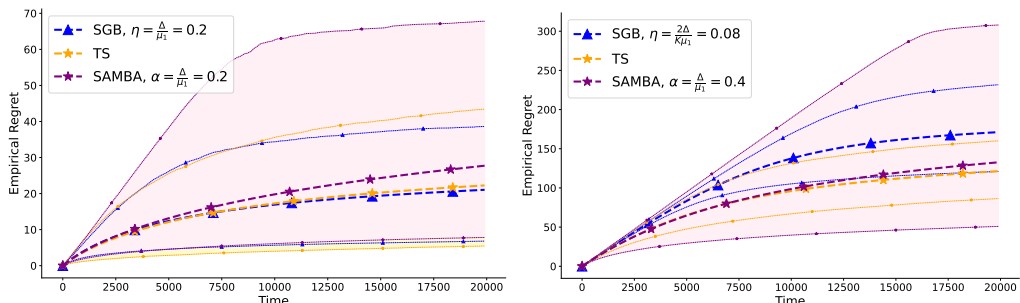

Figure 11: Empirical regret in logarithmic scale up to horizon $T = 10^4$, averaged over $N = 10^4$ trajectories on two bandit problems with small averages: $\mu_1 = 0.05$ and $\mu_2 = 0.04$ (Left) and $K = 10$, $\mu_1 = 0.05$, $\mu_3 = \cdots = \mu_{10} = 0.03$ (Right). In both cases, SGB is tuned with $\eta = \frac{2\Delta}{K\mu_1}$ while SAMBA uses $\alpha = \frac{\Delta}{\mu_1}$.

experiment in Figure 5, the performance of PG policies degrades (specifically, compared to optimal policies like TS) when the learning rate is tailored with respect to the second-best arm, while larger learning rates could improve performance for arms with larger gaps. However, considering arm-dependent learning rates would require oracle knowledge of *all* arm means, which might be unrealistic in many applications.

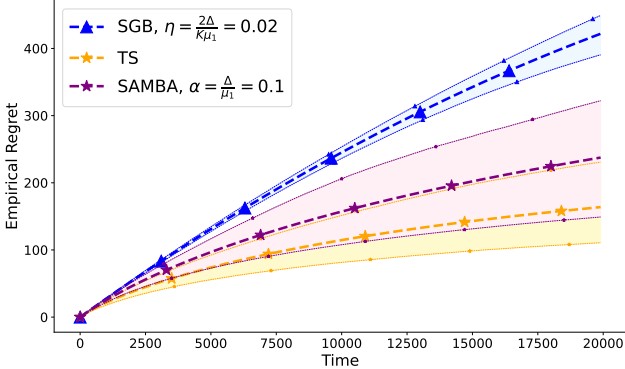

Figure 12: Empirical regret in logarithmic scale up to horizon $T = 10^4$, averaged over $N = 10^4$ trajectories on the bandit problem defined by $K = 10$ nd $\mu = \mu_{\text{xp.3}}$. SGB is tuned with $\eta = \frac{2\Delta}{K\mu_1}$ while SAMBA uses $\alpha = \frac{\Delta}{\mu_1}$.

Finally, in order to further understand why SGB compares less favorably to SAMBA as in the experiments of Section G.2, we investigated whether SGB could perform better when rewards are centered around $0$ (Rademacher) rather than always positive, as in the case of Bernoulli rewards. From a statistical perspective, an instance with Rademacher distributions of means $(\mu_k)_{k \in [K]}$ is strictly equivalent to an instance with Bernoulli distributions of means $\left(\frac{\mu_k+1}{2}\right)_{k \in [K]}$. However, the experiment presented in Figure 13 shows that the results of SGB tuned with $\eta = 2\Delta/K$ for Rademacher distributions are significantly better than the results of SGB with $\eta = \frac{\Delta}{K}$ for the corresponding Bernoulli distributions, which is nevertheless an "equivalent" tuning of SGB, more precisely both match the critical learning rate in their respective setting. This can be understood through Theorem 1, which shows that even though both algorithms use the same oracle knowledge, the logarithmic term of the bound in the Bernoulli case is twice as large as the bound in the Rademacher case.

This last result suggests that the performance of SGB is optimized when the rewards cover a centered interval like $[-1, 1]$, in particular if $\eta$ is tuned solely using that the rewards are bounded within a known range. Intuitively, it is natural to expect that the full potential of the theoretical tuning from Theorem 1 is achieved when the restriction on $\eta$ is tight due to upper bounding the failure regret is tight, which we proved to be the case for Rademacher arms (see the paragraph "tightness of the bound"

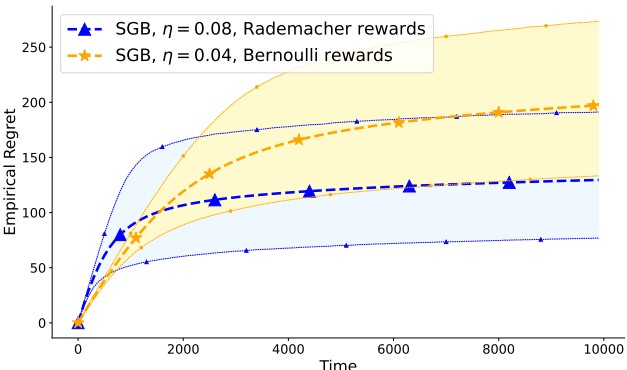

Figure 13: Empirical regret up to horizon $T = 10^4$, averaged over $N = 10^4$ trajectories on the five-armed bandit problem defined by $\mu_1 = 0.2$ and $\mu_2 = \cdots = \mu_5 = 0$ for the Rademacher instance, and the corresponding Bernoulli instance.

after Theorem 1). In scenarios where $\eta$ is tuned using additional moment knowledge—for instance, on the second moment $s^2$—any transformation that minimizes $\frac{\Delta}{s^2}$ would also lead to better empirical performance. For a closely related discussion, centered on the properties of adding baselines in `SGB` 's update, we refer to Section 6 of [1].

