# OpenReview forum: "Does Stochastic Gradient really succeed for bandits?"
_NeurIPS.cc/2025/Conference — NeurIPS 2025 oral_

### Official Review · Reviewer_kYCm · 2025-06-24

**Clarity:** 4
**Significance:** 3
**Originality:** 4
**Rating:** 5
**Confidence:** 3

**Summary:**

The paper revisits the SGB algorithm and provides a sharp characterization of when a constant learning rate yields log regret. For two-armed bandits it establishes a threshold at which the regret transitions from logarithmic to polynomial, while for K-armed bandits it shows the step size must scale with the gap divided by K. Synthetic experiments confirm a clear phase transition that validate the theory.

**Questions:**

1. Could an adaptive or decaying step-size schedule dispense with explicit gap knowledge while still attaining logarithmic (or better) regret? This question is especially pertinent for large $K$, where your bounds dictate a very small constant step size.

2. Does the analysis point to practical modifications, such as adding an exploration bonus (e.g., entropy regularization) or incorporating a simple variance-reduction technique, that would allow a larger step size without slipping into the polynomial-regret regime?

**Ethical Concerns:**

["NO or VERY MINOR ethics concerns only"]

**Final Justification:**

The authors have addressed my questions, and I will reserve my judgment on the paper.

**Quality:**

4

**Strengths And Weaknesses:**

The work offers an elegant, elementary analysis that pinpoints the precise learning-rate regimes in which constant-step SGB remains efficient. A new regret decomposition isolates a “post-convergence” term that always stays logarithmic from a “failure” term that explodes if the learning rate is too large. The results are clean and tight for the two-arm case and extend to the multi-arm setting. It is an overall solid theoretical paper.

---

> ### Author Rebuttal · Authors · 2025-07-25
>
> We thank the reviewer for their thoughtful and positive assessment of our work. We are particularly grateful for the insightful questions, which raise important directions for future research. Below we share our current intuitions regarding these points.
>
> 1. **On avoiding gap-dependent tuning via adaptive or decaying step sizes**
>
> Using a decaying or adaptive step-size schedule to bypass explicit gap-dependent tuning is a natural and appealing idea. However, it presents significant technical and conceptual challenges. Ideeed, estimating $\Delta$ is essentially as difficult as the bandit problem itself, so it might not be possible to adapt the step size (e.g. based on empirical estimates of the gap) fast enough to preserve logarithmic regret.
>
> In particular, the key challenge lies in controlling $\mathbb{E}[p_{1,t}^{-1}]$ during early stages where the step size might be too large. Without this control, it is possible that the policy drifts too far from optimality, and by the time the learning rate decays, the policy is effectively "trapped" in a bad region, requiring a prohibitively long time to recover. This phenomenon is visible in our experiments with time-decaying learning rates (Figures 3 and 4, Appendix G.1), where a poorly tuned initial step size leads to severe regret spikes on some trajectories.
>
> Our intuition is that using an increasing step-size schedule could offer a more robust solution: one could start with a small step size (e.g., $1/\sqrt{T}$) to avoid early failures, then gradually increase it toward a conservative estimate of the optimal learning rate (i.e. close to $\Delta$) as information is gathered. Investigating such schedules is an interesting direction for future work.
>
> 2. **On modifying the algorithm to tolerate larger step sizes**
>
>
> As discussed in our literature review, several prior works have considered softmax-based policy gradient algorithms with modifications such as entropy regularization or variance reduction. However, to the best of our knowledge, none of these methods achieve logarithmic regret with a constant step size.
>
> Whether such modifications could simultaneously avoid gap-dependent tuning and preserve optimal logarithmic regret rates remains a fundamental open question and doesn’t directly follow from the current analysis.
>
> ### _Regularization_
> Entropy regularization has been empirically observed to help with smoothing the objective function enabling policy gradient (PG) methods to escape flat regions and allowing the use of larger step-sizes [1]. Closest to the setup considered in our work is [2], which considers PG with softmax parameterized policies and adds a discounted entropy regularization w.r.t to the previous policies and proves $O(1/T^{1/2})$ convergence rate. The work of [3] introduced a more practical multi-staged algorithm based on entropy regularized PG that achieves $O(1/T^{1/3})$ convergence rate for tabular MDPs. A different line of work [4] considers stochastic cubic-regularized policy gradient that employs a 2nd order method with cubic-regularized updates and proves convergence with $O(1/T^{1/4})$ rate to a 2nd-order optimal policy .
>
> ### _Variance Reduction_
>
> Having stochastic updates with lower variance, e.g. as a result of a minibatch, is equivalent to considering an alternative problem, where we receive a reward with the same mean but decreased variance compared to the initial reward distribution. As such, since the knowledge of the second moment $s^2$ of the reward distribution can allow for more precise step-size tuning scaling as $\Delta/s^2$ (see related discussion in Section G.4, lines 2101-2104), stochastic gradient updates with lower variance would also allow for larger step-sizes. However, the design of a practical variance-reduction SGB that achieves logarithmic regret remains an interesting open question.
>
> The existing works can prove only polynomial convergence rate and do not achieve logarithmic regret. For example, the work of [6] develops a variance reduction method for PG that achieves provable $O(1/T^{1/2})$ convergence rate to optimal policy. Another work [5] shows convergence rate of $O(1/T^{3/5})$, i.e., an improvement of $T^{1/10}$ but with more restrictive assumptions.
>
>
> [1] Ahmed Z., Le Roux N., Norouzi M., Schuurmans D., Understanding the impact of entropy on policy optimization, 2018
>
> [2] Ding Y., Zhang J., Lee H., Lavaei J., Beyond Exact Gradients: Convergence of Stochastic Soft-Max Policy Gradient Methods with Entropy Regularization, 2024
>
> [3] Lu M., Aghai M., Raj A., Vaswani S., Towards Principled, Practical Policy Gradient for Bandits and Tabular MDPs, 2024
>
> [4] Wang P., Wang H., Zheng N., Stochastic cubic-regularized policy gradient method, 2022
>
> [5] P. Xu, F. Gao, Q. Gu, An improved convergence analysis of stochastic variance-reduced policy gradient, 2020
>
> [6] Zhang J., Chegzhuo N., Zheng Y., Szepesvari C., Wang M., On the Convergence and Sample Eﬃciency of Variance-Reduced Policy Gradient Method, 2021

---

### Official Review · Reviewer_ijbh · 2025-06-28

**Clarity:** 3
**Significance:** 3
**Originality:** 3
**Rating:** 5
**Confidence:** 3

**Summary:**

The paper explores the Stochastic Gradient Bandit policy with a focus on the dependence of the regret on the learning rate $\eta$. Relying on the work of Mei et al, 2023,2024, it exhibits different regret regimes depending on the positioning of $\eta$ as compared to the reward gap $\delta$ through a decomposition of the regret in two terms. Starting with a study of the 2-arms instance (upper bound then lower bound for larger $\eta$), it then studies the case where $K\geq 2$. Deepening our understanding of SGB, I believe that it is a solid theoretical contribution to the field.

**Questions:**

Could the authors provide a more detailed paragraph that would help to understand the (possible?) link between policy gradient algorithms for RL and SGB?

Is there hope for really practical applications of SGB?

**Ethical Concerns:**

["NO or VERY MINOR ethics concerns only"]

**Final Justification:**

I was in favor of acceptance, which explains the score that I gave to the paper. The discussion phase with the authors and the reviewers confirmed this feeling.

**Limitations:**

The paper explores a setup proposed in previous works: in that sense, it is not extremely novel.

**Paper Formatting Concerns:**

No.

**Quality:**

4

**Strengths And Weaknesses:**

I appreciate the following strengths of the paper:
- it is an interesting contribution to improve the understanding of SGB, a promising area of the bandit literature.
- there is a **very solid** amount of theoretical work to obtain the theorems of the paper, also supported by some synthetic experiments.
- I appreciate the fact that it starts with the 2-arms case to help the understanding before diving in the more complex theorems.

On the weaknesses side, I have the following concern:
- the initial setup of SGB could be more clearly formulated at the beginning of the paper (even though it is mostly a matter of (re)organization).

---

> ### Author Rebuttal · Authors · 2025-07-25
>
> We thank the reviewer for their positive and constructive feedback. We are glad that you found our theoretical contribution solid and appreciated the presentation. We address your comments and questions below.
>
> 1. **Clarity of the introduction of SGB**
>
> We agree that our introduction of SGB is a bit abrupt. We made this presentation choice because the algorithm is simple and well established in the literature (e.g., Sutton & Barto, Chapter 2.8), and we thought that providing the pseudo-code (Algorithm 1 in Appendix A.1), along with Equations (3) and (4), would suffice to understand it. To improve clarity, we will add a short derivation of the gradient in Appendix A.1 to illustrate how SGB corresponds to gradient ascent on the expected reward under a softmax parameterization, with estimated gradients. We hope this addition helps fully clarify the update rule in Equation (4).
>
> 2. **Relation between SGB and policy gradient algorithms**
>
> Apart from the bandit problem, policy gradient method with softmax parameterization (softmax policy gradient) is a widely used approach applied also in other settings, e.g. reinforcement learning (RL) and tabular MDPs:
> * In RL, policy gradient methods were shown to achieve effective empirical performance [1] and are successfully used in real-world applications such robotics [7] and large language models [8]. The SGB algorithm is directly comparable to using policy gradient method with softmax parameterization applied to the Markov Decision Process (MDP) with a single step, as studied more generally in [2, 3].
> * For tabular MDPs (a setting where the optimal policy is contained in the class of paramterized policies), the SGB algorithm is also comparable to the use of softmax policy gradient whose convergence has been studied in [4,5].
>
> Another interesting point of comparison is the recent work [6], which studies softmax PG in the bandit setting (i.e. the same as the SGB algorithm in our paper), but with low-dimensional parameterization of the probability vector: $\mathrm{softmax}(X\theta)$, where $X\in\mathbb{R}^{K\times d}$ is the feature matrix ($K$ is the number of arms, $d$ is the dimension of the feature space) and $\theta\in\mathbb{R}^d$ is the low-dimensional paramterization, i.e., $d<K$.
>
>
> [1] Schulman J., Wolski F., Dhariwal P., Radford A., Klimov O., Proximal policy optimization algorithms, 2017
>
> [2] Agarwal, A., Kakade, S. M., Lee, J. D., Mahajan, G., Optimality and approximation with policy gradient methods in Markov decision processes, 2019.
>
> [3] Mei J., Xiao C., Szepesvari C., Schuurmans D., On the Global Convergence Rates of Softmax Policy Gradient Methods, 2022
>
> [4] Agarwal, A., Kakade, S. M., Lee, J. D., and Mahajan, G., On the theory of policy gradient methods: Optimality, approximation, and distribution shift, 2021.
>
> [5] Lu M., Aghai M., Raj A., Vaswani S., Towards Principled, Practical Policy Gradient for Bandits and Tabular MDPs, 2024
>
> [6] Lin M. Q., Mei J., Aghaei M., Lu M., Dai B., Agarwal A., Schuurmans D., Szepesvari C., Vaswani S., Rethinking the Global Convergence of Softmax Policy Gradient with Linear Function Approximation, 2025
>
> [7] Kober J., Bagnell J. A., Peters J., Reinforcement learning in robotics: A survey, 2013
>
> [8] Uc-Cetina V., Navarro-Guerrero N., Martin-Gonzalez A., Weber C., Wermter S., Survey on reinforcement learning for language processing, 2025
>
>
> 3. **Practical relevance of SGB**
>
> While our focus is theoretical, we do believe that the insights from this study could have practical implications. In bandit settings, the simplicity of SGB, as a fast-to-update gradient-based policy makes it attractive, provided its limitations are well understood. Our findings can inform applications requiring quick updates and data storage constraints, such as online advertising or personalized content recommendation. The analysis characterizes conditions under which constant-step SGB can be safely used, while highlighting the risks of improper tuning. These insights could inform more robust and reliable deployments of softmax-based learning rules. Furthermore, we hope that the regret decomposition introduced in our analysis will inspire new algorithmic variants with improved empirical stability and performance.

---

> ### Comment · Reviewer_ijbh · 2025-08-03
>
> I thank the authors for their clarifying answer to my review and keep my score.

---

### Official Review · Reviewer_SAiY · 2025-07-01

**Clarity:** 3
**Significance:** 2
**Originality:** 4
**Rating:** 5
**Confidence:** 4

**Summary:**

The paper considers "stochastic gradient bandits", a simple instance of the stochastic policy gradient method where the actions are sampled from a softmax policy whose parameters are updated via a specific randomized estimate of the gradient of the value of the policy. This algorithm is noteworthy due to its simplicity. Recent previous work has established that any constant learning rate guarantees convergence to the optimal policy, and a small enough (but "gap-dependent") learning is sufficient to get an O(1/t) convergence of the policy (implying a log(t) regret bound). This paper refines these previous results in a number of ways:
(1) It introduces a novel analysis technique that resembles potential function (or Lyapunov function) techniques used in online learning and beyond to directly analyze the regret, rather than analyzing policy convergence.
(2) With this tool in place, for the 2-armed case, it establishes that if the constant stepsize is large compared to the size of the gap between the rewards of the two arms then the regret will be polynomial on some instances.
(3) Also for the 2-armed case, it also establishes that for a constant stepsize that is smaller than the gap, the regret will be logarithmic.
(4) The paper also shows that for the K-armed setting, the regret will be polynomial if the stepsize is large relative to 1/K.
(5) Preliminary results are given for upper bounding the regret for the general K-armed setting.
(6) Experimental results illustrate the findings (and in particular, highlight that the phenomenon exploited in the lower bound proofs are real).

**Questions:**

With some simplification, Theorems 1 & 2 suggest that there are two regimes for the regret.

For eta<Delta, the regret is min(Delta T,log(T)/Delta).
For eta>Delta, the regret is min(Delta T, T^{1- (Delta/(2eta))}).

Now, if I fix some eta, compared to this there are those problems where the gap is small. This seems to be the second case above (realm of Theorem 2).
Here, if Delta goes to zero, the minimum is always Delta T. Hence, for very small values of Delta, Theorem 2 is not relevant.

So it seems that Theorem 2 says that there is an intermediate range of values of Delta, such that regret is nontrivially large.

I guess what really goes on here is that both theorems are essentially asymptotic (they only hold for T > f(1/Delta) for some increasing function f). This sorta means that Delta needs to be large anyways to use the theorems.

I guess I am struggling to understand a little the implications of these results.

To formulate this as a question, what is the best way to understand what these results are telling us? Simply put, say, I use eta = 1/sqrt{T} or something like this. When should I worry about the performance of this algorithm (T is fixed)?

**Ethical Concerns:**

["NO or VERY MINOR ethics concerns only"]

**Final Justification:**

This paper makes an interesting contribution by changing the way policy gradient methods are analyzed. The main concern is whether the approach will be strong enough to handle the more interesting cases (eg K-armed bandits), however, I remain optimistic about that the paper will be useful for getting that done. A secondary minor concern is that the paper does not feel like a nice polished piece of work, but I trust the authors will be able to improve the paper to make it a nice contribution in the final revision.

**Limitations:**

n.a.

**Paper Formatting Concerns:**

no concerns

**Quality:**

3

**Strengths And Weaknesses:**

Strengths:
1) New, neat analysis ideas that perhaps others can build on
2) Nice "characterization" of how constant stepsize influences regret growth
3) The paper is quite readable

Weaknesses:
1) Regret is a weaker concept than policy convergence, though I would not say this is a big problem. Yet, because the paper focuses on regret, it won't have results on policy convergence. Perhaps the price to be paid for the nice analysis
2) Results for the K-armed case are "unfinished"

--------
Since there is no separate box for various miscellaneous comments, I will use this box for small comments/questions.

1) How important was that the parameters are initialized at zero? (Line 79)
2) Line 84: E_t must be standing for NOT E[.|H_t] but for E[.|H_{t-1}] from what I see.
3) Line 107: "fully characterize [..] except" sounds funny at best. Consider rephrasing
4) Displayed eqn, first line after line 134: Why is this not an equality?
5) Line 187 Theorem 2 and proof: This feels a bit messy. The theorem should be stating something like this? "For every eta, there exist Delta such that for all epsilon>0 and T large enough XYZ holds". Correct? The proof sketch is also messy (though eventually what is happening is clear). Perhaps consider rephrasing these. Related to this comment, see also a question below.
6) Line 202: "Lemma 6)" delete ")".
7) Line 247, Theorem 3: This should start with "Fix Delta>0"? Otherwise, can't I just use Theorem 2 to get this result by making Delta->0?
8) The histograms reminded me of similar observations made in the UCB-Tuned paper by Audibert et al. who also observe similar bimodal regret distributions. Perhaps it would be interesting to compare/contrast to this.

---

> ### Author Rebuttal · Authors · 2025-07-26
>
> We thank the reviewer for the detailed and thoughtful comments, as well as for highlighting the strengths of our work. Below, we respond to the specific questions and suggestions.
>
> W4. **regret vs policy convergence**
>
> We agree that the regret perspective is complementary to the recent convergence analysis of Mei et al. (referenced as [2] by reviewer 9S4k). In that work, the authors establish that (1) SGB converges asymptotically to the optimal policy, and (2) the average suboptimality decays as $\mathcal{O}(\frac{\log(T)}{T - \tau})$ after some (random) time $\tau$, corresponding to the moment when the policy enters (and stays in) a “nice” regime where the optimal arm is selected with high probability (Theorem 3 in their paper).
>
> Our interpretation is that proving a logarithmic regret bound essentially amounts to showing that $\mathbb{E}[\tau] < \infty$, while in the polynomial regime we identify, this quantity may be infinite. Thus, our regret analysis provides a finer-grained understanding of how constant step-sizes influence the *transient* behavior of SGB, not just its asymptotic convergence.
>
> 1. **initialization**
>
> Initializing the parameters at $0$ is not essential for our analysis. More generally, suppose the initial parameters are drawn from some distribution. In Theorem 1, for instance, the upper bound on the post-convergence term would simply include an additional term $\frac{\mathbb{E}[\theta_{0}]}{\eta}$. For the failure regret term, the expectation $\mathbb{E}[x_1] = 1$ used in the bound (l. 139) would be replaced by $\mathbb{E}\left[\frac{1 - p_{1}}{p_{1}}\right]$, where $p_{1}$ depends on the distribution of the initial parameters. The other results in the paper can be adapted in a similar fashion.
>
> 2. 3. 4. 6. **Typos and phrasing**
>
> Thank you very much for your careful reading, we will include the corresponding corrections in the revision.
>
> 5. **Theorem 2**
>
> Following your suggestions, we will clarify the statement of the theorem in the revision and the proof sketch. We will state the theorem as follows: "Consider any learning rate $\eta>0$ and minimum gap $\Delta \in (0,1)$. Then, if $\eta > \lambda_\Delta \coloneqq ...$, there exists a bandit instance defined by ... that for which the regret of SGB is polynomial, $R_T=\widetilde \Omega(...)$."
>
> 7. **Theorem 3**
>
> You're absolutely right, thank you for pointing out this subtlety. We will revise the statement by fixing $\Delta$, which leads us to adjust the condition on $\eta$ to $\eta \leq \frac{\log(1+\Delta)}{\Delta} \cdot \frac{1}{K - 1}$. This modification does not affect how we discuss or interpret the result in the rest of the paper, since the gap-dependent factor is bounded.
>
> 8. **Histograms**
>
> Thank you very much for pointing out this very interesting connection. We were not familiar with the paper by Audibert et al., but we did cite in appendix a recent paper by Fan & Glynn (ref. [60]) that makes a similar observation. We will add a sentence in the experimental section to highlight the connection to these works. More broadly, it appears that the bimodal distribution of regret is a feature inherent to algorithms that aim for logarithmic regret while managing a nonzero probability of failure. A similar bimodal behavior could likely be observed when running badly tuned instances of TS or UCB, e.g. for Gaussian distributions if the algorithms were tuned using underestimated variance parameters.
>
> **Question**
>
> Thank you for this question. For the two-armed case, we believe that Theorem 1 and Corollary 1 offer a comprehensive guideline on how to choose $\eta$ when using SGB in practice. In Corollary 1, we prove that a stepsize $\eta \leq 1/\sqrt{T}$ yields a regret of order $\sqrt{T}$, which provides a robust, horizon-dependent tuning of SGB.
>
> Then, Theorem 1 indicates that, if the learner has access to a bound $\Delta$ such that the gap-dependent regret bound of Theorem 1 (essentially, $\frac{\log(1+4\Delta^2T)}{2\Delta}$) is smaller than the $\sqrt{T}$ bound, they can opt for the gap-dependent tuning $\eta \approx \Delta$, which leads to much better performance when $\sqrt{T}$ is large compared to $\Delta^{-1}$.
>
> Since our regret bounds are quite tight, we believe this methodology for selecting $\eta$, guided by theoretical insights, is effective for maximizing empirical performance on finite horizons. We also expect the same approach to generalize to $K > 2$, assuming the critical learning rate is $\eta_c = \frac{2\Delta}{K}$ (Conjecture 1), and that logarithmic regret scales as $\frac{K-1}{K}\cdot\frac{\log(T)}{\eta}$ for $\eta < \eta_c$ (Conjecture 3, supported by experiments in Appendix G). In the revision we will also group these conjectures into a single conjecture in the main paper to help guide future research on this topic.
>
> We will reflect this discussion in the revision by expanding the discussion following Corollary 1, building on the current final sentence of the paragraph.

---

> ### Comment · Reviewer_SAiY · 2025-08-06
>
> Thanks for the clarifications. My questions have been satisfactorily addressed! Maybe just a small comment on whether it is "badly designed" bandit algorithms that show the bimodal behavior of regret. I think the issue is deeper: There is a tradeoff between expected regret and regret variance. I think Audibert and his students followed up the paper I mentioned exploring this (not unexpected) tradeoff. That the tradeoff manifests itself in this bimodal behavior is quite interesting though.

---

> > ### Author Response · Authors · 2025-08-06
> >
> > Thank you very much for sharing additional insights! We agree with your intuition, and we will carefully credit this line or work in the camera ready version of the paper.

---

### Official Review · Reviewer_9S4k · 2025-07-02

**Clarity:** 3
**Significance:** 3
**Originality:** 4
**Rating:** 5
**Confidence:** 4

**Summary:**

This paper considers the regret bounds of softmax policy gradient (SPG) for Multi-Armed bandits when using a constant learning rate that depends on the minimum reward gap of an bandit instance. Rather than analyzing the method using techniques from optimization, the authors instead consider the regret decomposition using the SPG update. As a result, the author show that depending how close the learning rate is to the reward gap, SPG either achieves logarithmic or polynomial regret. Additionally, the authors conduct experiments on simple bandit problems to support their theoretical results.

**Questions:**

1. This work mostly focuses on constant step-size since the objective satisfies a condition similar to the strong growth condition [1].
However, is it possible to technique introduce in the paper can be extended for decreasing step-sizes (or a reason why it cannot) for $K > 2$ arms?

2. Just to clarify, is not required to assume that all rewards are distinct, i.e. $\mu(i) \neq \mu(j)$ for $i \neq j$. This is a common assumption in prior work [2] and it's surprising to me that it's not not needed.

3. In the proof sketch of Lemma 1, why are the rewards deterministic?


[1] Mei, J., Zhong, Z., Dai, B., Agarwal, A., Szepesvari, C., & Schuurmans, D. (2023, July). Stochastic gradient succeeds for bandits.
[2] Mei, J., Dai, B., Agarwal, A., Vaswani, S., Raj, A., Szepesvári, C., & Schuurmans, D. (2024). Small steps no more: Global convergence of stochastic gradient bandits for arbitrary learning rates.

**Ethical Concerns:**

["NO or VERY MINOR ethics concerns only"]

**Final Justification:**

I recommend to accept this paper. My initial questions were fully addressed and am happy with the authors response. It would be nice to have an extension of Corollary 1 for $K >2 $ arms, but I believe their contributions are already sufficient for acceptance.

**Limitations:**

Yes, the authors sufficiently address the limitations.

**Quality:**

3

**Strengths And Weaknesses:**

**Strengths**
1. Provides a throughout analysis for SGD when using constants step-sizes and obtain regret bounds that do not depend on $\inf_{t \geq 1} \mathbb{E}[p_{1, t}]$ which appear frequently in prior work.
2. The paper is well-written and motivated. The proof and proof sketches in the main paper were digestible.
3. The empirical results support the theoretical findings.

**Weaknesses**
1. The results appear to require the rewards to have support $[-1, 1]$ which restricts common distributions such as Gaussian distributions
2. The results for $K >2$ arms seem to be fixated for constant step-sizes depending on $\Delta$ and does not seem to have analogous results such as Corollary 1.

**Clarity**
It would be good to explicitly mention that the smaller dashed-lines represented the percentiles in Figures 1 & 2.
**Originality**
I believe this work offers a novel technique for analyzing SPG methods when using constant step-sizes and introduces different view when considering such problems.

---

> ### Author Rebuttal · Authors · 2025-07-25
>
> We thank the reviewer for their thoughtful comments and are glad that you appreciated our results. Below, we address the specific questions and concerns raised.
>
> W1. **Assumption on reward support**
>
> Throughout the paper, we assume that rewards lie in $[-1,1]$ for clarity and simplicity, following a common assumption in the literature, particularly in prior work on SGB. However, our analysis can be extended under alternative assumptions, such as sub-Gaussianity. The key step where boundedness is used is Equation (8), where we apply a second-order Taylor expansion and upper-bound the quadratic term by leveraging the boundedness of $r\in [-1,1]$. If instead $r$ is $\sigma$-sub-Gaussian, we can use the standard bound
>  $$\mathbb{E}[e^{qr}]\leq  \mathbb{E}[e^{q \mathbb{E}[r]+\frac{q^2 \sigma^2}{2}}], $$
>
> and then apply a second-order expansion to this expression (e.g., assuming $|\mathbb{E}[r]|\leq 1$). The rest of the proof of Theorem 1 would follow similarly, yielding a slightly different condition on $\eta$ depending on the variance proxy $\sigma^2$ and $\Delta$.
>
> In summary, the boundedness assumption can be replaced by the assumption that rewards are sub-Gaussian, with means bounded in a known range (e.g. $[-1,1]$). We omitted this observation for space reasons in the submission but will include it in the final version.
>
> **Clarification on Figure 1 & 2**
>
> Thank you for the suggestion, we will update the captions accordingly.
>
> 1. **Decaying step-size for $K>2$**
>
> You are right that we do not provide a counterpart to Corollary 1 for $K>2$ in the main paper. This is because completing the regret bound in the constant step-size case (specifically, bounding the second term in Lemma 3) is also essential for extending the analysis to decreasing step sizes.
>
> Nevertheless, in Lemma 15 (Appendix E), we show that with a step size $\eta=1/\sqrt{T}$, the regret is $\mathcal{O}(K\sqrt{T})$ in the special case where all sub-optimal arms have the same gap. This is a promising first step, and we will highlight this result in the main text below Theorem 4 in the revised version.
>
> 2. **Do we require distinct means?**
>
> Our analysis does not require all rewards to have distinct means. In fact, Theorem 4 specifically addresses the case where all sub-optimal arms share the same mean. This indeed contrasts with some prior works (e.g., [2]) and suggests that such a separation assumption may not be necessary. Moreover, in Appendix F.1 (starting at line 1718), we briefly discuss how parts of our analysis can be adapted to handle multiple optimal arms.
>
> We believe that the general case with $K>2$  can be addressed without requiring this assumption, and plan to explore this in future work, by building on Lemma 3.
>
> 3. **Why are the rewards deterministic in Lemma 1's sketch?**
>
> Lemma 1 establishes a regret upper bound for *all* $\eta>0$ on an *easy instance* with deterministic rewards. Hence, on this instance SGB admits logarithmic regret for all learning rates. We use this result in Theorem 3 to show that this regret bound is “too good” to guarantee consistent performance over all instances. Specifically, by comparing it to the Lai & Robbins (1985) lower bound, we deduce that if $\eta>1/(K-1)$, the algorithm cannot guarantee logarithmic regret on all instances. More precisely, there exists a more difficult problem for which the regret must be polynomial.
>
> Theorems 2 and 3 provide, to our knowledge, the first necessary conditions in the literature on the constant learning rate $\eta$ for SGB to achieve logarithmic regret. However, we emphasize that these conditions are necessary but not sufficient. In Appendix F, we formulate conjectures suggesting that the true critical threshold for $\eta$ may in fact be $\frac{2\Delta}{K}$ for general $K \geq 2$.

---

> > ### Comment · Reviewer_9S4k · 2025-08-04
> >
> > Thank you for the clarifications. I will keep my rating and confidence as is.

---

### Decision · Program_Chairs · 2025-09-17

**Decision:**

Accept (oral)

**Comment:**

(a) Summary: The paper explores the stochastic gradient bandit algorithm. In two-armed case, it shows that a sharp threshold depending on the optimality gap $\Delta$ will determine whether the regret of SGD is logarithmic or polynomial. In the general $K$-armed cases, the learning rate should scale with $K$ to avoid polynomial regret.


(b) Main reasons for decision/Pros: Novel techniques are introduced and the findings will definitely help further exploration of the popular and important SGD technique.

(c) Change during rebuttal: The author(s) provided good response to allay the concerns from reviewers, especially the concerns about Definition 2.5 and Theorem 2.6 (lower bound). Some reviewers (e.g. Reviewer 5V1N) raised the rating during rebuttal. Such explanation should be included in the camera-to-ready version.

(c) Decision: I vote for Accept (oral) as the rating is 5 from all reviewers and I do believe this work is important. The author(s) may consider to implement some discussion during rebuttal in their camera-to-ready submission.